# An integrated group decision-making method under q-rung orthopair fuzzy 2-tuple linguistic context with partial weight information

**Fatima Abbas[1]☯, Jawad Ali[1]☯, Wali Khan Mashwani[1]☯, Muhammad I. Syam[2]***

**1** Institute of Numerical Sciences, Kohat University of Science and Technology, Kohat, KPK, Pakistan,
**2** Department of Mathematical Sciences, United Arab Emirates University, Al-Ain, UAE

☯ These authors contributed equally to this work.
* m.Syam@uaeu.ac.ae

**Data Availability Statement:** All relevant information needed to replicate the study in its entirety are within the paper.

## Abstract

Considering the advantages of q-rung orthopair fuzzy 2-tuple linguistic set (q-RFLS), which includes both linguistic and numeric data to describe evaluations, this article aims to design a new decision-making methodology by integrating Vlsekriterijumska Optimizacija I Kompromisno Resenje (VIKOR) and qualitative flexible (QUALIFLEX) methods based on the revised aggregation operators to solve multiple criteria group decision making (MCGDM). To accomplish this, we first revise the extant operational laws of q-RFLSs to make up for their shortcomings. Based on novel operational laws, we develop q-rung orthopair fuzzy 2-tuple linguistic (q-RFL) weighted averaging and geometric operators and provide the corresponding results. Next, we develop a maximization deviation model to determine the criterion weights in the decision-making procedure, which accounts for partial weight unknown information. Then, the VIKOR and QUALIFLEX methodologies are combined, which can assess the concordance index of each ranking combination using group utility and individual maximum regret value of alternative and acquire the ranking result based on each permutation's general concordance index values. Consequently, a case study is conducted to select the best bike-sharing recycling supplier utilizing the suggested VIKOR-QUALIFLEX MCGDM method, demonstrating the method's applicability and availability. Finally, through sensitivity and comparative analysis, the validity and superiority of the proposed method are demonstrated.

## 1 Introduction

Bike-sharing removes the "last mile" barrier to public transportation use and creates a synergistic impact with other modes of public transportation, solving the transportation industry's final "puzzle" [1]. Bike-sharing is a novel green and environmentally friendly sharing economy using the time-sharing rental model. Bike sharing (BS) has gained significant popularity due to its several benefits, which include reducing road congestion, lowering greenhouse gas emissions, and being convenient and fast. BS services of several brands, such as Harrow Bike,

**Funding:** The authors received no specific funding for this work.

**Competing interests:** The authors have declared that no competing interests exist.

Qingju Bike, Mobike, and others, have proliferated in an unending flood. Anyhow, the prompt rise of the BS industry has resulted in a staggering magnitude of wasted BS, resulting in a slew of unsolvable social challenges. Waste plastics and metals from discarded BS generate pollution and resource waste, while road occupation increases traffic congestion [2, 3]. As a result, recycling discarded sharing bicycles is a critical and pressing problem.

However, owing to the expensive rate of recycling and the fact that recycling is a non-core business for many BS operators, numerous prefer to disregard abandoned BS. Third-party logistics (3PL) providers are currently a proper solution to the challenge of recycling BS [4]. In particular, the BS operator should first negotiate a collaboration deal with the 3PL supplier, which should include service payment, service time, and service content. Because of the advantages of 3PL suppliers to organizations, for instance, lower operating expenses, greater emphasis on core business, and lower contribution, 3PL is becoming more popular. However, selecting a 3PL supplier is a highly difficult MCDM challenge. Furthermore, to the best of our knowledge, prior research on BS scarcely considers the choice of recycling supplier. After reviewing and summarising, the BS investigation concentrates mainly on site selection [5–7], demand prediction [8–10], and rebalancing approach [11, 12]. Mete et al. [5] conducted a case study focused on the Gaziantep University campus to identify potential station locations for students. Meanwhile, Mix et al. [7] introduced an integrated approach to model bike-sharing trip demand and determine optimal station placements within the system, utilizing variables related to the built environment and accessibility. Reference [6] introduced a two-stage station location and rebalancing vehicle service design approach to tackle the issue of demand ambiguity arising from potential biases and data loss. Zhao et al. [10] studied a hyper-clustering technique to improve the performance of a spatiotemporal deep neural network used for traffic prediction in the context of BS. A Bipartite station clustering algorithm was proposed by Li et al. [13]. This method was based on geographical distance as well as bike transition patterns between stations. It used geo-clustering and transition clustering iteratively until no additional changes in the clustering outcome were seen. On the other hand, Chen et al. [14] pioneered the development of a station correlation network within the cycling system. Stations were represented as nodes in this network, and connection weights showed the similarity in bike utilization trends across stations. In a recent study, Jin et al. [11] explored a simulation framework to evaluate various rebalancing and maintenance strategies. However, it's worth noting that, to the best of our knowledge, there has been no prior focus on the selection of recycling suppliers. Therefore, the choice of recycling suppliers for BS has become an overlooked issue that must be addressed. As a result, we chose the choice of BS recycling suppliers as one of the research topics in the present article.

The MCDM approach for choosing a BS recycling supplier involves numerous experts providing assessment information to suppliers based on multiple factors. However, due to the complication and variety of decision-making concerns in real life, the presence and non-ignorability of fuzziness have been disclosed. Fuzzy set (FS) theory [15] was developed and steadily added to deal with such situations. Because FS only has the membership grade (MG), it may be hard to narrate certain complex information [16–20]. As an expansion of the FS, Atanassov's intuitionistic fuzzy set (IFS) [21] enabled decision makers (DMks) with pair form of MG and non-membership grade (NMG) to illustrate their fuzziness. IFS considers that the total of the MG and NMG ordered pairs is less than or equal to one ($0 \leq v_1 + \mu_1 \leq 1$). Yager [22] propound the Pythagorean fuzzy set (PFS) to allow DMks better flexibility in coping with uncertainties. In the PFS, which is a generalization of the IFS, the square sum of MG and NMG is less than or equal to one ($0 \leq v_1^2 + \mu_1^2 \leq 1$). Yager [23] pioneered the q-ROFS, which states that the sum of the MG's and NMG's qth powers is less than or equal to one

$(0 \leq v_1^q + \mu_1^q \leq 1)$. Because they have a large MG and NMG description space, q-ROFSs are more effectual and valuable than IFSs and PFSs. Over the last couple of years, A vast study on q-ROFS has been considered by notable scholars and applied in numerous sectors [24–27]. For instance, various types of aggregation operators have been devised by the researchers such as Hamacher norm-based [28], Heronian mean [29], neutrality operational rules [24], normalized bidirectional projection [30], exponential operation rules [31], to overcome the challenges [32–35].

The processing mode of 2-tuple linguistic (2TL) learning can effectively evade information distortion and loss. The scholars in [36] presented the 2TL interpretation paradigm, which is one of the essential approaches for addressing human language decision-making challenges. Numerous decision-making methodologies and 2TL aggregation operators are later created. Deng et al. [37] put forward the generalized 2TL Pythagorean fuzzy Heronian mean aggregation operators by combining the generalized Heronian mean aggregation operators and their weighted version with 2TL Pythagorean fuzzy numbers. In [38], Wei and Gao proposed several Pythagorean fuzzy 2TL power aggregation operators to sort out the MCDM challenges by combining power geometric and power average operations with Pythagorean fuzzy 2TL data. Wei et al. [29] introduced the doctrine of q-rung orthopair fuzzy 2TL (q-ROFTL) sets to express the MG and NMG of an element to a 2TL variable along with its certain operational rules. They also propound various q-ROFTL Heronian mean aggregation operators. In accordance with q-ROFTL weighted averaging and q-ROFTL weighted geometric operators, Ju et al. [39] devised a method for resolving MAGDM problems with q-ROFTL input. They also offer the q-ROFTL Muirhead mean and the dual Muirhead mean operators.

The study of aggregation operators (AOs) is one of the most significant and challenging aspects of developing MCGDM algorithms [40–45]. Based on Hamy mean (HM) operations, Dend et al. [46] developed the 2-tuple linguistic Pythagorean fuzzy HM operator and its dual version. In Garg and Kumar [47] provided an MCGDM technique based on Einstein AOs and the possibility degree measure of linguistic intuitionistic fuzzy numbers. Peng et al. [48] introduced exponential operators and obtained acceptable results after evaluating the learning management system employing them. In a q-ROFSs context, Zhong et al. [49] used Dombi power partitioned weighted Heronian mean AOs to minimize the negative consequences of specific criteria degrees during the aggregation processing. Leveraging the advantages of the Choquet integral, Wan et al. [50] proposed the IVq-ROF Choquet integral operators, containing the average and geometric frameworks. Akram et al. [51] extended the combinative distance based assessment (CODAS) approach in accordance of 2-tuple linguistic Fermatean fuzzy Hamacher AOs. A study by Wang et al. [52] put forth several hesitant Fermatean 2-tuple linguistic weighted Bonferroni mean AOs by taking into account the interrelationship betwixt any two variables. They also examined some basic properties and peculiar cases of their expounded operators by varying parameter values. Ju et al. [39] developed a framework for solving MCGDM problem based on q-ROFTL Muirhead mean and its dual form. In [53], Yang et al. studied the q-ROFTL generalized Maclaurin symmetric mean AOs and utilized them in three-way decisions. However, these existing q-ROFTL AOs have the disadvantage of producing illogical outcomes in some cases, specifically when one of the q-rung orthopair fuzzy numbers considered has a NMG equal to 0 (or a MG equal to 0). Therefore, it is critical to create new AOs to overcome the limitations of the present ones.

The qualitative flexible multiple criteria (QUALIFLEX) methodology initiated by [54] is one of the efficient outranking ways to handle MCDM issues, specifically when the number of criteria significantly surpasses the range of options. It is contingent on pair-wise comparisons of alternatives about each criteria and determines the optimum permutation that maximizes the concordance/discordance index measure. It has garnered a significant amount of attention

and has been used in a variety of fields. Zhang [55] deployed a hierarchical QUALIFLEX approach based on the closeness index to obtain orders. Wang and his coauthors [56] evaluated green providers using a cloud model and QUALIFLEX. Considering criteria interaction, Liang et al. [57] put forth a heterogeneous QUALIFLEX method to tackle MCGDM dilemmas. The authors in [58] built a rough QUALIFLEX method and utilized it for selecting a sustainable shelter site in an uncertain environment. Banerjee et al. [59] expanded the QUALIFLEX approach to q-rung orthopair fuzzy setting based on bipolar Choquet integral. To assess the overall quality of operation employees in engineering projects, He et al. [60] studied Pythagorean 2-tuple linguistic fuzzy methodology to decision making process. However, the Pythagorean 2-tuple linguistic FS is incapable of dealing with components where the square sum of the MG and NMG is more than one. Thereby, to overwhelm this gap, the goal of our research is to not only expand the QUALIFLEX method in the q-ROFTL environment but also to integrate it with the VIKOR method (which stands for 'VlseKriterijumska Optimizacija I Kompromisno Resenje), because the sole utilization of VIKOR does not give the unique solution.

**Challenges:**

In view of the above literature review, several significant challenges prevalent in the current body of research can be delineated as follows:

**i).** The existing operations for q-RFLSs, as outlined in Wei et al. [29], which play a pivotal role in aggregating criteria values, appear to have certain limitations and deficiencies. This assertion will be substantiated by presenting concrete counterintuitive examples in Section 3.

**ii).** In numerous MCGDM problems, various DMKs have different weights due to differences in their seniority, expertise, or other factors. Due to the uncertainty of the index weights, it may be challenging to assign them weight values explicitly. To ascertain the weight of DMKs, a mechanism corresponding to the q-ROFTL setting is required.

**iii).** The existing q-RFL maximizing deviation model is only capable of operating when criteria weights are entirely unknown. However, it lacks the capability when confronted with scenarios where weight information is only partially known—a situation frequently encountered in real-world decision-making processes. Thus, the challenge of determining criteria weights from incomplete information remains unresolved.

**iv).** Numerous MCGDM approaches and their corresponding applications have been documented in the literature [39, 61, 62]. However, a noticeable gap exists regarding integrated approaches within the context of the q-RFL environment. This critical gap underscores the need for focused attention on the exploration and development of integrated approaches in the q-RFL setting.

**Contributions:**

To address the limitations of previous research, this work focuses on the following aspects:

**i).** Numerous counterexamples are presented to vividly illustrate the limitations inherent in the prior operational laws governing q-RFLSs. In a proactive effort to surmount these deficiencies, pioneering operational laws are introduced. Leveraging these novel operations as a foundation, the existing core AOs undergo a comprehensive reformulation. Furthermore, the redefined operators are accompanied by establishing noteworthy characteristics, including idempotency, monotonicity, boundedness, and commutativity.

**ii).** The technique for determining the weights of DMKs is intricately modeled using the deployed AOs as a foundational framework. Furthermore, the maximizing deviation

framework is improved to operate effectively in both scenarios, whether the weight information is entirely unknown or only partially known.

**iii).** To effectively address MCGDM challenges, we present the VIKOR-QUALIFLEX approach, which is grounded in the utilization of q-RFLSs. This approach enables the assessment of the concordance index for each ranking combination by considering both the group utility and the individual maximum regret value associated with each evaluation option. Consequently, it yields a more robust and stable evaluation outcome.

**iv).** An illustrative case concerning the supplier selection problem is examined to elucidate the implementation procedure of the devised approach. Subsequently, a comparative analysis is conducted with previous studies to underscore the validity and reliability of the proposed research.

The remaining sections are categorized as follows: Section 2 gives a concise overview of certain fundamental notions related to q-RFLS. In section 3, inefficiencies of available operations for q-RFLSs are highlighted via counterexamples, and new operational rules for q-RFLSs are proposed. Section 4 proposes modified AOs and discusses their characteristics at length. In Section 5, we describe the VIKOR-QUALIFLEX approach for q-RFLSs to tackle MCGDM problems. An illustrative case is bestowed in Section 6 to showcase the practicability and sensitivity analysis of the pioneered method. Afterward, a comparative analysis is carried out in Section 7, and some conclusions are drawn at the end of the article.

## 2 Background knowledge

In this section, we revisit several basic ideas regarding LTS and q-RFLS.

Suppose $S = \{s_\theta | \theta = 1, 2, \ldots, \ell\}$ represents an LTS with an odd cardinality. Any label, $s_\theta$ signifies the possible value for a linguistic variable, and it has to adhere to the stipulations [36, 63] listed below:

1. Ordered set: $s_{\theta_1} \leq s_{\theta_2} \Leftrightarrow \theta_1 \leq \theta_2$;

2. Negation operator: $Neg(s_{\theta_1}) = s_{\theta_2}$, such that $\theta_1 + \theta_2 = \ell$;

3. Max operator: $\max(s_{\theta_1}, s_{\theta_2}) = s_{\theta_1}$ if $s_{\theta_1} \geq s_{\theta_2}$;

4. Min operator: Min operator: $\min(s_{\theta_1}, s_{\theta_2}) = s_{\theta_1}$ if $s_{\theta_1} \leq s_{\theta_2}$.

For instance, $S$ can be defined as

$$S = \left\{ \begin{array}{l} s_0 = \text{extremely poor}, s_1 = \text{very poor}, s_2 = \text{poor}, s_3 = \text{medium}, \\ s_4 = \text{good}, s_5 = \text{very good}, s_6 = \text{extremely good} \end{array} \right\}.$$

In light of the concept of symbolic translation, Herrera and Martinez [36, 63] set up the 2-tuple fuzzy linguistic representation approach. It is employed to convey linguistic assessment information as a 2-tuple $(s_\theta, \triangle)$ where $s_\theta$ is a linguistic label from the pre-defined linguistic term set $S$, $\triangle$ is the measure of symbolic translation, and $\triangle \in [-0.5, 0.5)$.

**Definition 1** [36, 63] *Let $\vartheta$ be the calculated result of an aggregation of the indices of a set of labels assessed in an LTS S, i.e., the outcome of a symbolic aggregation operation, $\vartheta \in [1, \ell]$, with $\ell$ being the cardinality of S. If $r = round(\vartheta)$ and $\triangle = \vartheta - r$ are two numbers such that $r \in [1, \ell]$ and $\triangle \in [-0.5, 0.5)$, then $\triangle$ is known as a symbolic translation.*

**Definition 2** [36, 63] *Let $S = \{s_\theta | \theta = 0, 1, \ldots, \ell\}$ e an LTS and $\vartheta \in [1, \ell]$ be a numerical value representing the linguistic symbolic aggregation outcome. Then, the function $\lambda$ that retrieves the*

*2-tuple linguistic information equivalent to $\vartheta$ is then described as*

$$\lambda : [0, \ell] \longrightarrow S \times [-0.5, 0.5) \tag{1}$$

$$\lambda(\vartheta) = \begin{cases} s_r, & r = round(\vartheta) \\ \triangle = \vartheta - r, & \triangle \in [-0.5, 0.5). \end{cases} \tag{2}$$

*where round (.) is the conventional round function, $s_r$ is the index label closest to $\vartheta$, and $\triangle$ is the symbolic translation value.*

**Definition 3** [36, 63] *Let $S = \{s_\theta | \theta = 0, 1, \ldots, \ell\}$ be an LTS and $(s_r, \triangle)$ be a 2-tuple. There is always a function $\Upsilon$ can be described, such that, from a 2-tuple $(s_r, \triangle)$ it yield its equivalent numerical value $\vartheta \in [1, \ell]$, which is*

$$\Upsilon : S \times [-0.5, 0.5) \longrightarrow [0, \ell] \tag{3}$$

$$\Upsilon(s_r, \triangle) = r + \triangle = \vartheta. \tag{4}$$

**Definition 4** [29] *A q-RFLS $\mathcal{F}$ on a universal set $\mathbb{Z}$ is described as*

$$\mathcal{F} = \left\{ ((s_{r(z_i)}, \triangle(z_i)), \langle v(z_i), \mu(z_i) \rangle) \right\}, \tag{5}$$

*where $s_{r(z_i)} \in ; S$, $\triangle(z_i) \in [-0.5, 0.5)$, $v(z_i), \mu(z_i) \in [0, 1]$, with the restriction $0 \leq v^q(z_i) + \mu^q(z_i) \leq 1$ $(q \geq 1)$ $\forall z_i \in \mathbb{Z}$. The numbers $v(z_i), \mu(z_i)$ symbolize the grade of MG and grade of NMG of the element $z_i$ to the linguistic variable $(s_{r(z_i)}, \triangle(z_i))$, respectively. Furthermore, $\pi_{\mathcal{F}}(z_i) = 1 - (v;^q(z_i) + \mu^q(z_i))^{1/q}$ is named refusal grade, and the q-rung orthopair fuzzy 2-tuple linguistic number (q-RFLN) is marked by $\partial = ((s_r, \triangle), \langle v, \mu \rangle)$.*

**Definition 5** [29] *Let $\partial_1 = ((s_{r_1}, \triangle_1), \langle v_1, \mu_1 \rangle)$ and $\partial_2 = ((s_{r_2}, \triangle_2), \langle v_2, \mu_2 \rangle)$ be two q-RFLNs. Then, their basic operational rules are listed as follows:*

1. $\partial_1 \oplus \partial_2 = \left( \lambda(\Upsilon(s_{r_1}, \triangle_1) + \Upsilon(s_{r_2}, \triangle_2)), \left\langle (v_1^q + v_2^q - v_1^q v_2^q)^{1/q}, \mu_1 \mu_2 \right\rangle \right)$;

2. $\partial_1 \otimes \partial_2 = \left( \lambda(\Upsilon(s_{r_1}, \triangle_1) \cdot \Upsilon(s_{r_2}, \triangle_2)), \left\langle v_1 v_2, (\mu_1^q + \mu_2^q - \mu_1^q \mu_2^q)^{1/q} \right\rangle \right)$;

3. $\alpha \partial_1 = \left( \lambda(\alpha \Upsilon(s_{r_1}, \triangle_1)), \left\langle (1 - (1 - v_1^q)^\alpha)^{1/q}, \mu_1^\alpha \right\rangle \right)$ $\alpha > 0$;

4. $\partial_1^\alpha = \left( \lambda((\Upsilon(s_{r_1}, \triangle_1))^\alpha), \left\langle v_1^\alpha, (1 - (1 - \mu_1^q)^\alpha)^{1/q} \right\rangle \right)$ $\alpha > 0$;

5. $\partial_1^c = ((s_{r_1}, \triangle_1), \langle \mu_1, v_1 \rangle)$.

**Definition 6** [29] *Let $\partial_1 = ((s_{r_1}, \triangle_1), \langle v_1, \mu_1 \rangle)$ be a q-RFLN. Then its score function $\widehat{S}(\partial_1)$ and accuracy function $\widehat{A}(\partial_1)$ are described as:*

$$\widehat{S}(\partial_1) = \frac{\Upsilon(s_{r_1}, \triangle_1) \cdot (1 + v_1^q - \mu_1^q)}{2\ell}, \tag{6}$$

$$\widehat{A}(\partial_1) = \Upsilon(s_{r_1}, \triangle_1) \cdot (v_1^q + \mu_1^q). \tag{7}$$

**Definition 7** [29] *Let $\partial_1 = ((s_{r_1}, \triangle_1), \langle v_1, \mu_1 \rangle)$ and $\partial_2 = ((s_{r_2}, \triangle_2), \langle v_2, \mu_2 \rangle)$ be two q-RFLNs. Then, they can be compared according to the following laws:*

**1).** *If* $\widehat{S}(\partial_1) > \widehat{S}(\partial_2)$, *then* $\partial_1 \succ \partial_2$;

**2).** *If* $\widehat{S}(\partial_1) = \widehat{S}(\partial_2)$, *then*:

 **i).** *if* $\widehat{A}(\partial_1) > \widehat{A}(\partial_2)$, *then* $\partial_1 \prec \partial_2$;

 **ii).** *if* $\widehat{A}(\partial_1) = \widehat{A}(\partial_2)$, *then* $\partial_1 = \partial_2$.

**Definition 8** *Let* $\partial_\iota = ((s_{r_i}, \triangle_i), \langle v_i, \mu_i \rangle)(i = 1, 2, \ldots, m)$ *be a family of q-RFLNs, then q-rung orthopair fuzzy 2-tuple linguistic weighted averaging (q-RFLWA) operator is:*

$$q - RFLWA(\partial_1, \partial_2, \ldots, \partial_m) = \oplus_{i=1}^{m} w_i \partial_i$$

$$= \left( \curlywedge \left( \sum_{i=1}^{m} w_i \curlyvee \left( s_{r_i}, \triangle_i \right) \right), \left\langle \left( 1 - \prod_{i=1}^{m} (1 - v_i^q)^{w_i} \right)^{1/q}, \prod_{i=1}^{m} \mu_i^{w_i} \right\rangle \right) \tag{8}$$

*where* $w = (w_1, w_2, \ldots, w_\flat)^T$ *is the weight vector of* $\partial_i(\iota = 1, 2, \ldots, m)$ *such that* $w_i > 0$ *and* $\sum_{i=1}^{m} w_i = 1$. *Especially, if* $w = \left( \frac{1}{m}, \frac{1}{m}, \ldots, \frac{1}{m} \right)^T$, *then the q-RFLWA operator reduces to q-rung orthopair fuzzy 2-tuple linguistic averaging (q-RFLA) operator of dimension m, which is given as follows*:

$$q - RFLA(\partial_1, \partial_2, \ldots, \partial_m) = \frac{1}{m} \oplus_{i=1}^{m} \partial_i$$

$$= \left( \curlywedge \left( \frac{1}{m} \sum_{i=1}^{m} \curlyvee \left( s_{r_i}, \triangle_i \right) \right), \left\langle \left( 1 - \prod_{i=1}^{m} (1 - v_i^q) \frac{1}{m} \right)^{1/q}, \prod_{i=1}^{m} \mu_i^{\frac{1}{m}} \right\rangle \right). \tag{9}$$

**Definition 9** [62] *Let* $\partial_1 = \left( \left( s_{r_1}, \triangle_1 \right), \langle v_1, \mu_1 \rangle \right)$ *and* $\partial_2 = \left( \left( s_{r_2}, \triangle_2 \right), \langle v_2, \mu_2 \rangle \right)$ *be two q-RFLNs. Then the Hamming distance between* $\partial_1$ *and* $\partial_2$ *is defined as*

$$d(\partial_1, \partial_2) = \frac{1}{2(\ell + 1)} \left[ |(1 + v_1^q - \mu_1^q) . \curlyvee \left( s_{r_1}, \triangle_1 \right) - (1 + v_2^q - \mu_2^q) . \curlyvee \left( s_{r_2}, \triangle_2 \right) | \right]. \tag{10}$$

## 3 Novel q-ROPF2TLS operational laws

It is widely known from Section 2 that a q-RFLN is comprised of a linguistic 2-tuple and a q-rung orthopair fuzzy number (q-ROFN). Clearly, the operations of two q-RFLNs are based on those of two q-ROFNs. However, we have noted that the existing operations for q-RFLNs have some drawbacks. Therefore, the fundamental operations outlined in Definition 5 are not yet acceptable:

To disclose the shortcomings of the existing operational laws, the following examples are analyzed.

**Remark 1** *It is worth noting that throughout this manuscript, we will take*

$$S = \left\{ \begin{array}{c} s_0 = \text{extremely poor}, s_1 = \text{very poor}, s_2 = \text{poor}, s_3 = \text{medium}, \\ s_4 = \text{good}, s_5 = \text{very good}, s_6 = \text{extremely good} \end{array} \right\}.$$

*as the linguistic term set.*

**Example 1** *Let us consider two q-RFLNs $\partial_1 = ((s_3, 0), \langle 0.4, 0.7 \rangle)$ and $\partial_2 = ((s_3, 0), \langle 0.8, 0 \rangle)$. Then, using the basic operational laws [29] of q-RFLNs, we have $\partial_1 \oplus \partial_2 = ((s_6, 0), \langle 0.8352, 0 \rangle)$ which means that the non-zero non-membership has no impact on the output. This makes the operation "$\oplus$" unreasonable.*

**Example 2** *Let us consider two q-RFLNs $\partial_1 = ((s_3, 0), \langle 0.5, 0.8 \rangle)$ and $\partial_2 = ((s_5, 0), \langle 0, 1 \rangle)$. Then using the basic operational laws [29] of q-RFLNs, we get $\partial_1 \otimes \partial_2 = ((s_{15}, 0), \langle 0, 1 \rangle)$. Thus, the non-zero membership has no effect on output. Additionally, the subscript of the linguistic term in the obtained result is $15 > \ell = 6$, which could be nonsensical. The subscript of any linguistic term should lie in $[0, \ell]$. This makes the operation "$\otimes$" unreasonable.*

**Example 3** *Let us consider a q-RFLN $\partial_1 = ((s_2, 0), \langle 0.5, 0.5 \rangle)$ and $\alpha = 3$. Then using the basic operational laws [29] of q-RFLNs, we have $\partial_1^3 = ((s_8, 0), \langle 0.125, 0.875 \rangle)$. Evidently, the subscript of the linguistic term in the obtained result is $8 > \ell = 6$. This renders the existing operation nonsensical.*

**Example 4** *Let us consider a q-RFLN $\partial_1 = ((s_3, 0), \langle 0.5, 0.5 \rangle)$ and $\alpha = 3$. Then using the basic operational laws [29] of q-RFLNs, we have $3\partial_1 = ((s_9, 0), \langle 0.875, 0.125 \rangle)$. Clearly, the subscript of the linguistic term in the obtained result is $9 > \ell = 6$. This makes the current operation meaningless.*

To cope with the counterintuitive cases, in the following, we frame some novel operational laws of q-RFLNs based on prior research.

**Definition 10** *Let $S = \{s_\theta | \theta = 0, 1, \ldots, \ell\}$ be an LTS and let $\partial_1 = \left( \left( s_{r_1}, \triangle_1 \right), \langle v_1, \mu_1 \rangle \right)$ and $\partial_2 = \left( \left( s_{r_2}, \triangle_2 \right), \langle v_2, \mu_2 \rangle \right)$ be two q-RFLNs. Then, their basic operational laws are defined as follows:*

1. $\partial_1 \oplus \partial_2 = \left( \begin{array}{c} \curlywedge \left( (\ell + 1) \left( \begin{array}{c} \curlyvee \left( s_{r_1}, \triangle_1 \right) / (\ell + 1) + \curlyvee \left( s_{r_2}, \triangle_2 \right) / (\ell + 1) - \\ \curlyvee \left( s_{r_1}, \triangle_1 \right) \curlyvee \left( s_{r_2}, \triangle_2 \right) / (\ell + 1)^2 \end{array} \right) \right), \\ \left\langle \sqrt[q]{1 - \prod_{r=1}^2 (1 - v_r^q)}, \sqrt[q]{\prod_{r=1}^2 (1 - v_r^q) - \prod_{r=1}^2 (1 - v_r^q - \mu_r^q)} \right\rangle \end{array} \right)$;

2. $\partial_1 \otimes \partial_2 = \left( \begin{array}{c} \curlywedge \left( (\ell + 1) \left( \curlyvee \left( s_{r_1}, \triangle_1 \right) \cdot \curlyvee \left( s_{r_2}, \triangle_2 \right) / (\ell + 1)^2 \right) \right), \\ \left\langle \sqrt[q]{\prod_{r=1}^2 (1 - \mu_r^q) - \prod_{r=1}^2 (1 - v_r^q - \mu_r^q)}, \sqrt[q]{1 - \prod_{r=1}^2 (1 - \mu_r^q)} \right\rangle \end{array} \right)$;

3. $\alpha \partial_1 = \left( \begin{array}{c} \curlywedge \left( (\ell + 1) \left( 1 - \left( 1 - \curlyvee \left( s_{r_1}, \triangle_1 \right) / (\ell + 1) \right)^\alpha \right) \right), \\ \left\langle \sqrt[q]{1 - (1 - v_1^q)^\alpha}, \sqrt[q]{(1 - v_1^q)^\alpha - (1 - v_1^q - \mu_1^q)^\alpha} \right\rangle \end{array} \right) \quad \alpha > 0$;

4. $\partial_1^\alpha = \left( \begin{array}{c} \curlywedge ((\ell + 1)((\curlyvee(s_{r_1}, \triangle_1) / (\ell + 1))^\alpha)), \\ \left\langle \sqrt[q]{(1 - \mu_1^q)^\alpha - (1 - v_1^q - \mu_1^q)^\alpha}, \sqrt[q]{1 - (1 - \mu_1^q)^\alpha} \right\rangle \end{array} \right) \quad \alpha > 0$.

In order to comprehend the superiority of the diagnosed operations in what follows, the aforementioned examples are reconsidered.

**Example 5** *(Continued to Example 1) According to the proposed operational laws, the results are obtained as follows:*

1. $\partial_1 \oplus \partial_2 = ((s_5, -0.286), \langle 0.8352, 0.4200 \rangle)$;

2. $\partial_1 \otimes \partial_2 = ((s_1, 0.286), \langle 0.6387, 0.7000 \rangle)$;

3. $3\partial_1 = ((s_6, -0.306), \langle 0.6382, 0.7415 \rangle)$;

4. $\partial_1^3 = ((s_1, -0.4490), \langle 0.2996, 0.9313 \rangle)$.

Hence, the proposed operations of q-RFLNs presented in Definition 10 can overcome the limitations of Wei et al.' s operations, specifically the addition operation of q-RFLNs, as illustrated in Example 1.

**Example 6** *(Continued to Example 2) According to the suggested operational laws, the following outcomes are obtained*:

1. $\partial_1 \oplus \partial_2 = ((s_6, -0.143), \langle 0.5000, 0.8660 \rangle)$;

2. $\partial_1 \otimes \partial_2 = ((s_2, 0.143), \langle 0.0, 1 \rangle)$;

3. $2\partial_2 = ((s_6, 0.429), \langle 0.0, 1 \rangle)$;

4. $\partial_2^2 = ((s_3, -0.449), \langle 0.0, 1 \rangle)$.

Hence, the propound operations of q-RFLNs presented in Definition 10 can conquer the limitations of Wei et al.' s operations, specially the addition operation of q-RFLNs, as exemplified in Example 2.

**Example 7** *(Continued to Example 3) According to the novel operational laws, the results are obtained as follows*:

1. $\partial_1 \oplus \partial_1 = ((s_3, 0.429), \langle 0.75, 0.25 \rangle)$;

2. $\partial_1 \otimes \partial_1 = ((s_1, -0.4286), \langle 0.25, 0.75 \rangle)$;

3. $3\partial_1 = ((s_4, 0.449), \langle 0.875, 0.125 \rangle)$;

4. $\partial_1^3 = ((s_0, 0.1633), \langle 0.125, 0.875 \rangle)$.

**Example 8** *(Continued to Example 4) According to the novel operational laws, the results are obtained as follows*:

1. $\partial_1 \oplus \partial_1 = ((s_4, -0.286), \langle 0.75, 0.25 \rangle)$;

2. $\partial_1 \otimes \partial_1 = ((s_1, 0.286), \langle 0.25, 0.75 \rangle)$;

3. $3\partial_1 = ((s_6, -0.306), \langle 0.875, 0.125 \rangle)$;

4. $\partial_1^3 = ((s_1, -0.4490), \langle 0.125, 0.875 \rangle)$.

Hence, the diagnosed operations of q-RFLNs presented in Definition 10 can overcome the limitations of Wei et al.' s operations, specifically the scalar power operation of q-RFLNs, as indicated in Example 3.

**Theorem 1** *Let* $\partial_1 = \left( \left( s_{r_1}, \triangle_1 \right), \langle v_1, \mu_1 \rangle \right)$ *and* $\partial_2 = \left( \left( s_{r_2}, \triangle_2 \right), \langle v_2, \mu_2 \rangle \right)$ *be two q-RFLNs and* $\alpha, \alpha_1, \alpha_2 > 0$. *Then*

1. $\partial_1 \oplus \partial_2 = \partial_2 \oplus \partial_1$;

2. $\partial_1 \otimes \partial_2 = \partial_2 \otimes \partial_1$;

3. $\alpha(\partial_1 \oplus \partial_2) = \alpha\partial_1 \oplus \alpha\partial_2$;

4. $(\partial_1 \otimes \partial_2)^\alpha = \partial_1^\alpha \otimes \partial_2^\alpha$;

5. $(\alpha_1 + \alpha_2)\partial_1 = \alpha_1\partial_1 \oplus \alpha_2\partial_1$;

6. $\partial_1^{\alpha_1 + \alpha_2} = \partial_1^{\alpha_1} \otimes \partial_2^{\alpha_2}$.

**Proof.** From new operational laws

1. $\partial_1 \oplus \partial_2 =$

$$
\begin{pmatrix}
\curlywedge\left((\ell+1)\left(\Upsilon\left(s_{r_1}, \triangle_1\right)/(\ell+1) + \Upsilon\left(s_{r_2}, \triangle_2\right)/(\ell+1) - \Upsilon\left(s_{r_1}, \triangle_1\right)\Upsilon\left(s_{r_2}, \triangle_2\right)/(\ell+1)^2\right)\right), \\
\left\langle \sqrt[q]{1 - (1 - v_1^q)(1 - v_2^q)}, \sqrt[q]{(1 - v_1^q)(1 - v_2^q) - (1 - v_1^q - \mu_1^q)(1 - v_2^q - \mu_2^q)} \right\rangle
\end{pmatrix} =
$$

$$
\begin{pmatrix}
\curlywedge\left((\ell+1)\left(\Upsilon\left(s_{r_2}, \triangle_2\right)/(\ell+1) + \Upsilon\left(s_{r_1}, \triangle_1\right)/(\ell+1) - \Upsilon\left(s_{r_2}, \triangle_2\right)\Upsilon\left(s_{r_1}, \triangle_1\right)/(\ell+1)^2\right)\right), \\
\left\langle \sqrt[q]{1 - (1 - v_2^q)(1 - v_1^q)}, \sqrt[q]{(1 - v_2^q)(1 - v_1^q) - (1 - v_2^q - \mu_2^q)(1 - v_1^q - \mu_1^q)} \right\rangle
\end{pmatrix} = \partial_2 \oplus \partial_1.
$$

3. $\alpha(\partial_1 \oplus \partial_2) =$

$$
\alpha\begin{pmatrix}
\curlywedge\left((\ell+1)\begin{pmatrix}\Upsilon\left(s_{r_1}, \triangle_1\right)/(\ell+1) + \Upsilon\left(s_{r_2}, \triangle_2\right)/(\ell+1) - \\ \Upsilon\left(s_{r_1}, \triangle_1\right)\Upsilon\left(s_{r_2}, \triangle_2\right)/(\ell+1)^2\end{pmatrix}\right), \\
\left\langle \sqrt[q]{1 - \prod_{r=1}^{2}(1 - v_r^q)}, \sqrt[q]{\prod_{r=1}^{2}(1 - v_r^q) - \prod_{r=1}^{2}(1 - v_r^q - \mu_r^q)} \right\rangle
\end{pmatrix} =
$$

$$
\begin{pmatrix}
\curlywedge\left((\ell+1)\left(1 - \left(\left(1 - \left(s_{r_1}, \triangle_1\right)/(\ell+1)\right)^\alpha\left(1 - \left(s_{r_2}, \triangle_2\right)/(\ell+1)\right)^\alpha\right)\right)/(\ell+1)^2\right), \\
\left\langle \sqrt[q]{1 - \prod_{r=1}^{2}(1 - v_r^q)^\alpha}, \sqrt[q]{\prod_{r=1}^{2}(1 - v_r^q)^\alpha + \prod_{r=1}^{2}(1 - v_r^q) - \mu_r^q)^\alpha} \right\rangle
\end{pmatrix}
$$

$$
\alpha\partial_1 \oplus \alpha\partial_2 = \begin{pmatrix}
\curlywedge\left((\ell+1)\left(1 - \left(1 - \Upsilon\left(s_{r_1}, \triangle_1\right)/(\ell+1)\right)^\alpha\right)\right), \\
\left\langle \sqrt[q]{1 - (1 - v_1^q)^\alpha}, \sqrt[q]{(1 - v_1^q)^\alpha - (1 - v_1^q - \mu_1^q)^\alpha} \right\rangle
\end{pmatrix} \oplus
$$

$$
\begin{pmatrix}
\curlywedge\left((\ell+1)\left(1 - \left(1 - \Upsilon\left(s_{r_2}, \triangle_2\right)/(\ell+1)\right)^\alpha\right)\right), \\
\left\langle \sqrt[q]{1 - (1 - v_2^q)^\alpha}, \sqrt[q]{(1 - v_2^q)^\alpha - (1 - v_2^q - \mu_2^q)^\alpha} \right\rangle
\end{pmatrix} =
$$

$$
\begin{pmatrix}
\curlywedge\left((\ell+1)\left(1 - \left(\left(1 - \left(s_{r_1}, \triangle_1\right)/(\ell+1)\right)^\alpha\left(1 - \left(s_{r_2}, \triangle_2\right)/(\ell+1)\right)^\alpha\right)\right)/(\ell+1)^2\right), \\
\left\langle \sqrt[q]{1 - \prod_{r=1}^{2}(1 - v_r^q)^\alpha}, \sqrt[q]{\prod_{r=1}^{2}(1 - v_r^q)^\alpha + \prod_{r=1}^{2}(1 - v_r^q) - \mu_r^q)^\alpha} \right\rangle
\end{pmatrix}.
$$

$$5.\ (\alpha_1 + \alpha_2)\partial_1 = \left( \begin{array}{c} \curlywedge\left((\ell+1)\left(1 - \left(1 - \curlyvee\left(s_{r_1}, \triangle_1\right)/(\ell+1)\right)^{\alpha_1+\alpha_2}\right)\right), \\ \left\langle \sqrt[q]{1 - (1-v_1^q)^{\alpha_1+\alpha_2}}, \sqrt[q]{(1-v_1^q)^{\alpha_1+\alpha_2} - (1-v_1^q-\mu_1^q)^{\alpha_1+\alpha_2}} \right\rangle \end{array} \right)$$

$$\alpha_1\partial_1 \oplus \alpha_2\partial_1 = \left( \begin{array}{c} \curlywedge\left((\ell+1)\left(1 - \left(1 - \curlyvee\left(s_{r_1}, \triangle_1\right)/(\ell+1)\right)^{\alpha_1}\right)\right), \\ \left\langle \sqrt[q]{1 - (1-v_1^q)^{\alpha_1}}, \sqrt[q]{(1-v_1^q)_1^{\alpha} - (1-v_1^q-\mu_1^q)^{\alpha_1}} \right\rangle \end{array} \right) \oplus$$

$$\left( \begin{array}{c} \curlywedge\left((\ell+1)\left(1 - \left(1 - \curlyvee\left(s_{r_1}, \triangle_1\right)/(\ell+1)\right)^{\alpha_2}\right)\right), \\ \left\langle \sqrt[q]{1 - (1-v_1^q)^{\alpha_2}}, \sqrt[q]{(1-v_1^q)_2^{\alpha} - (1-v_1^q-\mu_1^q)^{\alpha_2}} \right\rangle \end{array} \right) =$$

$$\left( \begin{array}{c} \curlywedge\left((\ell+1)\left(1 - \left(1 - \curlyvee\left(s_{r_1}, \triangle_1\right)/(\ell+1)\right)^{\alpha_1+\alpha_2}\right)\right), \\ \left\langle \sqrt[q]{1 - (1-v_1^q)^{\alpha_1+\alpha_2}}, \sqrt[q]{(1-v_1^q)^{\alpha_1+\alpha_2} - (1-v_1^q-\mu_1^q)^{\alpha_1+\alpha_2}} \right\rangle \end{array} \right).$$

# 4 Novel q-RFL AOs

## 4.1 q-RFL weighted averaging operators

**Definition 11** *Let $\partial_i = \left(\left(s_{r_i}, \triangle_i\right), \langle v_i, \mu_i \rangle\right)(i = 1, 2, \ldots, m)$ be a range of q-RFLNs with their associated weight $w_i(i = 1, 2, \ldots, m)$ such that $w_i \in [0, 1]$ and $\sum_{i=1}^{m} w_i = 1$ then the operator q-RFLWA: $\partial^m \rightarrow \partial$ is given as*

$$q - RFLWA(\partial_1, \partial_2, \ldots, \partial_m) = \oplus_{i=1}^{m} w_i\partial_i. \tag{11}$$

**Theorem 2** *Let $\partial_i = \left(\left(s_{r_i}, \triangle_i\right), \langle v_i, \mu_i \rangle\right)(i = 1, 2, \ldots, m)$ be a range of q-RFLNs with their associated weight $w_i(i = 1, 2, \ldots, m)$ such that $w_i \in [0, 1]$ and $\sum_{i=1}^{m} w_i = 1$. Then the aggregated value of the $q - RFLWA$ operator is still a q-RFLN, and*

$$q - RFLWA(\partial_1, \partial_2, \ldots, \partial_m) = \oplus_{i=1}^{m} w_i\partial_i$$

$$= \left( \begin{array}{c} \curlywedge\left((\ell+1)\left(1 - \prod_{i=1}^{m}\left(1 - \curlyvee\left(s_{r_i}, \triangle_i\right)/(\ell+1)\right)^{w_i}\right)\right), \\ \left\langle \sqrt[q]{1 - \prod_{i=1}^{m}(1-v_i^q)^{w_i}}, \sqrt[q]{\prod_{i=1}^{m}(1-v_i^q)^{w_i} - \prod_{i=1}^{m}(1-v_i^q-\mu_i^q)^{w_i}} \right\rangle \end{array} \right). \tag{12}$$

**Proof.** We demonstrate this theorem by adopting the mathematical induction approach.

For $n = 2$, let $\partial_1 = \left(\left(s_{r_1}, \triangle_1\right), \langle v_1, \mu_1 \rangle\right)$ and $\partial_2 = \left(\left(s_{r_2}, \triangle_2\right), \langle v_2, \mu_2 \rangle\right)$ be two q-RFLNs. Then, using Definition 10, we obtain

$$w_1 \partial_1 = \left( \frac{\lambda\left((\ell+1)\left(1 - \left(1 - \Upsilon\left(s_{r_1}, \triangle_1\right)/(\ell+1)\right)^{w_1}\right)\right),}{\left\langle \sqrt[q]{1 - (1 - v_1^q)^{w_1}}, \sqrt[q]{(1 - v_1^q)^{w_1} - (1 - v_1^q - \mu_1^q)^{w_1}} \right\rangle} \right) \text{ and }$$

$$w_2 \partial_2 = \left( \frac{\lambda\left((\ell+1)\left(1 - \left(1 - \Upsilon\left(s_{r_2}, \triangle_2\right)/(\ell+1)\right)^{w_2}\right)\right),}{\left\langle \sqrt[q]{1 - (1 - v_2^q)^{w_2}}, \sqrt[q]{(1 - v_2^q)^{w_2} - (1 - v_2^q - \mu_2^q)^{w_2}} \right\rangle} \right)$$

$$w_1 \partial_1 \oplus w_2 \partial_2 = \left( \frac{\lambda\left((\ell+1)\left(1 - \left(1 - \Upsilon\left(s_{r_1}, \triangle_1\right)/(\ell+1)\right)^{w_1}\right)\right),}{\left\langle \sqrt[q]{1 - (1 - v_1^q)^{w_1}}, \sqrt[q]{(1 - v_1^q)^{w_1} - (1 - v_1^q - \mu_1^q)^{w_1}} \right\rangle} \right) \oplus$$

$$\left( \frac{\lambda\left((\ell+1)\left(1 - \left(1 - \Upsilon\left(s_{r_2}, \triangle_2\right)/(\ell+1)\right)^{w_2}\right)\right),}{\left\langle \sqrt[q]{1 - (1 - v_2^q)^{w_2}}, \sqrt[q]{(1 - v_2^q)^{w_2} - (1 - v_2^q - \mu_2^q)^{w_2}} \right\rangle} \right) =$$

$$\left( \frac{\lambda\left((\ell+1)\left(1 - \prod_{i=1}^{2}\left(1 - \Upsilon\left(s_{r_i}, \triangle_i\right)/(\ell+1)\right)^{w_i}\right)\right),}{\left\langle \sqrt[q]{1 - \prod_{i=1}^{2}(1 - v_i^q)^{w_i}}, \sqrt[q]{\prod_{i=1}^{2}(1 - v_i^q)^{w_i} - \prod_{i=1}^{2}(1 - v_i^q - \mu_i^q)^{w_i}} \right\rangle} \right).$$

Thereby, for $m = 2$, the outcome is accurate. Suppose the result is true for $m = k$; that is, we suppose

$$q - RFLWA(\partial_1, \partial_2, \ldots, \partial_k) =$$

$$\left( \lambda\left((\ell+1)\left(1 - \prod_{i=1}^{k}\left(1 - \Upsilon\left(s_{r_i}, \triangle_i\right)/(\ell+1)\right)^{w_i}\right)\right), \left\langle \sqrt[q]{1 - \prod_{i=1}^{k}(1 - v_i^q)^{w_i}}, \sqrt[q]{\prod_{i=1}^{k}(1 - v_i^q)^{w_i} - \prod_{i=1}^{k}(1 - v_i^q - \mu_i^q)^{w_i}} \right\rangle \right).$$

Now, for $m = k + 1$,

$$q - RFLWA\left(\partial_1, \partial_2, \ldots, \partial_{k+1}\right) = \oplus_{i=1}^{k+1} w_i \partial_i = \oplus_{i=1}^{k} w_i \partial_i \oplus w_{k+1} \partial_{k+1}$$

$$= \left( \lambda\left((\ell+1)\left(1 - \prod_{i=1}^{k}\left(1 - \Upsilon\left(s_{r_i}, \triangle_i\right)/(\ell+1)\right)^{w_i}\right)\right), \left\langle \sqrt[q]{1 - \prod_{i=1}^{k}(1 - v_i^q)^{w_i}}, \sqrt[q]{\prod_{i=1}^{k}(1 - v_i^q)^{w_i} - \prod_{i=1}^{k}(1 - v_i^q - \mu_i^q)^{w_i}} \right\rangle \right) \oplus$$

$$\left( \frac{\lambda\left((\ell+1)\left(1 - \left(1 - \Upsilon\left(s_{r_{k+1}}, \triangle_{k+1}\right)/(\ell+1)\right)^{w_{k+1}}\right)\right),}{\left\langle \sqrt[q]{1 - \left(1 - v_{k+1}^q\right)^{w_{k+1}}}, \sqrt[q]{\left(1 - v_{k+1}^q\right)^{w_{k+1}} - \left(1 - v_{k+1}^q - \mu_1^q\right)^{w_{k+1}}} \right\rangle} \right)$$

$$= \left( \lambda\left((\ell+1)\left(1 - \prod_{i=1}^{k+1}\left(1 - \Upsilon\left(s_{r_i}, \triangle_i\right)/(\ell+1)\right)^{w_i}\right)\right), \left\langle \sqrt[q]{1 - \prod_{i=1}^{k+1}(1 - v_i^q)^{w_i}}, \sqrt[q]{\prod_{i=1}^{k+1}(1 - v_i^q)^{w_i} - \prod_{i=1}^{k+1}(1 - v_i^q - \mu_i^q)^{w_i}} \right\rangle \right)$$

Thus Eq (12) is true for $n = k + 1$. Hence, it is true for all $m$.

**Theorem 3** (Permutation) Let $\partial_i = \left(\left(s_{r_i}, \triangle_i\right), \langle v_i, \mu_i \rangle\right)$ and $\partial_{(i)} = \left(\left(s_{r_{(i)}}, \triangle_{(i)}\right), \left\langle v_{(i)}, \mu_{(i)} \right\rangle\right) (i = 1, 2, \ldots, m)$ be two collections of q-RFLNs, where $(\cdot)$ indicates

*any permutation of i. Then*

$$q - RFLWA(\partial_1, \partial_2, \ldots, \partial_m) = q - RFLWA\left(\partial_{(1)}, \partial_{(2)}, \ldots, \partial_{(m)}\right). \tag{13}$$

**Proof.**

$$q - RFLWA(\partial_1, \partial_2, \ldots, \partial_m) =$$

$$= \left( \curlywedge\left( (\ell+1)\left( 1 - \prod_{i=1}^{m}\left( 1 - \Upsilon\left(s_{r_i}, \triangle_i\right)/(\ell+1)\right)^{w_i} \right) \right), \left\langle \sqrt[q]{1 - \prod_{i=1}^{m}(1 - v_i^q)^{w_i}}, \sqrt[q]{\prod_{i=1}^{m}(1 - v_i^q)^{w_i} - \prod_{i=1}^{m}(1 - v_i^q - \mu_i^q)^{w_i}} \right\rangle \right)$$

$$= \left( \curlywedge\left( (\ell+1)\left( 1 - \prod_{(i)=1}^{m}\left( 1 - \Upsilon\left(s_{r_{(i)}}, \triangle_{(i)}\right)/(\ell+1)\right)^{w_{(i)}} \right) \right), \left\langle \sqrt[q]{1 - \prod_{(i)=1}^{m}\left( 1 - v_{(i)}^q \right)^{w_{(i)}}}, \sqrt[q]{\prod_{(i)=1}^{m}\left( 1 - v_{(i)}^q \right)^{w_{(i)}} - \prod_{(i)=1}^{m}\left( 1 - v_{(i)}^q - \mu_{(i)}^q \right)^{w_{(i)}}} \right\rangle \right)$$

$$= q - RFLWA\left( \partial_{(1)}, \partial_{(2)}, \ldots, \partial_{(m)} \right)$$

This ends the proof of Theorem 3.

**Theorem 4** *(Idempotency) Let* $\partial_i = \left( \left( s_{r_i}, \triangle_i \right), \langle v_i, \mu_i \rangle \right)(i = 1, 2, \ldots, m)$ *be a range of q-RFLNs with their associated weight* $w_i(i = 1, 2, \ldots, m)$ *such that* $w_i \in [0, 1]$ *and* $\sum_{i=1}^{m} w_i = 1$ *if* $\partial_i = \partial \, \forall i$ *then*

$$q - RFLWA(\partial_1, \partial_2, \ldots, \partial_m) = \partial. \tag{14}$$

**Proof.** Since $\partial_i = \partial \, \forall i$ and $\sum_{i=1}^{m} w_i = 1$, so according to Theorem 2, we have

$$q - RFLWA(\partial_1, \partial_2, \ldots, \partial_m) =$$

$$\left( \curlywedge\left( (\ell+1)\left( 1 - \prod_{i=1}^{m}(1 - \Upsilon(s_r, \triangle)/(\ell+1))^{w_i} \right) \right), \left\langle \sqrt[q]{1 - \prod_{i=1}^{m}(1 - v^q)^{w_i}}, \sqrt[q]{\prod_{i=1}^{m}(1 - v^q)^{w_i} - \prod_{i=1}^{m}(1 - v^q - \mu^q)^{w_i}} \right\rangle \right) =$$

$$\left( \curlywedge\left( (\ell+1)\left( 1 - (1 - \Upsilon(s_r, \triangle)/(\ell+1))^{\sum_{i=1}^{m} w_i} \right) \right), \left\langle \sqrt[q]{1 - (1 - v^q)^{\sum_{i=1}^{m} w_i}}, \sqrt[q]{(1 - v^q)^{\sum_{i=1}^{m} w_i} - (1 - v^q - \mu^q)^{\sum_{i=1}^{m} w_i}} \right\rangle \right) =$$

$$= ((s_r, \triangle), \langle v, \mu \rangle) = \partial.$$

**Theorem 5** *(Boundedness) Let* $\partial_i = \left( \left( s_{r_i}, \triangle_i \right), \langle v_i, \mu_i \rangle \right)(i = 1, 2, \ldots, m)$ *be a range of q-RFLNs. If* $\partial^- = \min(\partial_1, \partial_2, \ldots, \partial_m)$ *and* $\partial^+ = \max(\partial_1, \partial_2, \ldots, \partial_m)$, *then*

$$\partial^- \leq q - RFLWA(\partial_1, \partial_2, \ldots, \partial_m) \leq \partial^+. \tag{15}$$

**Proof.** Since $\partial^- = \min(\partial_1, \partial_2, \ldots, \partial_m) = ((s_r, \triangle)^-, \langle v^-, \mu^- \rangle)$ and $\partial^+ = \max(\partial_1, \partial_2, \ldots, \partial_m) = ((s_r, \triangle)^+, \langle v^+, \mu^+ \rangle)$, where $(s_r, \triangle)^- = \min\left\{ \left( s_{r_1}, \triangle_1 \right), \left( s_{r_2}, \triangle_2 \right), \ldots, \left( s_{r_m}, \triangle_m \right) \right\}$, $v^- = \min\{v_1, v_2, \ldots, v_m\}, \mu^- = \max\{\mu_1, \mu_2, \ldots, \mu_m\}, (s_r, \triangle)^+ = \max\left\{ \left( s_{r_1}, \triangle_1 \right), \left( s_{r_2}, \triangle_2 \right), \ldots, \left( s_{r_m}, \triangle_m \right) \right\}$, $v^+ = \max\{v_1, v_2, \ldots, v_m\}, \mu^+ = \min\{\mu_1, \mu_2, \ldots, \mu_m\}$.

Now,

$$\lambda\left((\ell+1)\left(1-\prod_{i=1}^{m}(1-\Upsilon(s_r,\triangle)^-/(\ell+1))^{w_i}\right)\right) \leq$$

$$\lambda\left((\ell+1)\left(1-\prod_{i=1}^{m}(1-\Upsilon(s_r,\triangle)/(\ell+1))^{w_i}\right)\right) \leq$$

$$\lambda\left((\ell+1)\left(1-\prod_{i=1}^{m}(1-\Upsilon(s_r,\triangle)^+/(\ell+1))^{w_i}\right)\right),$$

$$\sqrt[q]{1-\prod_{i=1}^{m}(1-v^{-q})^{w_i}} \leq \sqrt[q]{1-\prod_{i=1}^{m}(1-v^q)^{w_i}} \leq \sqrt[q]{1-\prod_{i=1}^{m}(1-v^{+q})^{w_i}},$$

and

$$\sqrt[q]{\prod_{i=1}^{m}(1-v^{+q})^{w_i}-\prod_{i=1}^{m}(1-v^{+q}-\mu^{-q})^{w_i}} \leq \sqrt[q]{\prod_{i=1}^{m}(1-v^q)^{w_i}-\prod_{i=1}^{m}(1-v^q-\mu^q)^{w_i}}$$

$$\leq \sqrt[q]{\prod_{i=1}^{m}(1-v^{-q})^{w_i}-\prod_{i=1}^{m}(1-v^{-q}-\mu^{+q})^{w_i}}.$$

Based on the above inequalities, we can write

$$\partial^- \leq q - RFLWA(\partial_1, \partial_2, \ldots, \partial_m) \leq \partial^+.$$

**Theorem 6** *(Monotonicity) Let* $\partial_i = \left(\left(s_{r_i}, \triangle_i\right), \langle v_i, \mu_i\rangle\right)$ *and* $\bar{\partial}_i =$
$\left(\overline{\left(s_{r_i}, \triangle_i\right)}, \langle \bar{v}_i, \bar{\mu}_i\rangle\right) (i = 1, 2, \ldots, m)$ *be two collections of q-RFLNs. If* $\left(s_{r_i}, \triangle_i\right) \geq$
$\overline{\left(s_{r_i}, \triangle_i\right)}, \langle \bar{v}_i, \bar{\mu}_i\rangle\right), v_i \geq \bar{v}_i$ *and* $\mu_i \leq \bar{\mu}_i \, \forall i,$ *then*

$$q - RFLWA(\partial_1, \partial_2, \ldots, \partial_m) \geq q - RFLWA(\bar{\partial}_1, \bar{\partial}_2, \ldots, \bar{\partial}_m). \tag{16}$$

**Proof.** Since $\left(s_{r_i}, \triangle_i\right) \geq \overline{\left(s_{r_i}, \triangle_i\right)}, v_i \geq \bar{v}_i$ and $\mu_i \leq \bar{\mu}_i \, \forall i.$ Based on these, we have

$$\lambda\left((\ell+1)\left(1-\prod_{i=1}^{m}(1-\Upsilon(s_r,\triangle)/(\ell+1))^{w_i}\right)\right) \geq$$

$$\lambda\left((\ell+1)\left(1-\prod_{i=1}^{m}\left(1-\Upsilon\overline{(s_r,\triangle)}/(\ell+1)\right)^{w_i}\right)\right),$$

$$\sqrt[q]{1-\prod_{i=1}^{m}(1-v_i^q)^{w_i}} \geq \sqrt[q]{1-\prod_{i=1}^{m}(1-\bar{v}_i^q)^{w_i}},$$

and

$$\sqrt[q]{\prod_{i=1}^{m}(1-v_i^q)^{w_i}-\prod_{i=1}^{m}(1-v_i^q-\mu_i^q)^{w_i}} \leq \sqrt[q]{\prod_{i=1}^{m}(1-\bar{v}_i^q)^{w_i}-\prod_{i=1}^{m}(1-\bar{v}_i^q-\bar{\mu}_i^q)^{w_i}}.$$

The above inequalities imply that

$$\left(\begin{array}{c} \lambda\left((\ell+1)\left(1-\prod_{i=1}^{m}\left(1-\Upsilon\left(s_{r_i},\triangle_i\right)/(\ell+1)\right)^{w_i}\right)\right), \\ \left\langle\sqrt[q]{1-\prod_{i=1}^{m}(1-v_i^q)^{w_i}},\sqrt[q]{\prod_{i=1}^{m}(1-v_i^q)^{w_i}-\prod_{i=1}^{m}(1-v_i^q-\mu_i^q)^{w_i}}\right\rangle \end{array}\right) \geq$$

$$
\left(
\begin{array}{c}
\curlywedge\left((\ell+1)\left(1-\prod_{i=1}^{m}\left(1-\Upsilon\overline{\left(s_{r_i},\triangle_i\right)}/(\ell+1)\right)^{w_i}\right)\right), \\
\left\langle\sqrt[q]{1-\prod_{i=1}^{m}(1-\bar{v}_i^q)^{w_i}}, \sqrt[q]{\prod_{i=1}^{m}(1-\bar{v}_i^q)^{w_i}-\prod_{i=1}^{m}(1-\bar{v}_i^q-\bar{\mu}_i^q)^{w_i}}\right\rangle
\end{array}
\right).
$$

Hence,

$q-RFLWA(\partial_1,\partial_2,\ldots,\partial_m) \geq q-RFLWA(\bar{\partial}_1,\bar{\partial}_2,\ldots,\bar{\partial}_m)$.

Following this, we discuss the q-rung orthopair fuzzy 2-tuple linguistic ordered weighted averaging (q-RFLOWA) operator and its properties.

**Definition 12** *Let $\partial_i = \left(\left(s_{r_i},\triangle_i\right),\langle v_i,\mu_i\rangle\right)(i=1,2,\ldots,m)$ be a range of q-RFLNs with the position weights $\varpi_i(i=1,2,\ldots,m)$ such that $\varpi_i \in [0,1]$ and $\sum_{i=1}^{m}\varpi_i = 1$ then the operator q-RFLOWA: $\partial^m \to \partial$ is described as*

$$q-RFLOWA(\partial_1,\partial_2,\ldots,\partial_m) = \oplus_{i=1}^{m}\varpi_i\partial_{\hat{i}}, \tag{17}$$

*where $\left(\hat{1},\hat{2},\ldots,\hat{m}\right)$ is the permutation of $(i=1,2,\ldots,m)$, in such a way as $\partial_{\widehat{i-1}} \geq \partial_{\hat{i}}$ for i = 2, 3..., m.*

**Theorem 7** *Let $\partial_i = \left(\left(s_{r_i},\triangle_i\right),\langle v_i,\mu_i\rangle\right)(i=1,2,\ldots,m)$ be a family of q-RFLNs with the position weights $\varpi_i(i=1,2,\ldots,m)$ such that $\varpi_i \in [0,1]$ and $\sum_{i=1}^{m}\varpi_i = 1$. Then the aggregated value of the $q-RFLOWA$ operator is still a q-RFLN, and*

$$
\begin{aligned}
q-RFLOWA&(\partial_1,\partial_2,\ldots,\partial_m) = \oplus_{i=1}^{m}\varpi_i\partial_{\hat{i}} \\
&= \left(
\begin{array}{c}
\curlywedge\left((\ell+1)\left(1-\prod_{i=1}^{m}\left(1-\Upsilon\left(s_{r_i},\triangle_{\hat{i}}\right)/(\ell+1)\right)^{\varpi_i}\right)\right), \\
\left\langle\sqrt[q]{1-\prod_{i=1}^{m}\left(1-v_{\hat{i}}^q\right)^{\varpi_i}}, \sqrt[q]{\prod_{i=1}^{m}\left(1-v_{\hat{i}}^q\right)^{\varpi_i}-\prod_{i=1}^{m}\left(1-v_{\hat{i}}^q-\mu_{\hat{i}}^q\right)^{\varpi_i}}\right\rangle
\end{array}
\right).
\end{aligned} \tag{18}
$$

**Proof.** Straightforward from Theorem 2.

Utilizing the $q-RFLOWA$ operator, the following features can be easily verified.

**Theorem 8** *(Permutation) Let $\partial_i = \left(\left(s_{r_i},\triangle_i\right),\langle v_i,\mu_i\rangle\right)$ and $\partial_{(i)} = \left(\left(s_{r_{(i)}},\triangle_{(i)}\right),\left\langle v_{(i)},\mu_{(i)}\right\rangle\right)(i=1,2,\ldots,m)$ be two collections of q-RFLNs, where $(\cdot)$ indicates any permutation of i. Then*

$$q-RFLOWA(\partial_1,\partial_2,\ldots,\partial_m) = q-RFLOWA\left(\partial_{(1)},\partial_{(2)},\ldots,\partial_{(m)}\right). \tag{19}$$

**Theorem 9** *(Idempotency) Let $\partial_i = \left(\left(s_{r_i},\triangle_i\right),\langle v_i,\mu_i\rangle\right)(i=1,2,\ldots,m)$ be a range of q-RFLNs with their position weight $\varpi_i(i=1,2,\ldots,m)$ such that $\varpi_i \in [0,1]$ and $\sum_{i=1}^{m}\varpi_i = 1$ if $\partial_i = \partial \; \forall i$ then*

$$q-RFLOWA(\partial_1,\partial_2,\ldots,\partial_m) = \partial. \tag{20}$$

**Theorem 10** *(Boundedness) Let* $\partial_i = \left( \left( s_{r_i}, \triangle_i \right), \langle v_i, \mu_i \rangle \right) (i = 1, 2, \ldots, m)$ *be a range of q-RFLNs. If* $\partial^- = \min(\partial_1, \partial_2, \ldots, \partial_m)$ *and* $\partial^+ = \max(\partial_1, \partial_2, \ldots, \partial_m)$, *then*

$$\partial^- \leq q - RFLOWA(\partial_1, \partial_2, \ldots, \partial_m) \leq \partial^+. \tag{21}$$

**Theorem 11** *(Monotonicity) Let* $\partial_i = \left( \left( s_{r_i}, \triangle_i \right), \langle v_i, \mu_i \rangle \right)$ *and* $\bar{\partial}_i = \left( \overline{\left( s_{r_i}, \triangle_i \right)}, \langle \bar{v}_i, \bar{\mu}_i \rangle \right) (i = 1, 2, \ldots, m)$ *be two collections of q-RFLNs. If* $\left( s_{r_i}, \triangle_i \right) \geq \overline{\left( s_{r_i}, \triangle_i \right)}$, $v_i \geq \bar{v}_i$ *and* $\mu_i \leq \bar{\mu}_i$ $\forall i$, *then*

$$q - RFLOWA(\partial_1, \partial_2, \ldots, \partial_m) \geq q - RFLOWA(\bar{\partial}_1, \bar{\partial}_2, \ldots, \bar{\partial}_m). \tag{22}$$

In Definition 11, we see that q-RFLWA operator weights are the most basic form of q-RFLN, whereas q-RFLOWA operator weights are the special type of the organized locations of the q-RFLNs. As a result, the weights specified in the operators q-RFLWA and q-RFLOWA present various scenarios that are antagonistic to one another. In any event, in terms of the general approach, these perspectives are considered equivalent. To avoid such inconvenience, we provide the q-RFL hybrid averaging (q-RFLHA) operator in the following.

**Definition 13** *Let* $\partial_i = \left( \left( s_{r_i}, \triangle_i \right), \langle v_i, \mu_i \rangle \right) (i = 1, 2, \ldots, m)$ *be a range of q-RFLNs with the position weights* $\varpi_i (i = 1, 2, \ldots, m)$ *such that* $\varpi_i \in [0, 1]$ *and* $\sum_{i=1}^{m} \varpi_i = 1$ *then the operator q-RFLHA:* $\partial^m \to \partial$ *is described as*

$$q - RFLHA(\partial_1, \partial_2, \ldots, \partial_m) = \oplus_{i=1}^{m} \varpi_i \dot{\partial}_i, \tag{23}$$

*where* $\dot{\partial}_i = mw_i \partial_i, i = 1, 2, \ldots, m, \left( \dot{1}, \dot{2}, \ldots, \dot{m} \right)$ *is the permutation of* $(i = 1, 2, \ldots, m)$, *in such a way as* $\partial_{\widehat{i-1}} \geq \partial_{\dot{i}}$ *for* $i = 2, 3 \ldots, m$; $w = (w_1, w_2, \ldots, w_m)^T$ *is the weight vector of* $\partial_i$ *such that* $w_i \in [0, 1]$ *and* $\sum_{i=1}^{m} w_i = 1$, *and m is the balancing coefficient.*

**Theorem 12** *Let* $\partial_i = \left( \left( s_{r_i}, \triangle_i \right), \langle v_i, \mu_i \rangle \right) (i = 1, 2, \ldots, m)$ *be a range of q-RFLNs with the position weights* $\varpi_i (i = 1, 2, \ldots, m)$ *such that* $\varpi_i \in [0, 1]$ *and* $\sum_{i=1}^{m} \varpi_i = 1$. *Then the aggregated value of the* $q - RFLHA$ *operator is still a q-RFLN, and*

$$q - RFLHA(\partial_1, \partial_2, \ldots, \partial_m) = \oplus_{i=1}^{m} \varpi_i \dot{\partial}_i$$

$$= \left( \begin{array}{c} \curlywedge \left( (\ell + 1) \left( 1 - \prod_{i=1}^{m} \left( 1 - \curlyvee \left( s_{r_{\dot{i}}}, \triangle_{\dot{i}} \right) / (\ell + 1) \right)^{\varpi_i} \right) \right), \\ \left\langle \sqrt[q]{1 - \prod_{i=1}^{m} \left( 1 - v_{\dot{i}}^q \right)^{\varpi_i}}, \sqrt[q]{\prod_{i=1}^{m} \left( 1 - v_{\dot{i}}^q \right)^{\varpi_i} - \prod_{i=1}^{m} \left( 1 - v_{\dot{i}}^q - \mu_{\dot{i}}^q \right)^{\varpi_i}} \right\rangle \end{array} \right). \tag{24}$$

**Proof.** Straightforward from Theorem 2.

The q-RFLHA operator, like the q-RFLWA and q-RFLOWA operators, has permutation, idempotency, boundedness, and monotonicity features. Aside from the aforementioned properties, the q-RFLHA operator has the following special instances.

**Corollary 1** *q-RFLWA operator is a special case of q-RFLHA operator.*

**Proof.** Let $\varpi = \left(\frac{1}{m}, \frac{1}{m}, \dots, \frac{1}{m}\right)^T$, the $q - RFLHA(\partial_1, \partial_2, \dots, \partial_m) = \varpi_1 \partial_{\dot{1}} \oplus \varpi_2 \partial_{\dot{2}} \oplus \dots \oplus \varpi_m \partial_{\dot{m}} = \frac{1}{m}\left(\partial_{\dot{1}} \oplus \partial_{\dot{2}} \oplus \dots \oplus \partial_{\dot{m}}\right) = w_1 \partial_1 \oplus w_2 \partial_2 \oplus \dots \oplus w_m \partial_m = q - RFLWA(\partial_1, \partial_2, \dots, \partial_m)$.

**Corollary 2** *q-RFLOWA operator is a special case of q-RFLHA operator.*

**Proof.** Let $w = \left(\frac{1}{m}, \frac{1}{m}, \dots, \frac{1}{m}\right)^T$, the $q - RFLHA(\partial_1, \partial_2, \dots, \partial_m) = \varpi_1 \partial_{\dot{1}} \oplus \varpi_2 \partial_{\dot{2}} \oplus \dots \oplus \varpi_m \partial_{\dot{m}} = \varpi_1 \partial_{\hat{1}} \oplus \varpi_2 \partial_{\hat{2}} \oplus \dots \oplus \varpi_m \partial_{\hat{m}} = q - RFLOWA(\partial_1, \partial_2, \dots, \partial_m)$.

## 4.2 q-RFL weighted geometric operators

**Definition 14** *Let $\partial_i = \left(\left(s_{r_i}, \triangle_i\right), \langle v_i, \mu_i \rangle\right)(i = 1, 2, \dots, m)$ be a range of q-RFLNs with their associated weight $w_i(i = 1, 2, \dots, m)$ such that $w_i \in [0, 1]$ and $\sum_{i=1}^{m} w_i = 1$ then the operator q-RFLWG: $\partial^m \to \partial$ is described as*

$$q - RFLWG(\partial_1, \partial_2, \dots, \partial_m) = \otimes_{i=1}^{m} \partial_i^{w_i}. \tag{25}$$

**Theorem 13** *Let $\partial_i = \left(\left(s_{r_i}, \triangle_i\right), \langle v_i, \mu_i \rangle\right)(i = 1, 2, \dots, m)$ be a range of q-RFLNs with their associated weight $w_i(i = 1, 2, \dots, m)$ such that $w_i \in [0, 1]$ and $\sum_{i=1}^{m} w_i = 1$. Then the aggregated value of the $q - RFLWG$ operator is still a q-RFLN, and*

$$q - RFLWG(\partial_1, \partial_2, \dots, \partial_m) = \otimes_{i=1}^{m} \partial_i^{w_i} =$$

$$\left( \curlywedge\left((\ell+1)\prod_{i=1}^{m}\left(\curlyvee\left(s_{r_i}, \triangle_i\right)/(\ell+1)\right)^{w_i}\right), \left\langle \frac{\sqrt[q]{\prod_{i=1}^{m}(1-\mu_i^q)^{w_i} - \prod_{i=1}^{m}(1-v_i^q-\mu_i^q)^{w_i}}}{\sqrt[q]{1-\prod_{i=1}^{m}(1-\mu_i^q)^{w_i}}} \right\rangle \right). \tag{26}$$

**Proof.** We demonstrate this theorem by adopting the mathematical induction approach.

For $n = 2$, let $\partial_1 = \left(\left(s_{r_1}, \triangle_1\right), \langle v_1, \mu_1 \rangle\right)$ and $\partial_2 = \left(\left(s_{r_2}, \triangle_2\right), \langle v_2, \mu_2 \rangle\right)$ be two q-RFLNs. Then, using Definition 10, we obtain

$$\partial_1^{w_1} = \left( \begin{array}{c} \curlywedge\left((\ell+1)\left(\left(\curlyvee\left(s_{r_1}, \triangle_1\right)/(\ell+1)\right)^{w_1}\right)\right), \\ \left\langle \sqrt[q]{(1-\mu_1^q)^{w_1} - (1-v_1^q-\mu_1^q)^{\alpha}}, \sqrt[q]{1-(1-\mu_1^q)^{w_1}} \right\rangle \end{array} \right) \text{ and}$$

$$\partial_2^{w_2} = \left( \begin{array}{c} \curlywedge\left((\ell+1)\left(\left(\curlyvee\left(s_{r_1}, \triangle_1\right)/(\ell+1)\right)^{w_2}\right)\right), \\ \left\langle \sqrt[q]{(1-\mu_1^q)^{w_2} - (1-v_1^q-\mu_1^q)^{\alpha}}, \sqrt[q]{1-(1-\mu_1^q)^{w_2}} \right\rangle \end{array} \right)$$

$$\partial_1^{w_1} \otimes \partial_2^{w_1} = \left( \frac{\lambda\left( (\ell+1)\left( \left( \curlyvee\left(s_{r_1}, \triangle_1\right)/(\ell+1)\right)^{w_1}\right)\right),}{\left\langle \sqrt[q]{\left(1-\mu_1^q\right)^{w_1} - \left(1-v_1^q-\mu_1^q\right)^{\alpha}}, \sqrt[q]{1-\left(1-\mu_1^q\right)^{w_1}}\right\rangle} \right) \otimes$$

$$\left( \frac{\lambda\left( (\ell+1)\left( \left( \curlyvee\left(s_{r_1}, \triangle_1\right)/(\ell+1)\right)^{w_2}\right)\right),}{\left\langle \sqrt[q]{\left(1-\mu_1^q\right)^{w_2} - \left(1-v_1^q-\mu_1^q\right)^{\alpha}}, \sqrt[q]{1-\left(1-\mu_1^q\right)^{w_2}}\right\rangle} \right) =$$

$$\left( \frac{\lambda\left( (\ell+1)\prod_{i=1}^{2}\left( \curlyvee\left(s_{r_i}, \triangle_i\right)/(\ell+1)\right)^{w_i}\right),}{\left\langle \sqrt[q]{\prod_{i=1}^{2}(1-\mu_i^q)^{w_i} - \prod_{i=1}^{2}(1-v_i^q-\mu_i^q)^{w_i}}, \sqrt[q]{1-\prod_{i=1}^{2}(1-\mu_i^q)^{w_i}}\right\rangle} \right).$$

Thereby, for $m = 2$, the outcome is accurate. Suppose the result is true for $m = k$; that is, we suppose

$$q - RFLWG(\partial_1, \partial_2, \ldots, \partial_k) =$$

$$\left( \lambda\left( (\ell+1)\prod_{i=1}^{k}\left( \curlyvee\left(s_{r_i}, \triangle_i\right)/(\ell+1)\right)^{w_i}\right), \left\langle \sqrt[q]{\prod_{i=1}^{k}(1-\mu_i^q)^{w_i} - \prod_{i=1}^{k}(1-v_i^q-\mu_i^q)^{w_i}}, \sqrt[q]{1-\prod_{i=1}^{k}(1-\mu_i^q)^{w_i}}\right\rangle\right).$$

Now, for $m = k + 1$,

$$q - RFLWA\left(\partial_1, \partial_2, \ldots, \partial_{k+1}\right) = \otimes_{i=1}^{k+1}\partial_i^{w_i} = \otimes_{i=1}^{k}\partial_i^{w_i} \otimes \partial_{k+1}^{w_{k+1}}$$

$$= \left( \lambda\left( (\ell+1)\prod_{i=1}^{k}\left( \curlyvee\left(s_{r_i}, \triangle_i\right)/(\ell+1)\right)^{w_i}\right), \left\langle \sqrt[q]{\prod_{i=1}^{k}(1-\mu_i^q)^{w_i} - \prod_{i=1}^{k}(1-v_i^q-\mu_i^q)^{w_i}}, \sqrt[q]{1-\prod_{i=1}^{k}(1-\mu_i^q)^{w_i}}\right\rangle\right) \otimes$$

$$\left( \frac{\lambda\left( (\ell+1)\left( \left( \curlyvee\left(s_{r_1}, \triangle_1\right)/(\ell+1)\right)^{w_k}\right)\right),}{\left\langle \sqrt[q]{\left(1-\mu_1^q\right)^{\alpha} - \left(1-v_1^q-\mu_1^q\right)^{w_k}}, \sqrt[q]{1-\left(1-\mu_1^q\right)^{w_k}}\right\rangle} \right)$$

$$= \left( \lambda\left( (\ell+1)\prod_{i=1}^{k+1}\left( \curlyvee\left(s_{r_i}, \triangle_i\right)/(\ell+1)\right)^{w_i}\right), \left\langle \sqrt[q]{\prod_{i=1}^{k+1}(1-\mu_i^q)^{w_i} - \prod_{i=1}^{k+1}(1-v_i^q-\mu_i^q)^{w_i}}, \sqrt[q]{1-\prod_{i=1}^{k+1}(1-\mu_i^q)^{w_i}}\right\rangle\right).$$

Thus Eq (26) is true for $n = k + 1$. Hence, it is true for all $m$.

**Theorem 14** *(Permutation) Let* $\partial_i = \left( \left(s_{r_i}, \triangle_i\right), \langle v_i, \mu_i\rangle\right)$ *and* $\partial_{(i)} = \left( \left(s_{r_{(i)}}, \triangle_{(i)}\right), \left\langle v_{(i)}, \mu_{(i)}\right\rangle\right) (i = 1, 2, \ldots, m)$ *be two collections of q-RFLNs, where* $(\cdot)$ *indicates any permutation of i. Then*

$$q - RFLWG(\partial_1, \partial_2, \ldots, \partial_m) = q - RFLWG\left(\partial_{(1)}, \partial_{(2)}, \ldots, \partial_{(m)}\right). \tag{27}$$

**Proof.**

$$q - RFLWG(\partial_1, \partial_2, \ldots, \partial_m) =$$

$$= \left( \lambda\left( (\ell+1)\prod_{i=1}^{m}\left(\curlyvee\left(s_{r_i}, \triangle_i\right)/(\ell+1)\right)^{w_i}\right), \left\langle \sqrt[q]{\prod_{i=1}^{m}(1-\mu_i^q)^{w_i} - \prod_{i=1}^{m}(1-v_i^q-\mu_i^q)^{w_i}}, \sqrt[q]{1-\prod_{i=1}^{m}(1-\mu_i^q)^{w_i}}\right\rangle\right)$$

$$= \left( \lambda\left( (\ell+1)\prod_{(i)=1}^{m}\left(\curlyvee\left(s_{r_{(i)}}, \triangle_{(i)}\right)/(\ell+1)\right)^{w_{(i)}}\right), \left\langle \sqrt[q]{\prod_{(i)=1}^{m}\left(1-\mu_{(i)}^q\right)^{w_{(i)}} - \prod_{(i)=1}^{m}\left(1-v_{(i)}^q-\mu_{(i)}^q\right)^{w_{(i)}}}, \sqrt[q]{1-\prod_{(i)=1}^{m}\left(1-\mu_{(i)}^q\right)^{w_{(i)}}}\right\rangle\right)$$

$$= q - RFLWG\left(\partial_{(1)}, \partial_{(2)}, \ldots, \partial_{(m)}\right).$$

This ends the proof of Theorem 14.

**Theorem 15** (*Idempotency*) *Let* $\partial_i = \left(\left(s_{r_i}, \triangle_i\right), \langle v_i, \mu_i\rangle\right)(i = 1, 2, \ldots, m)$ *be a range of q-RFLNs with their associated weight* $w_i(i = 1, 2, \ldots, m)$ *such that* $w_i \in [0, 1]$ *and* $\sum_{i=1}^{m} w_i = 1$ *if* $\partial_i = \partial \; \forall i$ *then*

$$q - RFLWG(\partial_1, \partial_2, \ldots, \partial_m) = \partial. \tag{28}$$

**Proof.** Since $\partial_i = \partial \; \forall i$ and $\sum_{i=1}^{m} w_i = 1$ so according to Theorem 13, we have

$$q - RFLWG(\partial_1, \partial_2, \ldots, \partial_m) =$$

$$\left( \lambda\left( (\ell+1)\prod_{i=1}^{m}(\curlyvee(s_r, \triangle)/(\ell+1))^{w_i}\right), \left\langle \sqrt[q]{\prod_{i=1}^{m}(1-\mu^q)^{w_i} - \prod_{i=1}^{m}(1-v^q-\mu^q)^{w_i}}, \sqrt[q]{1-\prod_{i=1}^{m}(1-\mu^q)^{w_i}}\right\rangle\right) =$$

$$\left( \lambda\left( (\ell+1)(\curlyvee(s_r, \triangle)/(\ell+1))^{\sum_{i=1}^{m} w_i}\right), \left\langle \sqrt[q]{(1-\mu^q)^{\sum_{i=1}^{m} w_i} - (1-v^q-\mu^q)^{\sum_{i=1}^{m} w_i}}, \sqrt[q]{1-(1-\mu^q)^{\sum_{i=1}^{m} w_i}}\right\rangle\right) =$$

$$= ((s_r, \triangle), \langle v, \mu\rangle) = \partial.$$

**Theorem 16** (*Boundedness*) *Let* $\partial_i = \left(\left(s_{r_i}, \triangle_i\right), \langle v_i, \mu_i\rangle\right)(i = 1, 2, \ldots, m)$ *be a range of q-RFLNs. If* $\partial^- = \min(\partial_1, \partial_2, \ldots, \partial_m)$ *and* $\partial^+ = \max(\partial_1, \partial_2, \ldots, \partial_m)$, *then*

$$\partial^- \leq q - RFLWG(\partial_1, \partial_2, \ldots, \partial_m) \leq \partial^+. \tag{29}$$

**Proof.** Since $\partial^- = \min(\partial_1, \partial_2, \ldots, \partial_m) = ((s_r, \triangle)^-, \langle v^-, \mu^-\rangle)$ and $\partial^+ = \max(\partial_1, \partial_2, \ldots, \partial_m) = ((s_r, \triangle)^+, \langle v^+, \mu^+\rangle)$, where $(s_r, \triangle)^- = \min\left\{\left(s_{r_1}, \triangle_1\right), \left(s_{r_2}, \triangle_2\right), \ldots, \left(s_{r_m}, \triangle_m\right)\right\}$, $v^- = \min\{v_1, v_2, \ldots, v_m\}$, $\mu^- = \max\{\mu_1, \mu_2, \ldots, \mu_m\}$, $(s_r, \triangle)^+ = \max\left\{\left(s_{r_1}, \triangle_1\right), \left(s_{r_2}, \triangle_2\right), \ldots, \left(s_{r_m}, \triangle_m\right)\right\}$, $v^+ = \max\{v_1, v_2, \ldots, v_m\}$, $\mu^+ = \min\{\mu_1, \mu_2, \ldots, \mu_m\}$.

Now,

$$\curlywedge\left((\ell+1)\left(1-\prod_{i=1}^{m}(1-\Upsilon(s_r,\triangle)^-/(\ell+1))^{w_i}\right)\right)\leq$$

$$\curlywedge\left((\ell+1)\left(1-\prod_{i=1}^{m}(1-\Upsilon(s_r,\triangle)/(\ell+1))^{w_i}\right)\right)\leq$$

$$\curlywedge\left((\ell+1)\left(1-\prod_{i=1}^{m}\left(1-\Upsilon(s_r,\triangle)^+/(\ell+1)\right)^{w_i}\right)\right),$$

$$\sqrt[q]{\prod_{i=1}^{m}(1-\mu^{+q})^{w_i}-\prod_{i=1}^{m}(1-v^{-q}-\mu^{+q})^{w_i}}\leq\sqrt[q]{\prod_{i=1}^{m}(1-\mu^q)^{w_i}-\prod_{i=1}^{m}(1-v^q-\mu^q)^{w_i}}$$

$$\leq\sqrt[q]{\prod_{i=1}^{m}(1-\mu^{-q})^{w_i}-\prod_{i=1}^{m}(1-v^{+q}-\mu^{-q})^{w_i}},$$

and

$$\sqrt[q]{1-\prod_{i=1}^{m}(1-\mu^{-q})^{w_i}}\leq\sqrt[q]{1-\prod_{i=1}^{m}(1-\mu^q)^{w_i}}\leq\sqrt[q]{1-\prod_{i=1}^{m}(1-\mu^{+q})^{w_i}}.$$

Based on the above inequalities, we can write

$$\partial^-\leq q-RFLWG(\partial_1,\partial_2,\ldots,\partial_m)\leq\partial^+.$$

**Theorem 17** *(Monotonicity) Let* $\partial_i=\left(\left(s_{r_i},\triangle_i\right),\langle v_i,\mu_i\rangle\right)$ *and* $\bar{\partial}_i=$

$\left(\overline{\left(s_{r_i},\triangle_i\right)},\langle\bar{v}_i,\bar{\mu}_i\rangle\right)(i=1,2,\ldots,m)$ *be two collections of q-RFLNs. If* $\left(s_{r_i},\triangle_i\right)\geq\overline{\left(s_{r_i},\triangle_i\right)}$,
$v_i\geq\bar{v}_i$ *and* $\mu_i\leq\bar{\mu}_i\,\forall i$, *then*

$$q-RFLWG(\partial_1,\partial_2,\ldots,\partial_m)\geq q-RFLWG(\bar{\partial}_1,\bar{\partial}_2,\ldots,\bar{\partial}_m). \tag{30}$$

**Proof.** Since $\left(s_{r_i},\triangle_i\right)\geq\overline{\left(s_{r_i},\triangle_i\right)}$, $v_i\geq\bar{v}_i$ and $\mu_i\leq\bar{\mu}_i\,\forall i$. Based on these, we have

$$\curlywedge\left((\ell+1)\left(1-\prod_{i=1}^{m}(1-\Upsilon(s_r,\triangle)/(\ell+1))^{w_i}\right)\right)\geq$$

$$\curlywedge\left((\ell+1)\left(1-\prod_{i=1}^{m}\left(1-\Upsilon\overline{(s_r,\triangle)}/(\ell+1)\right)^{w_i}\right)\right),$$

$$\sqrt[q]{\prod_{i=1}^{m}(1-\mu_i^q)^{w_i}-\prod_{i=1}^{m}(1-v_i^q-\mu_i^q)^{w_i}}\leq\sqrt[q]{\prod_{i=1}^{m}(1-\bar{\mu}_i^q)^{w_i}-\prod_{i=1}^{m}(1-\bar{v}_i^q-\bar{\mu}_i^q)^{w_i}},$$

and

$$\sqrt[q]{1-\prod_{i=1}^{m}(1-\mu_i^q)^{w_i}}\geq\sqrt[q]{1-\prod_{i=1}^{m}(1-\bar{\mu}_i^q)^{w_i}}.$$

The above inequalities imply that

$$\left(\begin{matrix}\curlywedge\left((\ell+1)\left(1-\prod_{i=1}^{m}\left(1-\Upsilon\left(s_{r_i},\triangle_i\right)/(\ell+1)\right)^{w_i}\right)\right),\\\left\langle\sqrt[q]{\prod_{i=1}^{m}(1-\mu_i^q)^{w_i}-\prod_{i=1}^{m}(1-v_i^q-\mu_i^q)^{w_i}},\sqrt[q]{1-\prod_{i=1}^{m}(1-\mu_i^q)^{w_i}}\right\rangle\end{matrix}\right)\geq$$

$$\left(\begin{matrix}\curlywedge\left((\ell+1)\left(1-\prod_{i=1}^{m}\left(1-\Upsilon\overline{\left(s_{r_i},\triangle_i\right)}/(\ell+1)\right)^{w_i}\right)\right),\\\left\langle\sqrt[q]{\prod_{i=1}^{m}(1-\bar{\mu}_i^q)^{w_i}-\prod_{i=1}^{m}(1-\bar{v}_i^q-\bar{\mu}_i^q)^{w_i}},\sqrt[q]{1-\prod_{i=1}^{m}(1-\bar{\mu}_i^q)^{w_i}}\right\rangle\end{matrix}\right).$$

Hence,

$q - RFLWG(\partial_1, \partial_2, \ldots, \partial_m) \geq q - RFLWG(\bar{\partial}_1, \bar{\partial}_2, \ldots, \bar{\partial}_m)$.

Following this, we discuss the q-rung orthopair fuzzy 2-tuple linguistic ordered weighted geometric (q-RFLOWG) operator and its properties.

**Definition 15** *Let* $\partial_i = \left( \left( s_{r_i}, \triangle_i \right), \langle v_i, \mu_i \rangle \right) (i = 1, 2, \ldots, m)$ *be a range of q-RFLNs with the position weights* $\varpi_i (i = 1, 2, \ldots, m)$ *such that* $\varpi_i \in [0, 1]$ *and* $\sum\limits_{i=1}^{m} \varpi_i = 1$ *then the operator q-RFLOWG:* $\partial^m \to \partial$ *is described as*

$$q - RFLOWG(\partial_1, \partial_2, \ldots, \partial_m) = \otimes_{i=1}^{m} \partial_{\hat{i}}^{\varpi_i}, \tag{31}$$

*where* $\left( \hat{1}, \hat{2}, \ldots, \hat{m} \right)$ *is the permutation of* $(i = 1, 2, \ldots, m)$*, in such a way as* $\partial_{\widehat{i-1}} \geq \partial_i$ *for i = 2, 3. . ., m.*

**Theorem 18** *Let* $\partial_i = \left( \left( s_{r_i}, \triangle_i \right), \langle v_i, \mu_i \rangle \right) (i = 1, 2, \ldots, m)$ *be a range of q-RFLNs with the position weights* $\varpi_i (i = 1, 2, \ldots, m)$ *such that* $\varpi_i \in [0, 1]$ *and* $\sum\limits_{i=1}^{m} \varpi_i = 1$*. Then the aggregated value of the* $q - RFLOWG$ *operator is still a q-RFLN, and*

$$
\begin{aligned}
q - RFLOWG(\partial_1, \partial_2, \ldots, \partial_m) &= \otimes_{i=1}^{m} \partial_{\hat{i}}^{\varpi_i} \\
&= \left( \begin{array}{c} \curlywedge \left( (\ell + 1) \prod\limits_{i=1}^{m} \left( \Upsilon \left( s_{r_i}, \triangle_i \right) / (\ell + 1) \right)^{\varpi_i} \right), \\ \left\langle \sqrt[q]{\prod\limits_{i=1}^{m} \left( 1 - \mu_i^q \right)^{\varpi_i} - \prod\limits_{i=1}^{m} \left( 1 - v_i^q - \mu_i^q \right)^{\varpi_i}}, \sqrt[q]{1 - \prod\limits_{i=1}^{m} \left( 1 - \mu_i^q \right)^{\varpi_i}} \right\rangle \end{array} \right).
\end{aligned}
\tag{32}
$$

**Proof.** Straightforward from Theorem 13.

Utilizing the $q - RFLOWG$ operator, the following features can be easily verified.

**Theorem 19** *(Permutation) Let* $\partial_i = \left( \left( s_{r_i}, \triangle_i \right), \langle v_i, \mu_i \rangle \right)$ *and* $\partial_{(i)} = \left( \left( s_{r_{(i)}}, \triangle_{(i)} \right), \left\langle v_{(i)}, \mu_{(i)} \right\rangle \right) (i = 1, 2, \ldots, m)$ *be two collections of q-RFLNs, where* $(\cdot)$ *indicates any permutation of i. Then*

$$q - RFLOWA(\partial_1, \partial_2, \ldots, \partial_m) = q - RFLOWA\left( \partial_{(1)}, \partial_{(2)}, \ldots, \partial_{(m)} \right). \tag{33}$$

**Theorem 20** *(Idempotency) Let* $\partial_i = \left( \left( s_{r_i}, \triangle_i \right), \langle v_i, \mu_i \rangle \right) (i = 1, 2, \ldots, m)$ *be a range of q-RFLNs with their position weight* $\varpi_i (i = 1, 2, \ldots, m)$ *such that* $\varpi_i \in [0, 1]$ *and* $\sum\limits_{i=1}^{m} \varpi_i = 1$ *if* $\partial_i = \partial \; \forall i$ *then*

$$q - RFLOWA(\partial_1, \partial_2, \ldots, \partial_m) = \partial. \tag{34}$$

**Theorem 21** *(Boundedness) Let* $\partial_i = \left( \left( s_{r_i}, \triangle_i \right), \langle v_i, \mu_i \rangle \right) (i = 1, 2, \ldots, m)$ *be a range of q-RFLNs. If* $\partial^- = \min(\partial_1, \partial_2, \ldots, \partial_m)$ *and* $\partial^+ = \max(\partial_1, \partial_2, \ldots, \partial_m)$*, then*

$$\partial^- \leq q - RFLOWA(\partial_1, \partial_2, \ldots, \partial_m) \leq \partial^+. \tag{35}$$

**Theorem 22** (Monotonicity) Let $\partial_i = \left(\left(s_{r_i}, \triangle_i\right), \langle v_i, \mu_i\rangle\right)$ and $\bar{\partial}_i = \left(\overline{\left(s_{r_i}, \triangle_i\right)}, \langle \bar{v}_i, \bar{\mu}_i\rangle\right) (i = 1, 2, \ldots, m)$ be two collections of q-RFLNs. If $\left(s_{r_i}, \triangle_i\right) \geq \overline{\left(s_{r_i}, \triangle_i\right)}$, $v_i \geq \bar{v}_i$ and $\mu_i \leq \bar{\mu}_i \,\forall i$, then

$$q - RFLOWA(\partial_1, \partial_2, \ldots, \partial_m) \geq q - RFLOWA\left(\bar{\partial}_1, \bar{\partial}_2, \ldots, \bar{\partial}_m\right). \tag{36}$$

In Definition 14, we see that q-RFLWG operator weights are the most basic form of q-RFLN, whereas q-RFLOWG operator weights are the special type of the organized locations of the q-RFLNs. As a result, the weights specified in the operators q-RFLWG and q-RFLOWG present various scenarios that are antagonistic to one another. In any event, in terms of the general approach, these perspectives are considered equivalent. To avoid such inconvenience, we provide the q-RFL hybrid geometric (q-RFLHG) operator in the following.

**Definition 16** Let $\partial_i = \left(\left(s_{r_i}, \triangle_i\right), \langle v_i, \mu_i\rangle\right)(i = 1, 2, \ldots, m)$ be a range of q-RFLNs with the position weights $\varpi_i(i = 1, 2, \ldots, m)$ such that $\varpi_i \in [0, 1]$ and $\sum_{i=1}^{m} \varpi_i = 1$ then the operator q-RFLHG: $\partial^m \rightarrow \partial$ is described as

$$q - RFLHG(\partial_1, \partial_2, \ldots, \partial_m) = \otimes_{i=1}^{m} \dot{\partial}_i^{\varpi_i}, \tag{37}$$

where $\dot{\partial}_i = \partial_i^{mw_i}, i = 1, 2, \ldots, m, \left(\hat{\dot{1}}, \hat{\dot{2}}, \ldots, \hat{\dot{m}}\right)$ is the permutation of $(i = 1, 2, \ldots, m)$, in such a way as $\partial_{\widehat{i-1}} \geq \dot{\partial}_{\hat{i}}$ for $i = 2, 3\ldots, m$; $w = (w_1, w_2, \ldots, w_m)^T$ is the weight vector of $\partial_i$ such that $w_i \in [0, 1]$ and $\sum_{i=1}^{m} w_i = 1$, and m is the balancing coefficient.

**Theorem 23** Let $\partial_i = \left(\left(s_{r_i}, \triangle_i\right), \langle v_i, \mu_i\rangle\right)(i = 1, 2, \ldots, m)$ be a range of q-RFLNs with the position weights $\varpi_i(i = 1, 2, \ldots, m)$ such that $\varpi_i \in [0, 1]$ and $\sum_{i=1}^{m} \varpi_i = 1$. Then the aggregated value of the $q - RFLHG$ operator is still a q-RFLN, and

$$q - RFLHG(\partial_1, \partial_2, \ldots, \partial_m) = \otimes_{i=1}^{m} \dot{\partial}_{\hat{i}}^{\varpi_i}$$

$$= \left( \begin{array}{c} \curlywedge\left((\ell+1)\prod_{i=1}^{m}\left(\curlyvee\left(s_{r_{\hat{i}}}, \triangle_{\hat{i}}\right)/(\ell+1)\right)^{\varpi_i}\right), \\ \sqrt[q]{\prod_{i=1}^{m}\left(1 - \mu_{\hat{i}}^q\right)^{\varpi_i} - \prod_{i=1}^{m}\left(1 - v_{\hat{i}}^q - \mu_{\hat{i}}^q\right)^{\varpi_i}}, \left\langle \sqrt[q]{1 - \prod_{i=1}^{m}\left(1 - \mu_{\hat{i}}^q\right)^{\varpi_i}} \right\rangle \end{array} \right). \tag{38}$$

**Proof.** Straightforward from Theorem 13.

The q-RFLHG operator, like the q-RFLWG and q-RFLOWG operators, has permutation, idempotency, boundedness, and monotonicity features. Aside from the aforementioned properties, the q-RFLHG operator has the following special instances.

**Corollary 3** q-RFLWG operator is a special case of q-RFLHG operator.

**Proof.** Let $\varpi = \left(\frac{1}{m}, \frac{1}{m}, \ldots, \frac{1}{m}\right)^T$, the $q - RFLHG(\partial_1, \partial_2, \ldots, \partial_m) = \dot{\partial}_{\hat{1}}^{\varpi_1} \otimes \dot{\partial}_{\hat{2}}^{\varpi_2} \otimes \ldots \otimes \dot{\partial}_{\hat{m}}^{\varpi_m} = \left(\dot{\partial}_{\hat{1}} \otimes \dot{\partial}_{\hat{2}} \otimes \ldots \otimes \dot{\partial}_{\hat{m}}\right)^{\frac{1}{m}} = \partial_1^{w_1} \otimes \partial_2^{w_2} \otimes \ldots \otimes \partial_m^{w_m} = q - RFLWG(\partial_1, \partial_2, \ldots, \partial_m)$.

**Corollary 4** q-RFLOWG operator is a special case of q-RFLHG operator.

**Proof.** Let $w = \left(\frac{1}{m}, \frac{1}{m}, \ldots, \frac{1}{m}\right)^T$, the $q - RFLHG(\partial_1, \partial_2, \ldots, \partial_m) = \partial_{\dot{1}}^{\varpi_1} \otimes \partial_{\dot{2}}^{\varpi_2} \otimes \ldots \otimes \partial_{\dot{m}}^{\varpi_m} = \partial_{\dot{1}}^{\varpi_1} \otimes \partial_{\dot{2}}^{\varpi_2} \otimes \ldots \otimes \partial_{\dot{m}}^{\varpi_m} q - RFLOWG(\partial_1, \partial_2, \ldots, \partial_m).$

## 5 q-RFL VIKOR-QUALIFLEX method with unknown criteria weight

**Problem description**

It is presumed that $k$ DMks $D = \{d_1, d_2, \ldots, d_k\}$ need to rank the $m$ alternatives $X = \{x_1, x_2, \ldots, x_m\}$ for decision making. To do so, $n$ assessment criteria $C = \{c_1, c_2, \ldots, c_n\}$ are chosen, which are divided into two classes: benefit criteria set and cost criteria set. Suppose the DMk $d_t$ the q-RFL decision matrix (q-RFLDM) $M^t = \left(\partial_{ij}^t\right)_{m \times n}$ of which elements $\partial_{ij}^t = \left(\left(s_{r_{ij}}^t, \triangle_{ij}^t\right), \left\langle v_{ij}^t, \mu_{ij}^t \right\rangle\right)$ is a q-RFLN, which represents the alternative $x_i$ assessment information with respect to the criteria $c_j$. Let $\Omega = (\Omega_1, \Omega_2, \ldots, \Omega_k)^T$ be the weight vector of DMks, and $w = (w_1, w_2, \ldots, w_n)^T$ be the weight vector of criteria, where $0 \leq \Omega_t \leq 1, \sum_{t=1}^{k} \Omega_t = 1, 0 \leq w_j \leq 1$, and $\sum_{j=1}^{n} w_j = 1$.

To solve the aforesaid MCGDM problems, a VIKOR-QUALIFLEX approach based on q-RFL information is constructed in the present section. This method is separated into four stages: gathering assessment information, calculating criteria weight, calculating concordance index, and ranking alternatives. The details of each stage are added as follows:

### 5.1 Stage 1: Obtaining assessment information

Regarding a group assessment problem, $k$ DMks $\{d_1, d_2, \ldots, d_k\}$ attempt to evaluate $m$ alternatives $\{x_1, x_2, \ldots, x_m\}$. The evaluation information is represented by q-RFLNs to explain the information appropriately. In the process of evaluation, DMks are allowed to choose the linguistic terms in line with the LTS:

$$S = \left\{ \begin{array}{c} s_0 = \text{extremely poor}, s_1 = \text{very poor}, s_2 = \text{poor}, s_3 = \text{medium}, s_4 = \text{good}, s_5 = \text{very good}, \\ s_6 = \text{extremely good} \end{array} \right\}.$$

**Step 1:** Individual decision matrices: Frame the DMks' assessment information in terms of q-RFLDM

$$M_{m \times n}^t =$$

$$\begin{array}{c} \\ o_1 \\ \vdots \\ o_i \\ \vdots \\ o_m \end{array} \begin{pmatrix} \overset{c_1}{\left(\left(s_{r_{11}}^{(t)}, \triangle_{11}^{(t)}\right), \left\langle v_{11}^{(t)}, \mu_{11}^{(t)} \right\rangle\right)} & \cdots & \overset{c_j}{\left(\left(s_{r_{1j}}^{(t)}, \triangle_{1j}^{(t)}\right), \left\langle v_{1j}^{(t)}, \mu_{1j}^{(t)} \right\rangle\right)} & \cdots & \overset{c_n}{\left(\left(s_{r_{1n}}^{(t)}, \triangle_{1n}^{(t)}\right), \left\langle v_{1n}^{(t)}, \mu_{1n}^{(t)} \right\rangle\right)} \\ \vdots & \ddots & \vdots & \ddots & \vdots \\ \left(\left(s_{r_{i1}}^{(t)}, \triangle_{i1}^{(t)}\right), \left\langle v_{i1}^{(t)}, \mu_{i1}^{(t)} \right\rangle\right) & \cdots & \left(\left(s_{r_{ij}}^{(t)}, \triangle_{ij}^{(t)}\right), \left\langle v_{ij}^{(t)}, \mu_{ij}^{(t)} \right\rangle\right) & \cdots & \left(\left(s_{r_{in}}^{(t)}, \triangle_{in}^{(t)}\right), \left\langle v_{ij}^{(t)}, \mu_{in}^{(t)} \right\rangle\right) \\ \vdots & \ddots & \vdots & \ddots & \vdots \\ \left(\left(s_{r_{m1}}^{(t)}, \triangle_{m1}^{(t)}\right), \left\langle v_{m1}^{(t)}, \mu_{m1}^{(t)} \right\rangle\right) & \cdots & \left(\left(s_{r_{mj}}^{(t)}, \triangle_{mj}^{(t)}\right), \left\langle v_{mj}^{(t)}, \mu_{mj}^{(t)} \right\rangle\right) & \cdots & \left(\left(s_{r_{mn}}^{(t)}, \triangle_{mn}^{(t)}\right), \left\langle v_{mn}^{(t)}, \mu_{mn}^{(t)} \right\rangle\right) \end{pmatrix}$$

where $\left(\left(s_{r_{ij}}^{(t)}, \triangle_{ij}^{(t)}\right), \left\langle v_{ij}^{(t)}, \mu_{ij}^{(t)} \right\rangle\right)$ symbolizes q-RFLN provided by $t$th DMk to which alternative $o_i$ meets that the criteria $c_j$ having the constraint that $0 \leq v_{ij}^q + \mu_{ij}^q \leq 1$.

**Step 2:** Discriminate the criteria: Criteria are classified into two types: benefit criteria and cost criteria. The benefit criteria imply that a higher evaluation is preferable, but the cost criteria indicate that a lower assessment is preferable. To ensure consistency, we use the formulation Eq (39) to convert the assessments under the cost criterion into corresponding benefit types:

$$\widetilde{\partial^{(t)}}_{ij} = \begin{cases} \partial_{ij}^{(t)}, & c_j \text{ is benefit criteria,} \\ \partial_{ij}^{(t)^c}, & c_j \text{ is cost criteria,} \end{cases} \tag{39}$$

where $\partial_{ij}^{(t)^c}$ represents the complement of $\partial_{ij}^{(t)}$.

**Step 3:** DMks weight: Each DMk has a unique background, knowledge, and experience. As a result, they should not be regarded equally and should be assigned weights. To that purpose, their weight vector $\Omega = (\Omega_1, \Omega_2, \ldots, \Omega_k)^T$ is calculated by using the following technique: Based on the decision matrices $M_{m \times n}^t (t = 1, 2, \ldots, k)$, the average of the evaluations on alternative $x_i$ under the criteria $c_j$ is computed via the proposed AO Eq (11) as follows:

$$\overline{\partial_{ij}} = \frac{1}{k} \oplus_{t=1}^{k} \partial_{ij}^{(t)} =$$
$$\left( \begin{array}{c} \lambda\left( (\ell+1)\left( 1 - \prod_{t=1}^{k}\left( 1 - \Upsilon\left( s_{r_{ij}}^{(t)}, \triangle_{ij}^{(t)} \right)/(\ell+1) \right)^{\frac{1}{k}} \right) \right), \\ \left\langle \sqrt[q]{1 - \prod_{t=1}^{k}\left( 1 - v_{ij}^{(t)^q} \right)^{\frac{1}{k}}}, \sqrt[q]{\prod_{t=1}^{k}\left( 1 - v_{ij}^{(t)^q} \right)^{\frac{1}{k}} - \prod_{t=1}^{k}\left( 1 - v_{ij}^{(t)^q} - \mu_{ij}^{(t)^q} \right)^{\frac{1}{k}}} \right\rangle \end{array} \right). \tag{40}$$

Then, measure the degree of similarity between $\partial_{ij}^{(t)}$ and $\overline{\partial_{ij}}$ by Eq (10).

$$S_{ij}^{(t)} = 1 - \frac{d\left( \partial_{ij}^{(t)}, \overline{\partial_{ij}} \right)}{\sum_{t=1}^{k} d\left( \partial_{ij}^{(t)}, \overline{\partial_{ij}} \right)}, \tag{41}$$

where $d\left( \partial_{ij}^{(t)}, \overline{\partial_{ij}} \right)$ denotes the distance betwixt $\partial_{ij}^{(t)}$ and $\overline{\partial_{ij}}$.

Next, we measure the overall degree of similarity of DMk $d_t$ under the criteria $c_j$ as follows:

$$S_j^{(t)} = \sum_{i=1}^{m} S_{ij}^{(t)}. \tag{42}$$

Afterward, the weight of $d_t$ under the criteria $c_j$ is calculated as

$$\Omega_j^{(t)} = \frac{S_j^{(t)}}{\sum_{t=1}^{k} S_j^{(t)}}. \tag{43}$$

Finally, we determine the weight of each $d_t$ $(t = 1, 2, \ldots, k)$ by Eq (44)

$$\Omega^{(t)} = \frac{\max\limits_{1 \le j \le n} \Omega_j^{(t)}}{\sum\limits_{t=1}^{k} \left( \max\limits_{1 \le j \le n} \Omega_j^{(t)} \right)} . \tag{44}$$

**Step 4:** Global decision matrix:

Aggregate all q-RFLDMs $M^t(t = 1, 2, \ldots, k)$ into a single one

$M = \left( \partial_{ij} \right)_{m \times n} = \left( \left( \left( s_{r_{ij}}, \triangle_{ij} \right), \left\langle v_{ij}, \mu_{ij} \right\rangle \right) \right)_{m \times n}$, where $\partial_{ij}$ is generated by employing the q-RFLWA operator in Eq (11) as follows:

$$\partial_{ij} = q - RFLWA\left( \partial_{ij}^{(1)}, \partial_{ij}^{(2)}, \ldots, \partial_{ij}^{(k)} \right) =$$

$$\left( \begin{array}{c} \curlywedge\left( (\ell + 1)\left( 1 - \prod\limits_{t=1}^{k} \left( 1 - \curlyvee\left( s_{r_{ij}}^{(t)}, \triangle_{ij}^{(t)} \right)/(\ell + 1) \right)^{\Omega^{(t)}} \right) \right), \\ \left\langle \sqrt[q]{1 - \prod\limits_{t=1}^{k} \left( 1 - v_{ij}^{(t)q} \right)^{\Omega^{(t)}}}, \sqrt[q]{\prod\limits_{t=1}^{k} \left( 1 - v_{ij}^{(t)q} \right)^{\Omega^{(t)}} - \prod\limits_{t=1}^{k} \left( 1 - v_{ij}^{(t)q} - \mu_{ij}^{(t)q} \right)^{\Omega^{(t)}}} \right\rangle \end{array} \right) . \tag{45}$$

## 5.2 Stage 2: Criteria weight determination

In the practical MCGDM problems, it is possible for information about criteria weights to be fully unknown or only partially known. As a result, focusing on this subject is both interesting and necessary. Hwang and Yoon's TOPSIS [64] is a traditional MCGDM technique that chooses the best alternative with the smallest distance from the positive-ideal solution and the greatest distance from the negative-ideal solution. The objective of this stage is to create an optimization model based on the TOPSIS approach for objectively determining the criteria weights. Assume $w = (w_1, w_2, \ldots, w_n)^t$ is the unknown weight vector of the criteria $c_j(j = 1, 2, \ldots, n)$. The models for precisely calculating the weights of criteria are detailed below.

**Step 1:** Positive ideal solution and negative ideal solution: In the first step, the positive-ideal and negative-ideal solutions, labeled as $\mathcal{I}^+$ and $\mathcal{I}^-$, can be found by

$$\mathcal{I}^+ = [\partial_j^+]_{1 \times n} = \begin{cases} \max\limits_i \left\{ \partial_{ij} \right\}, & c_j \text{ is benefit criteria,} \\ \\ \min\limits_i \left\{ \partial_{ij} \right\}, & c_j \text{ is cost criteria,} \end{cases} \tag{46}$$

$$\mathcal{I}^- = [\partial_j^-]_{1 \times n} = \begin{cases} \min\limits_i \left\{ \partial_{ij} \right\}, & c_j \text{ is benefit criteria,} \\ \\ \max\limits_i \left\{ \partial_{ij} \right\}, & c_j \text{ is cost criteria,} \end{cases} \tag{47}$$

Note: The comparison technique of $\partial_{ij}$ is detailed in Section 2.

**Step 2:** Weighted closeness coefficients of alternatives: The closeness coefficient of each alternative to the criteria is evaluated by Eq (48) as shown:

$$D_{ij} = \frac{d\left(\partial_{ij}, \partial_j^-\right)}{d\left(\partial_{ij}, \partial_j^+\right) + d\left(\partial_{ij}, \partial_j^-\right)}, \ i = 1, 2, \ldots, m, \ j = 1, 2, \ldots, n, \tag{48}$$

where $d\left(\partial_{ij}, \partial_j^+\right) \left(d\left(\partial_{ij}, \partial_j^-\right)\right)$ represents the distance of $\partial_i$ from $\mathcal{I}^+$ $(\mathcal{I}^-)$ with respect to criteria $c_j$.

The weighted closeness coefficient of alternative $x_i$ is then computed by

$$D_i = \sum_{j=1}^n w_j D_{ij} = \sum_{j=1}^n w_j \frac{d\left(\partial_{ij}, \partial_j^-\right)}{\left(\partial_{ij}, \partial_j^+\right) + \left(\partial_{ij}, \partial_j^-\right)}, \ i = 1, 2, \ldots, m. \tag{49}$$

**Step 3:** Optimal weights of criteria:

The weighted closeness coefficient $D_i$ shows the alternative's $x_i$ relative closeness to the ideal solution; the higher the value of $D_i$, the better the alternative. When the weight information is fully unknown, the following linear programming model can be developed by taking all of the $m$ alternatives into account:

$$M_I \begin{cases} \max D(w) = \sum_{i=1}^m D_i = \sum_{i=1}^m \sum_{j=1}^n w_j D_{ij} \\ \\ s.t. w_j \geq 0, \ j = 1, 2, .., n, \ \sum_{j=1}^n w_j = 1. \end{cases} \tag{50}$$

By solving the aforesaid model, the optimal solutions are normalized to yield the weights of criteria as

$$w_j = \frac{\sum_{i=1}^m D_{ij}}{\sum_{j=1}^n \sum_{i=1}^m D_{ij}}. \tag{51}$$

Moreover, in certain circumstances, the information relating criteria weights is only partially recognized. The resulting weight information typically comprises the five structural forms [65, 66], for $i \neq j$: i). A weak ranking $R_1 = \{w_i \geq w_j\}$; ii). A strict ranking $R_2 = \{w_i - w_j \geq \alpha_j\}$ $(\alpha_j > 0)$; iii). A ranking of differences $R_3 = \{w_i - w_j \geq w_k - w_l\}$ $(j \neq k \neq l)$; iv). A ranking with multiples $R_4 = \{w_i \geq \alpha_j w_j\}$ $(0 \leq \alpha_j \leq 1)$; v). An interval form: $R_5 = \{\alpha_i \leq w_i \leq \alpha_i + \epsilon_i\}$ $(0 \leq \alpha_i \leq \alpha_i + \epsilon_i)$. For simplicity, $R$ is assumed to represent a set of existing weight information and $R = R_1 \bigcup R_2 \bigcup R_3 \bigcup R_4 \bigcup R_5$. In these cases, the single objective optimization model in Eq (52)

can be set up.

$$M_{II} \begin{cases} \max D(w) = \sum_{i=1}^{m} D_i = \sum_{i=1}^{m}\sum_{j=1}^{n} w_j D_{ij} \\ \\ s.t. w \in R, \ w_j \geq 0, \ j = 1, 2, .., n, \ \sum_{j=1}^{n} w_j = 1. \end{cases} \quad (52)$$

The criteria weights $w = (w_1, w_2, \ldots, w_n)^t$ can be derived by running model $M_{II}$ with Lingo software.

## 5.3 Stage 3: Concordance index based on VIKOR

The VIKOR technique was initially introduced by Opricovic and Tzeng [67]. The VIKOR technique's primary premise is to maximize collective gain while minimizing personal regret. The ranking is determined by first determining the ideal solutions, then computing the group utility value, the individual regret value, and the compromise solution. The strategy works for DMks who prioritise financial gain.

The process for determining the concordance index is displayed as follows:

**Step 1:** Optimal solutions: Following the approach outlined in Stage 2, evaluate the optimal solutions, i.e., positive ideal solution $\mathcal{I}^+$ and negative ideal solution $\mathcal{I}^-$.

**Step 2:** Hamming distance: Determine the Hamming distance between each q-RFLN and its accompanying positive ideal solution using the formula below.

$$d\left(\partial_{ij}, \partial_j^+\right) = \frac{1}{2(\ell+1)} \left[ \left| \left(1 + v_{ij}^q - \mu_{ij}^q\right).\Upsilon\left(s_{r_{ij}}, \triangle_{ij}\right) - \left(1 + v_j^{+q} - \mu_j^{+q}\right).\Upsilon\left(s_{r_j}^+, \triangle_j^+\right) \right| \right]. \quad (53)$$

**Step 3:** Group utility and individual regret values: Determine each evaluation alternative's group utility value $\mathcal{U}_i (i = 1, 2, \ldots, m)$ and individual regret value $\mathcal{V}_i (i = 1, 2, \ldots, m)$, where

$$\mathcal{U}_i = \sum_{j=1}^{n} \frac{w_j d\left(\partial_{ij}, \partial_j^+\right)}{d\left(\partial_j^-, \partial_j^+\right)} \ i = 1, 2, \ldots, m, \quad (54)$$

$$\mathcal{V}_i = \max_j \frac{w_j d\left(\partial_{ij}, \partial_j^+\right)}{d\left(\partial_j^-, \partial_j^+\right)} \ i = 1, 2, \ldots, m. \quad (55)$$

**Step 4:** Concordance index:
Compute the concordance index of $x_i (i = 1, 2, \ldots, m)$ over option $x_k (k = 1, 2, \ldots, m)$ using the formulation given below:

$$\mathscr{I}_{ik} = ¥ \frac{\mathcal{U}_k - \mathcal{U}_i}{\mathcal{U}^+ - \mathcal{U}^-} + (1 - ¥) \frac{\mathcal{V}_k - \mathcal{V}_i}{\mathcal{V}^+ - \mathcal{V}^-}; \ i, k = 1, 2, \ldots, m, \quad (56)$$

where $\mathcal{U}^+ = \max_i \mathcal{U}_i, \mathcal{U}^- = \min_i \mathcal{U}_i, \mathcal{V}^+ = \max_i \mathcal{V}_i, \mathcal{V}^- = \min_i \mathcal{V}_i$. ¥ is a decision-

making process coefficient capable of balancing the weights of group utility and individual regret.

## 5.4 Stage 4: Ranking result based on QUALIFLEX

A ranking order based on QUALIFLEX can be generated using the concordance index. It is extremely useful and efficient when the number of indexes far outnumbers the number of evaluation alternatives in a decision-making task.

Here are the details:

**Step 1:** Possible permutations of alternatives:

Itemize all probable rankings of $m$ alternatives, symbolized as $\Re_p = (\ldots, x_i, \ldots, x_k, \ldots)$, where alternative $x_i$ is better than $x_k$, and $k = 1, 2, \ldots, m!$.

**Step 2:** General concordance index:

Apply Eq (57) to each permutation to determine the general concordance index:

$$\mathscr{I}_p = \sum_{i=1}^{m!-1} \sum_{k=i+1}^{m!} \mathscr{I}_{ik}; \ p = 1, 2, \ldots, m!. \tag{57}$$

**Step 3:** Ranking result:

Get the rank corresponding to the values of $\mathscr{I}_p$ $(p = 1, 2, \ldots, m!)$. The greater the value of $\mathscr{I}_p$, the better the permutation $\Re_p$.

# 6 Case study

With the advancement of the economy, science, and technology, several challenges, just like global warming, have emerged as serious concerns for all states on the planet. One of the primary factors of global warming is the high release of greenhouse gases, which overloads the Earth's circulatory system. One of the biggest sources of greenhouse gases is the emission of varied vehicle exhaust during transportation. As a result, governments have long advocated for low-carbon and environmentally friendly travel.

Since its inauguration in Beijing universities in 2014, BS has been highly accepted by academics and students as a low-carbon and environmentally friendly mode of transportation. BS effectively resolves the challenge of the last kilometer of public transportation trips with other public transportation instruments. Furthermore, BS can be used to supplement some non-green transportation options, lowering greenhouse gas emissions and safeguarding the surroundings. Furthermore, it has been stated that there is a widespread deficit of physical activity and sports among today's young people. BS can also improve personal exercise, which is good for both physical and mental health. Because of these numerous advantages, BS has attracted significant attention and development. According to Ministry of Transport figures, 19.5 million BS services were introduced in 2019, with a total of 235 million customers. There is no doubt that BS provides several advantages to individuals, the community, and the environment; nonetheless, the waste of resources, road occupancy, and pollution produced by abandoned bicycles have become one of the major difficulties confronting the expansion of BS. However, if the abandoned BS program is not handled effectively, it will not only impede the advancement of the operators but will also hurt several other elements. As a result, recycling

rejected BS should become a top priority. How to choose the most appropriate supplier by using the MCGDM approach is the key to the recycling of BS. Several factors influence the selection of BS providers, and the entire procedure covers transportation, logistics, trash management, and other industries. As a result, choosing providers who meet all criteria standards is quite challenging. BS operators can utilize scientific decision-making methods to decide on suppliers, which is favorable to attaining good sustainable growth in the expanding global BS market trend. As a research incentive, this work implements the propound methodology to assess and decide recycling suppliers. The methodology for choosing the most acceptable recycling suppliers is detailed in this part, including aspects that influence decision-making, such as risk-attitude factors and group consensus, which are both evaluated in the group choice.

In this case, the BS operators are the direct clients, while the recycling service provider is the supplier. In this example, the four alternative recycling service suppliers are denoted by $x_1$, $x_2$, $x_3$ and $x_4$. The three experts are represented by $d_1$, $d_2$, and $d_3$. The five evaluation criteria are detailed in Table 1.

## 6.1 Method implementation process

**Case 1:** Assume that the criteria weights are totally unknown:

The detailed calculating steps for selecting the best one of the four suppliers are shown below.

**Stage 1:** Obtaining assessment information:

Step 1: The initial evaluation matrices provided by three DMks $d_1$, $d_2$, and $d_3$ are listed in Tables 2–4, respectively.

Step 2: Inspection reveals that the criteria $c_4$ is of the cost type, according to Eq (39), the normalized decision matrices are derived, as shown in Tables 5–7.

**Table 1. Supplier selection criteria.**

| Criteria | Illustration |
|---|---|
| $c_1$: Green image | Green customer market share; examination of firms' performance in terms of environmental obligations of suppliers; collaboration with green groups; environmental awareness and green business philosophy [68, 69]. |
| $c_2$: Financial ability | Including financial management, activities, connections, and efficiency to support the normal execution of numerous linkages such as retrieval, transit, disassembly, and process [68, 69]. |
| $c_3$: Recovery capacity | Total number of bikes retrieved, quantity retrieved each time, retrieval rate (ratio of quantity retrieved to actual quantity retrieved), and special infrastructure; technological input in the recycling process, such as recycling scheme design, degree of specialization, process level, and so on [69]. |
| $c_4$: Pollution and emission | Pollution and discharge are critical criteria for assessing nature conservation. They are quantifiable in terms of the amount of pollutants (waste water, waste gas, etc.) produced in a certain time period [69–71]. |
| $c_5$: Searching ability | Finding abandoned shared bikes [69, 72]. |

**Table 2. q-RFLDM $M^1$.**

| | $c_1$ | $c_2$ | $c_3$ | $c_4$ | $c_5$ |
|---|---|---|---|---|---|
| $x_1$ | $((s_5, 0), \langle 0.55, 0.40 \rangle)$ | $((s_6, 0), \langle 0.60, 0.45 \rangle)$ | $((s_5, 0), \langle 0.68, 0.44 \rangle)$ | $((s_6, 0), \langle 0.55, 0.30 \rangle)$ | $((s_4, 0), \langle 0.70, 0.53 \rangle)$ |
| $x_2$ | $((s_4, 0), \langle 0.50, 0.50 \rangle)$ | $((s_5, 0), \langle 0.66, 0.40 \rangle)$ | $((s_3, 0), \langle 0.44, 0.33 \rangle)$ | $((s_3, 0), \langle 0.69, 0.48 \rangle)$ | $((s_6, 0), \langle 0.28, 0.38 \rangle)$ |
| $x_3$ | $((s_2, 0), \langle 0.54, 48 \rangle)$ | $((s_5, 0), \langle 0.71, 0.50 \rangle)$ | $((s_3, 0), \langle 0.40, 0.55 \rangle)$ | $((s_4, 0), \langle 0.40, 0.20 \rangle)$ | $((s_0, 0), \langle 0.38, 0.33 \rangle)$ |
| $x_4$ | $((s_3, 0), \langle 0.41, 0.41 \rangle)$ | $((s_2, 0), \langle 0.50, 0.50 \rangle)$ | $((s_0, 0), \langle 0.58, 0.44 \rangle)$ | $((s_1, 0), \langle 0.60, 0.44 \rangle)$ | $((s_3, 0), \langle 0.45, 0.40 \rangle)$ |

**Table 3. q-RFLDM $M^2$.**

|        | $c_1$ | $c_2$ | $c_3$ | $c_4$ | $c_5$ |
|--------|-------|-------|-------|-------|-------|
| $x_1$ | $((s_6, 0), \langle 0.56, 0.41 \rangle)$ | $((s_5, 0), \langle 0.60, 0.45 \rangle)$ | $((s_6, 0), \langle 0.69, 0.45 \rangle)$ | $((s_5, 0), \langle 0.56, 0.30 \rangle)$ | $((s_4, 0), \langle 0.70, 0.54 \rangle)$ |
| $x_2$ | $((s_5, 0), \langle 0.48, 0.48 \rangle)$ | $((s_4, 0), \langle 0.65, 0.40 \rangle)$ | $((s_4, 0), \langle 0.45, 0.32 \rangle)$ | $((s_2, 0), \langle 0.69, 0.48 \rangle)$ | $((s_5, 0), \langle 0.27, 0.38 \rangle)$ |
| $x_3$ | $((s_3, 0), \langle 0.55, 49 \rangle)$ | $((s_4, 0), \langle 0.72, 0.52 \rangle)$ | $((s_4, 0), \langle 0.42, 0.55 \rangle)$ | $((s_5, 0), \langle 0.42, 0.25 \rangle)$ | $((s_0, 0), \langle 0.35, 0.30 \rangle)$ |
| $x_4$ | $((s_2, 0), \langle 0.40, 0.40 \rangle)$ | $((s_1, 0), \langle 0.52, 0.52 \rangle)$ | $((s_0, 0), \langle 0.60, 0.44 \rangle)$ | $((s_2, 0), \langle 0.62, 0.45 \rangle)$ | $((s_3, 0), \langle 0.45, 0.40 \rangle)$ |

**Table 4. q-RFLDM $M^3$.**

|        | $c_1$ | $c_2$ | $c_3$ | $c_4$ | $c_5$ |
|--------|-------|-------|-------|-------|-------|
| $x_1$ | $((s_4, 0), \langle 0.54, 0.38 \rangle)$ | $((s_6, 0), \langle 0.58, 0.42 \rangle)$ | $((s_4, 0), \langle 0.65, 0.40 \rangle)$ | $((s_5, 0), \langle 0.55, 0.30 \rangle)$ | $((s_4, 0), \langle 0.68, 0.50 \rangle)$ |
| $x_2$ | $((s_3, 0), \langle 0.55, 0.55 \rangle)$ | $((s_6, 0), \langle 0.68, 0.42 \rangle)$ | $((s_2, 0), \langle 0.45, 0.35 \rangle)$ | $((s_4, 0), \langle 0.70, 0.50 \rangle)$ | $((s_5, 0), \langle 0.30, 0.40 \rangle)$ |
| $x_3$ | $((s_1, 0), \langle 0.52, 45 \rangle)$ | $((s_4, 0), \langle 0.70, 0.50 \rangle)$ | $((s_2, 0), \langle 0.38, 0.52 \rangle)$ | $((s_3, 0), \langle 0.40, 0.20 \rangle)$ | $((s_0, 0), \langle 0.40, 0.35 \rangle)$ |
| $x_4$ | $((s_2, 0), \langle 0.42, 0.42 \rangle)$ | $((s_3, 0), \langle 0.50, 0.50 \rangle)$ | $((s_0, 0), \langle 0.60, 0.45 \rangle)$ | $((s_2, 0), \langle 0.35, 0.40 \rangle)$ | $((s_3, 0), \langle 0.55, 0.25 \rangle)$ |

**Table 5. Normalized q-RFLDM $M^1$.**

|        | $c_1$ | $c_2$ | $c_3$ | $c_4$ | $c_5$ |
|--------|-------|-------|-------|-------|-------|
| $x_1$ | $((s_5, 0), \langle 0.55, 0.40 \rangle)$ | $((s_6, 0), \langle 0.60, 0.45 \rangle)$ | $((s_5, 0), \langle 0.68, 0.44 \rangle)$ | $((s_6, 0), \langle 0.30, 0.55 \rangle)$ | $((s_4, 0), \langle 0.70, 0.53 \rangle)$ |
| $x_2$ | $((s_4, 0), \langle 0.50, 0.50 \rangle)$ | $((s_5, 0), \langle 0.66, 0.40 \rangle)$ | $((s_3, 0), \langle 0.44, 0.33 \rangle)$ | $((s_3, 0), \langle 0.48, 0.69 \rangle)$ | $((s_6, 0), \langle 0.28, 0.38 \rangle)$ |
| $x_3$ | $((s_2, 0), \langle 0.54, 48 \rangle)$ | $((s_5, 0), \langle 0.71, 0.50 \rangle)$ | $((s_3, 0), \langle 0.40, 0.55 \rangle)$ | $((s_4, 0), \langle 0.20, 0.40 \rangle)$ | $((s_0, 0), \langle 0.38, 0.33 \rangle)$ |
| $x_4$ | $((s_3, 0), \langle 0.41, 0.41 \rangle)$ | $((s_2, 0), \langle 0.50, 0.50 \rangle)$ | $((s_0, 0), \langle 0.58, 0.44 \rangle)$ | $((s_1, 0), \langle 0.44, 0.60 \rangle)$ | $((s_3, 0), \langle 0.45, 0.40 \rangle)$ |

**Table 6. Normalized q-RFLDM $M^2$.**

|        | $c_1$ | $c_2$ | $c_3$ | $c_4$ | $c_5$ |
|--------|-------|-------|-------|-------|-------|
| $x_1$ | $((s_6, 0), \langle 0.56, 0.41 \rangle)$ | $((s_5, 0), \langle 0.60, 0.45 \rangle)$ | $((s_6, 0), \langle 0.69, 0.45 \rangle)$ | $((s_5, 0), \langle 0.30, 0.56 \rangle)$ | $((s_4, 0), \langle 0.70, 0.54 \rangle)$ |
| $x_2$ | $((s_5, 0), \langle 0.48, 0.48 \rangle)$ | $((s_4, 0), \langle 0.65, 0.40 \rangle)$ | $((s_4, 0), \langle 0.45, 0.32 \rangle)$ | $((s_2, 0), \langle 0.48, 0.69 \rangle)$ | $((s_5, 0), \langle 0.27, 0.38 \rangle)$ |
| $x_3$ | $((s_3, 0), \langle 0.55, 49 \rangle)$ | $((s_4, 0), \langle 0.72, 0.52 \rangle)$ | $((s_4, 0), \langle 0.42, 0.55 \rangle)$ | $((s_5, 0), \langle 0.25, 0.42 \rangle)$ | $((s_0, 0), \langle 0.35, 0.30 \rangle)$ |
| $x_4$ | $((s_2, 0), \langle 0.40, 0.40 \rangle)$ | $((s_1, 0), \langle 0.52, 0.52 \rangle)$ | $((s_0, 0), \langle 0.60, 0.44 \rangle)$ | $((s_2, 0), \langle 0.45, 0.62 \rangle)$ | $((s_3, 0), \langle 0.45, 0.40 \rangle)$ |

**Table 7. Normalized q-RFLDM $M^3$.**

|        | $c_1$ | $c_2$ | $c_3$ | $c_4$ | $c_5$ |
|--------|-------|-------|-------|-------|-------|
| $x_1$ | $((s_4, 0), \langle 0.54, 0.38 \rangle)$ | $((s_6, 0), \langle 0.58, 0.42 \rangle)$ | $((s_4, 0), \langle 0.65, 0.40 \rangle)$ | $((s_5, 0), \langle 0.30, 0.55 \rangle)$ | $((s_4, 0), \langle 0.68, 0.50 \rangle)$ |
| $x_2$ | $((s_3, 0), \langle 0.55, 0.55 \rangle)$ | $((s_6, 0), \langle 0.68, 0.42 \rangle)$ | $((s_2, 0), \langle 0.45, 0.35 \rangle)$ | $((s_4, 0), \langle 0.50, 0.70 \rangle)$ | $((s_5, 0), \langle 0.30, 0.40 \rangle)$ |
| $x_3$ | $((s_1, 0), \langle 0.52, 45 \rangle)$ | $((s_4, 0), \langle 0.70, 0.50 \rangle)$ | $((s_2, 0), \langle 0.38, 0.52 \rangle)$ | $((s_3, 0), \langle 0.20, 0.40 \rangle)$ | $((s_0, 0), \langle 0.40, 0.35 \rangle)$ |
| $x_4$ | $((s_2, 0), \langle 0.42, 0.42 \rangle)$ | $((s_3, 0), \langle 0.50, 0.50 \rangle)$ | $((s_0, 0), \langle 0.60, 0.45 \rangle)$ | $((s_2, 0), \langle 0.40, 0.35 \rangle)$ | $((s_3, 0), \langle 0.55, 0.25 \rangle)$ |

Step 3: At first, the values of $\bar{\partial}_{ij}$; $i = 1, 2, \ldots, 4$, $j = 1, 2, \ldots, 5$ are computed on the basis of Eq (40), and then, the weight of each $d_t$ ($t = 1, 2, 3$) is obtained via Eqs (41)–(44) as given below.

$\Omega^{(1)} = 0.3302, \Omega^{(2)} = 0.3624, \Omega^{(3)} = 0.3074.$

Step 4: Based on Eq (45), we integrate the decision matrices 5-7 by utilizing the weight vector $\Omega = (0.3302, 0.3624, 0.3074)^t$, $q = 2$. The derived global q-RFLDM $M = (\partial_{ij})_{4 \times 5}$ is depicted in Table 8.

**Table 8. Global q-RFLDM $M$.**

| | $c_1$ | $c_2$ | $c_3$ | $c_4$ | $c_5$ |
|---|---|---|---|---|---|
| $x_1$ | $((s_5, 0.2378), \langle 0.5507, 0.1010 \rangle)$ | $((s_5, -0.2856), \langle 0.5940, 0.03809 \rangle)$ | $((s_5, 0.2378), \langle 0.6749, 0.1410 \rangle)$ | $((s_5, 0.4091), \langle 0.3000, 0.1771 \rangle)$ | $((s_4, 0.0000), \langle 0.6940, 0.2286 \rangle)$ |
| $x_2$ | $((s_4, 0.1705), \langle 0.5096, 0.1691 \rangle)$ | $((s_5, 0.1280), \langle 0.6628, 0.1202 \rangle)$ | $((s_3, 0.1401), \langle 0.4467, 0.06422 \rangle)$ | $((s_3, 0.0302), \langle 0.4863, 0.3420 \rangle)$ | $((s_5, 0.4091), \langle 0.2828, 0.08127 \rangle)$ |
| $x_3$ | $((s_2, 0.1226), \langle 0.5377, 0.1469 \rangle)$ | $((s_4, 0.3759), \langle 0.7107, 0.2169 \rangle)$ | $((s_3, 0.1401), \langle 0.4016, 0.1772 \rangle)$ | $((s_4, 0.1705), \langle 0.2196, 0.08921 \rangle)$ | $((s_0, 0.0000), \langle 0.3760, 0.05945 \rangle)$ |
| $x_4$ | $((s_2, 0.3552), \langle 0.4096, 0.09715 \rangle)$ | $((s_2, 0.0126), \langle 0.5074, 0.1654 \rangle)$ | $((s_0, 0.0000), \langle 0.5935, 0.1330 \rangle)$ | $((s_1, -0.3103), \langle 0.4321, 0.1932 \rangle)$ | $((s_3, 0.0000), \langle 0.4844, 0.07687 \rangle)$ |

**Stage 2:** Criteria weight determination:

In the light of Eq (51), the criteria weights are obtained as follows:

$w_1 = 0.1482$, $w_2 = 0.2457$, $w_3 = 0.1817$, $w_4 = 0.1829$, $w_5 = 0.2415$.

**Stage 3:** Concordance index based on VIKOR:

Step 1: In the light of Eqs (46) and (47), the positive-ideal solution $\mathcal{I}^+$ and negative-ideal solution $\mathcal{I}^-$ are determined as given below:

$$\mathscr{I}^+ = \begin{pmatrix} ((s_5, 0.2378), \langle 0.5507, 0.1010 \rangle), ((s_6, -0.0.2856), \langle 0.5940, 0.03809 \rangle), \\ ((s_5, 0.2378), \langle 0.6749, 0.1410 \rangle), ((s_2, -0.3103), \langle 0.4321, 0.1932 \rangle), \\ ((s_5, 0.4091), \langle 0.2828, 0.08127 \rangle) \end{pmatrix}$$

$$\mathscr{I}^- = \begin{pmatrix} ((s_2, 0.1226), \langle 0.5377, 0.1469 \rangle), ((s_2, 0.0126), \langle 0.5074, 0.1654 \rangle), \\ ((s_0, 0.0000), \langle 0.5935, 0.1330 \rangle), ((s_5, 0.4091), \langle 0.3000, 0.1771 \rangle), \\ ((s_0, 0.0000), \langle 0.3760, 0.05945 \rangle) \end{pmatrix}.$$

Step 2: Using Eq (53), the Hamming distance between each q-RFLN and its accompanying positive ideal solution is computed as:

$d(\partial_{11}, \partial_1^+) = 0.0000$, $d(\partial_{12}, \partial_2^+) = 0.0000$, $d(\partial_{13}, \partial_3^+) = 0.0000$, $d(\partial_{14}, \partial_4^+) = 0.2703$, $d(\partial_{15}, \partial_5^+) = 0.006319$, $d(\partial_{21}, \partial_1^+) = 0.1171$, $d(\partial_{22}, \partial_2^+) = 0.02969$, $d(\partial_{23}, \partial_3^+) = 0.2690$, $d(\partial_{24}, \partial_4^+) = 0.1036$, $d(\partial_{25}, \partial_5^+) = 0.0000$, $d(\partial_{31}, \partial_1^+) = 0.2916$, $d(\partial_{32}, \partial_2^+) = 0.09586$, $d(\partial_{33}, \partial_3^+) = 0.2836$, $d(\partial_{34}, \partial_4^+) = 0.1711$, $d(\partial_{35}, \partial_5^+) = 0.4147$, $d(\partial_{41}, \partial_1^+) = 0.2889$, $d(\partial_{42}, \partial_2^+) = 0.3748$, $d(\partial_{43}, \partial_3^+) = 0.5371$, $d(\partial_{44}, \partial_4^+) = 0.0000$, $d(\partial_{45}, \partial_5^+) = 0.1514$.

Step 3: According to Eqs (54) and (55), group utility values $\mathscr{U}_i(i = 1, 2, 3, 4)$ and individual regret values $\mathscr{V}_i(i = 1, 2, 3, 4)$ are obtained:

$\mathscr{U}_1 = 0.1866$, $\mathscr{U}_2 = 0.2401$, $\mathscr{U}_3 = 0.6643$, $\mathscr{U}_4 = 0.6624$.

$\mathscr{V}_1 = 0.1829$, $\mathscr{V}_2 = 0.09100e$, $\mathscr{V}_3 = 0.2415$, $\mathscr{V}_4 = 0.2457$.

Step 4: Under Eq (56), we get the concordance index as:

$\mathscr{I}_{11} = 0.0000$, $\mathscr{I}_{12} = -0.2410$, $\mathscr{I}_{13} = 0.6894$, $\mathscr{I}_{14} = 0.7010$,

$\mathscr{I}_{21} = 0.2410$, $\mathscr{I}_{22} = 0.0000$, $\mathscr{I}_{23} = 0.9304$, $\mathscr{I}_{24} = 0.9420$,

$\mathscr{I}_{31} = -0.6894$, $\mathscr{I}_{32} = -0.9304$, $\mathscr{I}_{33} = 0.0000$, $\mathscr{I}_{34} = 0.01159$,

$\mathscr{I}_{41} = -0.7010$, $\mathscr{I}_{42} = -0.9420$, $\mathscr{I}_{43} = -0.01159$, $\mathscr{I}_{44} = 0.0000$.

**Stage 4:** Ranking result based on QUALIFLEX:

Step 1: All 24 possible permutations are itemized:

$\Re_1 = (4, 3, 2, 1)$, $\Re_2 = (4, 3, 1, 2)$, $\Re_3 = (4, 2, 3, 1)$, $\Re_4 = (4, 2, 1, 3)$, $\Re_5 = (4, 1, 3, 2)$, $\Re_6 = (4, 1, 2, 3)$, $\Re_7 = (3, 4, 2, 1)$, $\Re_8 = (3, 4, 1, 2)$, $\Re_9 = (3, 2, 4, 1)$, $\Re_{10} = (3, 2, 1, 4)$, $\Re_{11} = (3, 1, 4, 2)$, $\Re_{12} = (3, 1, 2, 4)$, $\Re_{13} = (2, 4, 3, 1)$, $\Re_{14} = (2, 4, 1, 3)$, $\Re_{15} = (2, 3, 4, 1)$, $\Re_{16} = (2, 3, 1, 4)$, $\Re_{17} = (2, 1, 4, 3)$, $\Re_{18} = (2, 1, 3, 4)$, $\Re_{19} = (1, 4, 3, 2)$, $\Re_{20} = (1, 4, 2, 3)$, $\Re_{21} = (1, 3, 4, 2)$, $\Re_{22} = (1, 3, 2, 4)$, $\Re_{23} = (1, 2, 4, 3)$, $\Re_{24} = (1, 2, 3, 4)$.

Step 2: The general concordance index of each permutation is computed with Eq (57), as given below:

$\mathscr{I}_1 = -3.033, \mathscr{I}_2 = -3.515, \mathscr{I}_3 = -1.173, \mathscr{I}_4 = 0.2062, \mathscr{I}_5 = -2.137,$
$\mathscr{I}_6 = -0.2758,$

$\mathscr{I}_7 = -3.010, \mathscr{I}_8 = -3.492, \mathscr{I}_9 = -1.126, \mathscr{I}_{10} = 0.2758, \mathscr{I}_{11} = -2.090,$
$\mathscr{I}_{12} = -0.2062,$

$\mathscr{I}_{13} = 0.7114, \mathscr{I}_{14} = 2.090, \mathscr{I}_{15} = 0.7346, \mathscr{I}_{16} = 2.137, \mathscr{I}_{17} = 3.492, \mathscr{I}_{18} = 3.515,$

$\mathscr{I}_{19} = -0.7346, \mathscr{I}_{20} = 1.126, \mathscr{I}_{21} = -0.7114, \mathscr{I}_{22} = 1.173, \mathscr{I}_{23} = 3.010, \mathscr{I}_{24} = 3.033.$

Step 3: The largest overall concordance index is $\Re_{18}$, thus the ranking result is $x_2 > x_1 > x_3 > x_4$.

## 6.2 Discussion under partial weight information

In this case, the weight values for the evaluation criteria are only partially available, and the supplied weight information is as follows:

$$R = \left\{ \begin{array}{c} 0.15 \leq w_1 \leq 0.2, \ 0.16 \leq w_2 \leq 0.18, \ 0.05 \leq w_3 \leq 0.15, \\ 0.25 \leq w_4 \leq 0.35, \ 0.3 \leq w_5 \leq 0.45 \end{array} \right\}.$$

The following linear programming model can then be created employing Eqs (49) and (52):

$$M_{II} \left\{ \begin{array}{l} \max D(w) = 1.608w_1 + 2.665w_2 + 1.971w_3 + 1.984w_4 + 2.620w_5 \\ \\ s.t. \ w \in R, \ w_j \geq 0, \ j = 1, 2, .., 5, \ \sum_{j=1}^{5} w_j = 1. \end{array} \right.$$

Solving the above model via Lingo software, we get the following weight vector of criteria:

$$w = (0.1500, 0.1800, 0.0500, 0.2500, 0.3700)^T.$$

Step 3: According to Eqs (54) and (55), group utility values $\mathscr{U}_i (i = 1, 2, 3, 4)$ and individual regret values $\mathscr{V}_i (i = 1, 2, 3, 4)$ are obtained:

$\mathscr{U}_1 = 0.2556, \mathscr{U}_2 = 0.1954, \mathscr{U}_3 = 0.7507, \mathscr{U}_4 = 0.5137.$

$\mathscr{V}_1 = 0.2500, \mathscr{V}_2 = 0.09582, \mathscr{V}_3 = 0.3700, \mathscr{V}_4 = 0.1800.$

Step 4: Under Eq (56), we get the concordance index as:

$\mathscr{I}_{11} = 0.0000, \mathscr{I}_{12} = -0.3354, \mathscr{I}_{13} = 0.6646, \mathscr{I}_{14} = 0.1047,$

$\mathscr{I}_{21} = 0.3354, \mathscr{I}_{22} = 0.0000, \mathscr{I}_{23} = 1.000, \mathscr{I}_{24} = 0.4401,$

$\mathscr{I}_{31} = -0.6646, \mathscr{I}_{32} = -1.000, \mathscr{I}_{33} = 0.0000, \mathscr{I}_{34} = -0.5599,$

$\mathscr{I}_{41} = -0.1047, \mathscr{I}_{42} = -0.4401, \mathscr{I}_{43} = 0.5599, \mathscr{I}_{44} = 0.0000.$

**Stage 4:** Ranking result based on QUALIFLEX:

Step 1: All 24 possible permutations are itemized:

$\Re_1 = (4, 3, 2, 1), \Re_2 = (4, 3, 1, 2), \Re_3 = (4, 2, 3, 1), \Re_4 = (4, 2, 1, 3), \Re_5 = (4, 1, 3, 2), \Re_6 = (4, 1, 2, 3), \Re_7 = (3, 4, 2, 1), \Re_8 = (3, 4, 1, 2), \Re_9 = (3, 2, 4, 1), \Re_{10} = (3, 2, 1, 4), \Re_{11} = (3, 1, 4, 2), \Re_{12} = (3, 1, 2, 4), \Re_{13} = (2, 4, 3, 1), \Re_{14} = (2, 4, 1, 3), \Re_{15} = (2, 3, 4, 1), \Re_{16} = (2, 3, 1, 4), \Re_{17} = (2, 1, 4, 3), \Re_{18} = (2, 1, 3, 4), \Re_{19} = (1, 4, 3, 2), \Re_{20} = (1, 4, 2, 3), \Re_{21} = (1, 3, 4, 2), \Re_{22} = (1, 3, 2, 4), \Re_{23} = (1, 2, 4, 3), \Re_{24} = (1, 2, 3, 4).$

Step 2: The general concordance index of each permutation is computed with Eq (57), as shown below:

$\mathscr{I}_1 = -1.314, \mathscr{I}_2 = -1.985, \mathscr{I}_3 = 0.6859, \mathscr{I}_4 = 2.015, \mathscr{I}_5 = -0.6557, \mathscr{I}_6 = 1.344,$

$\mathscr{I}_7 = -2.434, \mathscr{I}_8 = -3.105, \mathscr{I}_9 = -1.554, \mathscr{I}_{10} = -1.344, \mathscr{I}_{11} = -2.895,$
$\mathscr{I}_{12} = -2.015,$

$\mathscr{I}_{13} = 1.566, \mathscr{I}_{14} = 2.895, \mathscr{I}_{15} = 0.4463, \mathscr{I}_{16} = 0.6557, \mathscr{I}_{17} = 3.105, \mathscr{I}_{18} = 1.985,$
$\mathscr{I}_{19} = -0.4463, \mathscr{I}_{20} = 1.554, \mathscr{I}_{21} = -1.566, \mathscr{I}_{22} = -0.6859, \mathscr{I}_{23} = 2.434,$
$\mathscr{I}_{24} = 1.314.$

Step 3: The largest overall concordance index is $\Re_{17}$, thus the ranking result is $x_2 > x_1 > x_4 > x_3$.

## 6.3 Sensitivity analysis

**6.3.1 Impact of the parameter ¥ on results.** In the fourth stage of the research procedure, a parameter ¥ is added that combines group utility and individual regret (see Eq (56)). Generally, a consensus-based decision-making process is implemented, and ¥ = 0.5 is determined from an equilibrium standpoint. However, the value of ¥ may differ among DMKs or groups. The value of this parameter will increase when DMKs prioritize the maximization of group utility, and $0.5 \leq$ ¥ $\leq 1$. In contrast, a small ¥ will be assigned if DMKs has emphasized the minimization of individual regret, and $0 \leq$ ¥ $\leq 0.5$ The value of ¥ impacts the value of the concordance index, and the concordance index plays a crucial role in determining rank. Consequently, it is essential to analyze the influence of this parameter ¥ on ranking orders. Table 9 describes the results of the presented mechanism utilizing various ¥ values.

It is evident from Table 9 that most of the ranking results are consistent, i.e., for all values of ¥, the final ranking result is $x_2 > x_1 > x_3 > x_4$, except ¥ = 0.5, 0.9, 1.0. For ¥ = 0.5, the ranking result is $x_2 > x_1 > x_4 > x_3$. The alternatives $x_3$ and $x_4$ have switched places, but the best option remains the same, i.e., $x_2$. Next, the ranking outcome for ¥ = 0.9 and ¥ = 1.0 is $x_1 > x_2 > x_4 > x_3$. In this case, the positions of the alternatives $x_3$ and $x_4$ corresponded to the ranking result for ¥ = 0.5. In addition, the second-highest general concordance value for ¥ = 0.5 also suggests this ranking, as shown in Table 9.

In a nutshell, the ranking result is significantly less sensitive to the ¥ values when employing the integrated VIKOR and QUALIFLEX decision-making mechanism. The sensitivity analysis certified the suggested approach's resilience to a certain point.

**6.3.2 Weight sensitivity analysis.** Sensitivity analysis is used to determine how sensitive the ranking order of an MCGDM technique is to changes in the weights of criteria. To test the results with little variation in the weights and see the rank reversal of the developed hybrid method, sensitivity analysis is conducted first by adding 0.1 to each criterion and then subtracting 0.1 from each criterion individually and adjusting other criteria accordingly to Eq (58).

$$w_d = \frac{1 - \breve{w}_c}{1 - \breve{w}_o} w_o, \tag{58}$$

where $\breve{w}_o$ is the weight associated with the criteria and $\breve{w}_c$ is the criteria's exchanged weight, $w_o$ is the old weight and $w_d$ is the derived criteria weight.

The derived weight values for the diagnosed MCGDM method are displayed in Table 10. These weights are then used to assess the sensitivity of the outlined method. The ten cases of sensitivity analysis are listed in Table 11.

The general concordance indices recorded in Table 11 show minor alteration in the values with respect to criteria weight fluctuations. It can be seen from Table 11 that in each scenario except $s_5$ and $s_8$, the best alternative is $x_2$. According to $S_5$ and $S_8$, the best alternative is $x_1$. These results indicate that the framed approach is more sensitive with respect to criteria $c_3$ and $c_4$. Anyhow, the overall sensitivity results suggest that the proposed method is quite stable with respect to criteria weights.

**Table 9. Ranking results with various ¥.**

| ¥ | General concordance indices | Ranking |
|---|---|---|
| 0.0 | $\mathscr{I}_1 = -2.191, \mathscr{I}_2 = -3.379, \mathscr{I}_3 = -0.2448, \mathscr{I}_4 = 0.5128, \mathscr{I}_5 = -2.621, \mathscr{I}_6 = -0.6754,$ $\mathscr{I}_7 = -2.136, \mathscr{I}_8 = -3.325, \mathscr{I}_9 = -0.1364, \mathscr{I}_{10} = 0.6754, \mathscr{I}_{11} = -2.513,$ $\mathscr{I}_{12} = -0.5128,$ <br> $\mathscr{I}_{13} = 1.755, \mathscr{I}_{14} = 2.513, \mathscr{I}_{15} = 1.809, \mathscr{I}_{16} = 2.621, \mathscr{I}_{17} = 3.325, \mathscr{I}_{18} = 3.379,$ <br> $\mathscr{I}_{19} = -1.809, \mathscr{I}_{20} = 0.1364, \mathscr{I}_{21} = -1.755, \mathscr{I}_{22} = 0.2448, \mathscr{I}_{23} = 2.136, \mathscr{I}_{24} = 2.191.$ | $x_2 > x_1 > x_3 > x_4$ |
| 0.1 | $\mathscr{I}_1 = -2.359, \mathscr{I}_2 = -3.406, \mathscr{I}_3 = -0.4305, \mathscr{I}_4 = 0.4513, \mathscr{I}_5 = -2.524,$ $\mathscr{I}_6 = -0.5955,$ <br> $\mathscr{I}_7 = -2.311, \mathscr{I}_8 = -3.358, \mathscr{I}_9 = -0.3345, \mathscr{I}_{10} = 0.5955, \mathscr{I}_{11} = -2.428,$ $\mathscr{I}_{12} = -0.4513,$ <br> $\mathscr{I}_{13} = 1.546, \mathscr{I}_{14} = 2.428, \mathscr{I}_{15} = 1.594, \mathscr{I}_{16} = 2.524, \mathscr{I}_{17} = 3.358, \mathscr{I}_{18} = 3.406,$ <br> $\mathscr{I}_{19} = -1.594, \mathscr{I}_{20} = 0.3345, \mathscr{I}_{21} = -1.546, \mathscr{I}_{22} = 0.4305, \mathscr{I}_{23} = 2.311, \mathscr{I}_{24} = 2.359.$ | $x_2 > x_1 > x_3 > x_4$ |
| 0.2 | $\mathscr{I}_1 = -2.528, \mathscr{I}_2 = -3.433, \mathscr{I}_3 = -0.6160, \mathscr{I}_4 = 0.3900, \mathscr{I}_5 = -2.427,$ $\mathscr{I}_6 = -0.5156,$ <br> $\mathscr{I}_7 = -2.486, \mathscr{I}_8 = -3.392, \mathscr{I}_9 = -0.5324, \mathscr{I}_{10} = 0.5156, \mathscr{I}_{11} = -2.344,$ $\mathscr{I}_{12} = -0.3900,$ <br> $\mathscr{I}_{13} = 1.338, \mathscr{I}_{14} = 2.344, \mathscr{I}_{15} = 1.379, \mathscr{I}_{16} = 2.427, \mathscr{I}_{17} = 3.392, \mathscr{I}_{18} = 3.433,$ <br> $\mathscr{I}_{19} = -1.379, \mathscr{I}_{20} = 0.5324, \mathscr{I}_{21} = -1.338, \mathscr{I}_{22} = 0.6160, \mathscr{I}_{23} = 2.486, \mathscr{I}_{24} = 2.528.$ | $x_2 > x_1 > x_3 > x_4$ |
| 0.3 | $\mathscr{I}_1 = -2.696, \mathscr{I}_2 = -3.461, \mathscr{I}_3 = -0.8016, \mathscr{I}_4 = 0.3288, \mathscr{I}_5 = -2.330,$ $\mathscr{I}_6 = -0.4356,$ <br> $\mathscr{I}_7 = -2.661, \mathscr{I}_8 = -3.425, \mathscr{I}_9 = -0.7304, \mathscr{I}_{10} = 0.4356, \mathscr{I}_{11} = -2.259,$ $\mathscr{I}_{12} = -0.3288,$ <br> $\mathscr{I}_{13} = 1.129, \mathscr{I}_{14} = 2.259, \mathscr{I}_{15} = 1.164, \mathscr{I}_{16} = 2.330, \mathscr{I}_{17} = 3.425, \mathscr{I}_{18} = 3.461,$ <br> $\mathscr{I}_{19} = -1.164, \mathscr{I}_{20} = 0.7304, \mathscr{I}_{21} = -1.129, \mathscr{I}_{22} = 0.8016, \mathscr{I}_{23} = 2.661, \mathscr{I}_{24} = 2.696.$ | $x_2 > x_1 > x_3 > x_4$ |
| 0.4 | $\mathscr{I}_1 = -2.865, \mathscr{I}_2 = -3.488, \mathscr{I}_3 = -0.9871, \mathscr{I}_4 = 0.2675, \mathscr{I}_5 = -2.234,$ $\mathscr{I}_6 = -0.3557,$ <br> $\mathscr{I}_7 = -2.836, \mathscr{I}_8 = -3.459, \mathscr{I}_9 = -0.9283, \mathscr{I}_{10} = 0.3557, \mathscr{I}_{11} = -2.175,$ $\mathscr{I}_{12} = -0.2675,$ <br> $\mathscr{I}_{13} = 0.9201, \mathscr{I}_{14} = 2.175, \mathscr{I}_{15} = 0.9495, \mathscr{I}_{16} = 2.234, \mathscr{I}_{17} = 3.459, \mathscr{I}_{18} = 3.488,$ <br> $\mathscr{I}_{19} = -0.9495, \mathscr{I}_{20} = 0.9283, \mathscr{I}_{21} = -0.9201, \mathscr{I}_{22} = 0.9871, \mathscr{I}_{23} = 2.836,$ $\mathscr{I}_{24} = 2.865.$ | $x_2 > x_1 > x_3 > x_4$ |
| 0.4 | $\mathscr{I}_1 = -1.314, \mathscr{I}_2 = -1.985, \mathscr{I}_3 = 0.6859, \mathscr{I}_4 = 2.015, \mathscr{I}_5 = -0.6557, \mathscr{I}_6 = 1.344,$ <br> $\mathscr{I}_7 = -2.434, \mathscr{I}_8 = -3.105, \mathscr{I}_9 = -1.554, \mathscr{I}_{10} = -1.344, \mathscr{I}_{11} = -2.895,$ $\mathscr{I}_{12} = -2.015,$ <br> $\mathscr{I}_{13} = 1.566, \mathscr{I}_{14} = 2.895, \mathscr{I}_{15} = 0.4463, \mathscr{I}_{16} = 0.6557, \mathscr{I}_{17} = 3.105, \mathscr{I}_{18} = 1.985,$ <br> $\mathscr{I}_{19} = -0.4463, \mathscr{I}_{20} = 1.554, \mathscr{I}_{21} = -1.566, \mathscr{I}_{22} = -0.6859, \mathscr{I}_{23} = 2.434, \mathscr{I}_{24} = 1.314.$ | $x_2 > x_1 > x_3 > x_4$ |
| 0.5 | $\mathscr{I}_1 = -3.202, \mathscr{I}_2 = -3.543, \mathscr{I}_3 = -1.358, \mathscr{I}_4 = 0.1449, \mathscr{I}_5 = -2.040, \mathscr{I}_6 = -0.1959,$ <br> $\mathscr{I}_7 = -3.185, \mathscr{I}_8 = -3.526, \mathscr{I}_9 = -1.324, \mathscr{I}_{10} = 0.1959, \mathscr{I}_{11} = -2.006,$ $\mathscr{I}_{12} = -0.1449,$ <br> $\mathscr{I}_{13} = 0.5027, \mathscr{I}_{14} = 2.006, \mathscr{I}_{15} = 0.5197, \mathscr{I}_{16} = 2.040, \mathscr{I}_{17} = 3.526, \mathscr{I}_{18} = 3.543,$ <br> $\mathscr{I}_{19} = -0.5197, \mathscr{I}_{20} = 1.324, \mathscr{I}_{21} = -0.5027, \mathscr{I}_{22} = 1.358, \mathscr{I}_{23} = 3.185, \mathscr{I}_{24} = 3.202.$ | $x_2 > x_1 > x_3 > x_4$ |
| 0.6 | $\mathscr{I}_1 = -3.370, \mathscr{I}_2 = -3.570, \mathscr{I}_3 = -1.543, \mathscr{I}_4 = 0.08376, \mathscr{I}_5 = -1.943,$ $\mathscr{I}_6 = -0.1159,$ <br> $\mathscr{I}_7 = -3.360, \mathscr{I}_8 = -3.559, \mathscr{I}_9 = -1.522, \mathscr{I}_{10} = 0.1159, \mathscr{I}_{11} = -1.921,$ $\mathscr{I}_{12} = -0.08376,$ <br> $\mathscr{I}_{13} = 0.2942, \mathscr{I}_{14} = 1.921, \mathscr{I}_{15} = 0.3049, \mathscr{I}_{16} = 1.943, \mathscr{I}_{17} = 3.559, \mathscr{I}_{18} = 3.570,$ <br> $\mathscr{I}_{19} = -0.3049, \mathscr{I}_{20} = 1.522, \mathscr{I}_{21} = -0.2942, \mathscr{I}_{22} = 1.543, \mathscr{I}_{23} = 3.360, \mathscr{I}_{24} = 3.370.$ | $x_2 > x_1 > x_3 > x_4$ |

(*Continued*)

**Table 9.** (Continued)

| ¥ | General concordance indices | Ranking |
|---|---|---|
| 0.8 | $\mathscr{I}_1 = -3.539$, $\mathscr{I}_2 = -3.597$, $\mathscr{I}_3 = -1.729$, $\mathscr{I}_4 = 0.02256$, $\mathscr{I}_5 = -1.846$, $\mathscr{I}_6 = -0.03586$, | $x_2 > x_1 > x_3 > x_4$ |
|  | $\mathscr{I}_7 = -3.535$, $\mathscr{I}_8 = -3.593$, $\mathscr{I}_9 = -1.720$, $\mathscr{I}_{10} = 0.03586$, $\mathscr{I}_{11} = -1.837$, $\mathscr{I}_{12} = -0.02256$, |  |
|  | $\mathscr{I}_{13} = 0.08536$, $\mathscr{I}_{14} = 1.837$, $\mathscr{I}_{15} = 0.08986$, $\mathscr{I}_{16} = 1.846$, $\mathscr{I}_{17} = 3.593$, $\mathscr{I}_{18} = 3.597$, |  |
|  | $\mathscr{I}_{19} = -0.08986$, $\mathscr{I}_{20} = 1.720$, $\mathscr{I}_{21} = -0.08536$, $\mathscr{I}_{22} = 1.729$, $\mathscr{I}_{23} = 3.535$, $\mathscr{I}_{24} = 3.539$. |  |
| 0.9 | $\mathscr{I}_1 = -3.708$, $\mathscr{I}_2 = -3.625$, $\mathscr{I}_3 = -1.915$, $\mathscr{I}_4 = -0.03873$, $\mathscr{I}_5 = -1.749$, $\mathscr{I}_6 = 0.04405$, | $x_1 > x_2 > x_4 > x_3$ |
|  | $\mathscr{I}_7 = -3.709$, $\mathscr{I}_8 = -3.626$, $\mathscr{I}_9 = -1.918$, $\mathscr{I}_{10} = -0.04405$, $\mathscr{I}_{11} = -1.752$, $\mathscr{I}_{12} = 0.03873$, |  |
|  | $\mathscr{I}_{13} = -0.1233$, $\mathscr{I}_{14} = 1.752$, $\mathscr{I}_{15} = -0.1251$, $\mathscr{I}_{16} = 1.749$, $\mathscr{I}_{17} = 3.626$, $\mathscr{I}_{18} = 3.625$, |  |
|  | $\mathscr{I}_{19} = 0.1251$, $\mathscr{I}_{20} = 1.918$, $\mathscr{I}_{21} = 0.1233$, $\mathscr{I}_{22} = 1.915$, $\mathscr{I}_{23} = 3.709$, $\mathscr{I}_{24} = 3.708$. |  |
| 1.0 | $\mathscr{I}_1 = -3.876$, $\mathscr{I}_2 = -3.652$, $\mathscr{I}_3 = -2.100$, $\mathscr{I}_4 = -0.1000$, $\mathscr{I}_5 = -1.652$, $\mathscr{I}_6 = 0.1240$, | $x_2 > x_1 > x_3 > x_4$ |
|  | $\mathscr{I}_7 = -3.884$, $\mathscr{I}_8 = -3.660$, $\mathscr{I}_9 = -2.116$, $\mathscr{I}_{10} = -0.2320$, $\mathscr{I}_{11} = -1.668$, $\mathscr{I}_{12} = 0.1000$, |  |
|  | $\mathscr{I}_{13} = -0.3320$, $\mathscr{I}_{14} = 1.668$, $\mathscr{I}_{15} = -0.3400$, $\mathscr{I}_{16} = 1.652$, $\mathscr{I}_{17} = 3.660$, $\mathscr{I}_{18} = 3.652$, |  |
|  | $\mathscr{I}_{19} = 0.3400$, $\mathscr{I}_{20} = 2.116$, $\mathscr{I}_{21} = 0.3320$, $\mathscr{I}_{22} = 2.100$, $\mathscr{I}_{23} = 3.884$, $\mathscr{I}_{24} = 3.876$. |  |

**Table 10. Criteria weights in ten scenarios.**

| Scenario | $c_1$ | $c_2$ | $c_3$ | $c_4$ | $c_5$ |
|---|---|---|---|---|---|
| $S_1(c_1 + 0.1)$ | 0.2482 | 0.2169 | 0.1604 | 0.1614 | 0.2131 |
| $S_2(c_1 - 0.1)$ | 0.0482 | 0.2745 | 0.2030 | 0.2044 | 0.2699 |
| $S_3(c_2 + 0.1)$ | 0.1286 | 0.3457 | 0.1576 | 0.1587 | 0.2094 |
| $S_4(c_2 - 0.1)$ | 0.1678 | 0.1457 | 0.2058 | 0.2071 | 0.2736 |
| $S_5(c_3 + 0.1)$ | 0.1301 | 0.2157 | 0.2817 | 0.1605 | 0.2120 |
| $S_6(c_3 - 0.1)$ | 0.1663 | 0.2757 | 0.0817 | 0.2053 | 0.2710 |
| $S_7(c_4 + 0.1)$ | 0.1301 | 0.2156 | 0.1595 | 0.2829 | 0.2119 |
| $S_8(c_4 - 0.1)$ | 0.1663 | 0.2758 | 0.2039 | 0.0829 | 0.2711 |
| $S_9(c_5 + 0.1)$ | 0.1287 | 0.2133 | 0.1577 | 0.1588 | 0.3415 |
| $S_{10}(c_5 - 0.1)$ | 0.1677 | 0.2781 | 0.2057 | 0.2070 | 0.1415 |

## 7 Comparative analysis and discussion

In this part, we compare the suggested q-RFL VIKOR-QUALIFLEX technique to existing methods to clearly show the rationale and efficacy of the technique presented in the present paper. The established technique is compared with the prevailing methodologies, including q-RFL PROMETHEE II method developed by Li et al. [62], q-RFL aggregation-based method introduced by Ju et al. [39], and q-rung orthopair fuzzy VIKOR method presented by Cheng et al. [73] in order to evaluate its reliability and practicality as well as its positive consequences. We thoroughly calculate the decision outcomes for the optimal recycling service supplier with these techniques and demonstrate the validity of the deployed methodology.

In order to facilitate a more accurate comparison, we will use the same weight vectors as determined in the deployed strategy while employing the prior methods.

**Table 11. Ranking results of ten scenarios.**

| Scenario | General concordance indices | Ranking |
|---|---|---|
| $S_1(c_1 + 0.1)$ | $\mathscr{I}_1 = -3.258, \mathscr{I}_2 = -3.499, \mathscr{I}_3 = -1.434, \mathscr{I}_4 = 0.1508, \mathscr{I}_5 = -1.914,$ $\mathscr{I}_6 = -0.08955,$ | $x_2 > x_1 > x_4 > x_3$ |
| | $\mathscr{I}_7 = -3.279, \mathscr{I}_8 = -3.519, \mathscr{I}_9 = -1.474, \mathscr{I}_{10} = 0.08955, \mathscr{I}_{11} = -1.955,$ $\mathscr{I}_{12} = -0.1508,$ | |
| | $\mathscr{I}_{13} = 0.3708, \mathscr{I}_{14} = 1.955, \mathscr{I}_{15} = 0.3504, \mathscr{I}_{16} = 1.914, \mathscr{I}_{17} = 3.519,$ $\mathscr{I}_{18} = 3.499,$ | |
| | $\mathscr{I}_{19} = -0.3504, \mathscr{I}_{20} = 1.474, \mathscr{I}_{21} = -0.3708, \mathscr{I}_{22} = 1.434, \mathscr{I}_{23} = 3.279,$ $\mathscr{I}_{24} = 3.258,$ | |
| $S_2(c_1 - 0.1)$ | $\mathscr{I}_1 = -3.111, \mathscr{I}_2 = -3.675, \mathscr{I}_3 = -1.168, \mathscr{I}_4 = 0.2112, \mathscr{I}_5 = -2.296,$ $\mathscr{I}_6 = -0.3528,$ | $x_2 > x_1 > x_3 > x_4$ |
| | $\mathscr{I}_7 = -3.052, \mathscr{I}_8 = -3.616, \mathscr{I}_9 = -1.085, \mathscr{I}_{10} = 0.3179, \mathscr{I}_{11} = -2.213,$ $\mathscr{I}_{12} = -0.2461,$ | |
| | $\mathscr{I}_{13} = 0.7992, \mathscr{I}_{14} = 2.178, \mathscr{I}_{15} = 0.8581, \mathscr{I}_{16} = 2.261, \mathscr{I}_{17} = 3.581,$ $\mathscr{I}_{18} = 3.640,$ | |
| | $\mathscr{I}_{19} = -0.8930, \mathscr{I}_{20} = 1.050, \mathscr{I}_{21} = -0.8341, \mathscr{I}_{22} = 1.133, \mathscr{I}_{23} = 3.017,$ $\mathscr{I}_{24} = 3.076,$ | |
| $S_3(c_2 + 0.1)$ | $\mathscr{I}_1 = -3.155, \mathscr{I}_2 = -3.349, \mathscr{I}_3 = -1.948, \mathscr{I}_4 = -0.9362, \mathscr{I}_5 = -2.337,$ $\mathscr{I}_6 = -1.131,$ | $x_2 > x_1 > x_3 > x_4$ |
| | $\mathscr{I}_7 = -2.466, \mathscr{I}_8 = -2.660, \mathscr{I}_9 = -0.5702, \mathscr{I}_{10} = 1.131, \mathscr{I}_{11} = -0.9594,$ $\mathscr{I}_{12} = 0.9362,$ | |
| | $\mathscr{I}_{13} = -0.05256, \mathscr{I}_{14} = 0.9594, \mathscr{I}_{15} = 0.6364, \mathscr{I}_{16} = 2.337, \mathscr{I}_{17} = 2.660,$ $\mathscr{I}_{18} = 3.349,$ | |
| | $\mathscr{I}_{19} = -0.6364, \mathscr{I}_{20} = 0.5702, \mathscr{I}_{21} = 0.05256, \mathscr{I}_{22} = 1.948, \mathscr{I}_{23} = 2.466,$ $\mathscr{I}_{24} = 3.155,$ | |
| $S_4(c_2 - 0.1)$ | $\mathscr{I}_1 = -2.141, \mathscr{I}_2 = -2.652, \mathscr{I}_3 = -0.2395, \mathscr{I}_4 = 1.150, \mathscr{I}_5 = -1.262,$ $\mathscr{I}_6 = 0.6393,$ | $x_2 > x_1 > x_4 > x_3$ |
| | $\mathscr{I}_7 = -2.737, \mathscr{I}_8 = -3.248, \mathscr{I}_9 = -1.433, \mathscr{I}_{10} = -0.6393, \mathscr{I}_{11} = -2.455,$ $\mathscr{I}_{12} = -1.150,$ | |
| | $\mathscr{I}_{13} = 1.065, \mathscr{I}_{14} = 2.455, \mathscr{I}_{15} = 0.4685, \mathscr{I}_{16} = 1.262, \mathscr{I}_{17} = 3.248,$ $\mathscr{I}_{18} = 2.652,$ | |
| | $\mathscr{I}_{19} = -0.4685, \mathscr{I}_{20} = 1.433, \mathscr{I}_{21} = -1.065, \mathscr{I}_{22} = 0.2395, \mathscr{I}_{23} = 2.737,$ $\mathscr{I}_{24} = 2.141,$ | |
| $S_5(c_3 + 0.1)$ | $\mathscr{I}_1 = -3.393, \mathscr{I}_2 = -3.330, \mathscr{I}_3 = -2.193, \mathscr{I}_4 = -0.9307, \mathscr{I}_5 = -2.068,$ $\mathscr{I}_6 = -0.8681,$ | $x_1 > x_2 > x_3 > x_4$ |
| | $\mathscr{I}_7 = -2.793, \mathscr{I}_8 = -2.731, \mathscr{I}_9 = -0.9939, \mathscr{I}_{10} = 0.8681, \mathscr{I}_{11} = -0.8687,$ $\mathscr{I}_{12} = 0.9307,$ | |
| | $\mathscr{I}_{13} = -0.3937, \mathscr{I}_{14} = 0.8687, \mathscr{I}_{15} = 0.2059, \mathscr{I}_{16} = 2.068, \mathscr{I}_{17} = 2.731,$ $\mathscr{I}_{18} = 3.330,$ | |
| | $\mathscr{I}_{19} = -0.2059, \mathscr{I}_{20} = 0.9939, \mathscr{I}_{21} = 0.3937, \mathscr{I}_{22} = 2.193, \mathscr{I}_{23} = 2.793,$ $\mathscr{I}_{24} = 3.393,$ | |
| $S_6(c_3 - 0.1)$ | $\mathscr{I}_1 = -2.830, \mathscr{I}_2 = -3.476, \mathscr{I}_3 = -0.8542, \mathscr{I}_4 = 0.4768, \mathscr{I}_5 = -2.145,$ $\mathscr{I}_6 = -0.1684,$ | $x_2 > x_1 > x_4 > x_3$ |
| | $\mathscr{I}_7 = -2.933, \mathscr{I}_8 = -3.578, \mathscr{I}_9 = -1.060, \mathscr{I}_{10} = 0.1684, \mathscr{I}_{11} = -2.350,$ $\mathscr{I}_{12} = -0.4768,$ | |
| | $\mathscr{I}_{13} = 1.019, \mathscr{I}_{14} = 2.350, \mathscr{I}_{15} = 0.9164, \mathscr{I}_{16} = 2.145, \mathscr{I}_{17} = 3.578,$ $\mathscr{I}_{18} = 3.476,$ | |
| | $\mathscr{I}_{19} = -0.9164, \mathscr{I}_{20} = 1.060, \mathscr{I}_{21} = -1.019, \mathscr{I}_{22} = 0.8542, \mathscr{I}_{23} = 2.933,$ $\mathscr{I}_{24} = 2.830,$ | |

(*Continued*)

**Table 11.** (Continued)

| Scenario | General concordance indices | Ranking |
|---|---|---|
| $S_7(c_4 + 0.1)$ | $\mathscr{I}_1 = -1.317, \mathscr{I}_2 = -2.388, \mathscr{I}_3 = 0.2758, \mathscr{I}_4 = 0.7982, \mathscr{I}_5 = -1.866,$ $\mathscr{I}_6 = -0.2726,$ | $x_2 > x_1 > x_4 >$ $x_3$ |
| | $\mathscr{I}_7 = -1.493, \mathscr{I}_8 = -2.563, \mathscr{I}_9 = -0.07456, \mathscr{I}_{10} = 0.2726, \mathscr{I}_{11} = -2.216,$ $\mathscr{I}_{12} = -0.7982,$ | |
| | $\mathscr{I}_{13} = 1.694, \mathscr{I}_{14} = 2.216, \mathscr{I}_{15} = 1.519, \mathscr{I}_{16} = 1.866, \mathscr{I}_{17} = 2.563, \mathscr{I}_{18} = 2.388,$ | |
| | $\mathscr{I}_{19} = -1.519, \mathscr{I}_{20} = 0.07456, \mathscr{I}_{21} = -1.694, \mathscr{I}_{22} = -0.2758, \mathscr{I}_{23} = 1.493,$ $\mathscr{I}_{24} = 1.317,$ | |
| $S_8(c_4 - 0.1)$ | $\mathscr{I}_1 = -3.777, \mathscr{I}_2 = -3.472, \mathscr{I}_3 = -2.223, \mathscr{I}_4 = -0.3618, \mathscr{I}_5 = -1.611,$ $\mathscr{I}_6 = -0.05601,$ | $x_1 > x_2 > x_3 >$ $x_4$ |
| | $\mathscr{I}_7 = -3.638, \mathscr{I}_8 = -3.332, \mathscr{I}_9 = -1.944, \mathscr{I}_{10} = 0.05601, \mathscr{I}_{11} = -1.332,$ $\mathscr{I}_{12} = 0.3618,$ | |
| | $\mathscr{I}_{13} = -0.5284, \mathscr{I}_{14} = 1.332, \mathscr{I}_{15} = -0.3892, \mathscr{I}_{16} = 1.611, \mathscr{I}_{17} = 3.332,$ $\mathscr{I}_{18} = 3.472,$ | |
| | $\mathscr{I}_{19} = 0.3892, \mathscr{I}_{20} = 1.944, \mathscr{I}_{21} = 0.5284, \mathscr{I}_{22} = 2.223, \mathscr{I}_{23} = 3.638,$ $\mathscr{I}_{24} = 3.777,$ | |
| $S_9(c_5 + 0.1)$ | $\mathscr{I}_1 = -2.536, \mathscr{I}_2 = -2.758, \mathscr{I}_3 = -0.6172, \mathscr{I}_4 = 1.079, \mathscr{I}_5 = -1.062,$ $\mathscr{I}_6 = 0.8562,$ | $x_2 > x_1 > x_4 >$ $x_3$ |
| | $\mathscr{I}_7 = -3.181, \mathscr{I}_8 = -3.403, \mathscr{I}_9 = -1.907, \mathscr{I}_{10} = -0.8562, \mathscr{I}_{11} = -2.352,$ $\mathscr{I}_{12} = -1.079,$ | |
| | $\mathscr{I}_{13} = 0.6562, \mathscr{I}_{14} = 2.352, \mathscr{I}_{15} = 0.0112, \mathscr{I}_{16} = 1.062, \mathscr{I}_{17} = 3.403,$ $\mathscr{I}_{18} = 2.758,$ | |
| | $\mathscr{I}_{19} = -0.0112, \mathscr{I}_{20} = 1.907, \mathscr{I}_{21} = -0.6562, \mathscr{I}_{22} = 0.6172, \mathscr{I}_{23} = 3.181,$ $\mathscr{I}_{24} = 2.536,$ | |
| $S_{10}(c_5 - 0.1)$ | $\mathscr{I}_1 = -2.647, \mathscr{I}_2 = -3.114, \mathscr{I}_3 = -1.571, \mathscr{I}_4 = -0.9611, \mathscr{I}_5 = -2.505,$ $\mathscr{I}_6 = -1.428,$ | $x_2 > x_1 > x_3 >$ $x_4$ |
| | $\mathscr{I}_7 = -1.851, \mathscr{I}_8 = -2.318, \mathscr{I}_9 = 0.0221, \mathscr{I}_{10} = 1.428, \mathscr{I}_{11} = -0.9119,$ $\mathscr{I}_{12} = 0.9611,$ | |
| | $\mathscr{I}_{13} = 0.3023, \mathscr{I}_{14} = 0.9119, \mathscr{I}_{15} = 1.099, \mathscr{I}_{16} = 2.505, \mathscr{I}_{17} = 2.318,$ $\mathscr{I}_{18} = 3.114,$ | |
| | $\mathscr{I}_{19} = -1.099, \mathscr{I}_{20} = -0.0221, \mathscr{I}_{21} = -0.3023, \mathscr{I}_{22} = 1.571, \mathscr{I}_{23} = 1.851,$ $\mathscr{I}_{24} = 2.647,$ | |

## 7.1 q-RFL PROMETHEE II method

In this part, the devised methodology is compared to q-RFL PROMETHEE II method developed by Li et al. [62] to assess its rationality.

Utilizing the q-ROPLWA operator Eq (8), the individual decision matrices are integrated into group decision matrix as shown in Table 12.

In the light of Eq (10), the deviations between two evaluation values are determined in Table 13.

The global preference of each option is assessed as follows:

**Table 12. Group q-RFLDM $M$.**

| | $c_1$ | $c_2$ | $c_3$ | $c_4$ | $c_5$ |
|---|---|---|---|---|---|
| $x_1$ | $((s_5, 0.0550), \langle 0.5507, 0.3973 \rangle)$ | $((s_6, -0.3624), \langle 0.5940, 0.4406 \rangle)$ | $((s_5, 0.0550), \langle 0.6749, 0.4308 \rangle)$ | $((s_5, 0.3302), \langle 0.3000, 0.5536 \rangle)$ | $((s_4, 0.0000), \langle 0.6940, 0.5241 \rangle)$ |
| $x_2$ | $((s_4, 0.0550), \langle 0.5096, 0.5073 \rangle)$ | $((s_5, -0.0550), \langle 0.6628, 0.4060 \rangle)$ | $((s_3, 0.0550), \langle 0.4467, 0.3323 \rangle)$ | $((s_3, -0.0550), \langle 0.4863, 0.6931 \rangle)$ | $((s_5, 0.3302), \langle 0.2828, 0.3860 \rangle)$ |
| $x_3$ | $((s_2, 0.0550), \langle 0.5377, 0.4741 \rangle)$ | $((s_4, 0.3302), \langle 0.7107, 0.5072 \rangle)$ | $((s_3, 0.0550), \langle 0.4016, 0.5406 \rangle)$ | $((s_4, 0.0550), \langle 0.2196, 0.4071 \rangle)$ | $((s_0, 0.0000), \langle 0.3760, 0.3246 \rangle)$ |
| $x_4$ | $((s_2, 0.3302), \langle 0.4096, 0.4094 \rangle)$ | $((s_2, -0.0550), \langle 0.5074, 0.5072 \rangle)$ | $((s_0, 0.0000), \langle 0.5935, 0.4431 \rangle)$ | $((s_2, -0.3302), \langle 0.4321, 0.5145 \rangle)$ | $((s_3, 0.0000), \langle 0.4844, 0.3462 \rangle)$ |

**Table 13. Deviations of pairwise evaluations.**

| | $c_1$ | $c_2$ | $c_3$ | $c_4$ | $c_5$ |
|---|---|---|---|---|---|
| $d_j(x_1, x_2)$ | 0.1233 | 0.01644 | 0.2209 | 0.1393 | 0.009607 |
| $d_j(x_1, x_3)$ | 0.2574 | 0.08064 | 0.2689 | 0.04270 | 0.3449 |
| $d_j(x_1, x_4)$ | 0.2471 | 0.3276 | 0.4585 | 0.1884 | 0.1059 |
| $d_j(x_2, x_1)$ | 0.04144 | 0.01644 | 0.2209 | 0.1393 | 0.009607 |
| $d_j(x_2, x_3)$ | 0.1341 | 0.06420 | 0.04803 | 0.09657 | 0.3544 |
| $d_j(x_2, x_4)$ | 0.1239 | 0.3112 | 0.2376 | 0.04909 | 0.1156 |
| $d_j(x_3, x_1)$ | 0.2574 | 0.08064 | 0.2689 | 0.04270 | 0.3449 |
| $d_j(x_3, x_2)$ | 0.1341 | 0.06420 | 0.04803 | 0.09657 | 0.3544 |
| $d_j(x_3, x_4)$ | 0.01024 | 0.2470 | 0.1896 | 0.1456 | 0.2389 |
| $d_j(x_4, x_1)$ | 0.2471 | 0.3276 | 0.4585 | 0.1884 | 0.1059 |
| $d_j(x_4, x_2)$ | 0.1239 | 0.3112 | 0.1591 | 0.1884 | 0.1156 |
| $d_j(x_4, x_3)$ | 0.01024 | 0.2470 | 0.1896 | 0.1456 | 0.2389 |

$$\ddagger\left(x_i, x_j\right) = \begin{pmatrix} 0 & 0.09025 & 0.1979 & 0.2605 \\ 0.07812 & 0 & 0.1476 & 0.1749 \\ 0.1979 & 0.1476 & 0 & 0.1810 \\ 0.2605 & 0.1861 & 0.1810 & 0 \end{pmatrix}.$$

The positive flow $\Phi^+(x_i)$ and negative flow $\Phi^-(x_i)$ of each option are computed:

$\Phi^+(x_1) = 0.1829$, $\Phi^+(x_2) = 0.1335$, $\Phi^+(x_3) = 0.1755$, $\Phi^+(x_4) = 0.2092$.

$\Phi^-(x_1) = 0.1788$, $\Phi^-(x_2) = 0.1413$, $\Phi^-(x_3) = 0.1755$, $\Phi^-(x_4) = 0.2055$.

Based on the values of $\Phi^+(x_i)$ and $\Phi^-(x_i)$, the net flows $\Phi(x_i)$ of each option are derived as:

$\Phi(x_1) = 0.0041$, $\Phi(x_2) = -0.0078$, $\Phi(x_3) = 0.0000$, $\Phi(x_4) = 0.0037$.

Thus, the ranking with q-RFL PROMETHEE II method [62] is as shown:

$x_1 > x_4 > x_3 > x_2$.

It is evident from the results that the rankings of alternatives for the two approaches are distinct. The primary reason for these differences is that the existing approach [62] is based on extant operational law-based operators [29], which suffer from several flaws pointed out in Section 3.

## 7.2 q-RFL aggregation-based method

This section is dedicated to derive the outcomes through q-RFL aggregation-based method [39]. The details of the steps are listed as:

Utilizing the q-RFL Muirhead mean (MM) operator, the individual decision matrices Tables 5–7 are integrated into group decision matrix, which is listed in Table 8.

To aggregate q-RFL assessment values $\partial_{ij}$ of alternative $x_i$ on all criteria $c_j (j = 1, 2, 3, 4, 5)$ into the overall assessment value $\partial_i$ of the alternative $x_i (i = 1, 2, 3, 4)$ the q-RFLWA operator Eq (12) is employed and the overall assessment values of alternatives $x_i (i = 1, 2, 3, 4)$ are obtained as:

$\partial_1 = ((s_5, -0.0063), \langle 0.6003, 0.4698 \rangle)$, $\partial_2 = ((s_4, 0.1969), \langle 0.5066, 0.4408 \rangle)$,

$\partial_3 = ((s_3, -0.3348), \langle 0.5100, 0.4381 \rangle)$, $\partial_4 = ((s_2, -0.1469), \langle 0.4954, 0.4383 \rangle)$.

According to Eq (6), the score values of overall assessment value $\partial_i (i = 1, 2, 3, 4)$ are determined as given below:

$S(\partial_1) = 0.4065$, $S(\partial_2) = 0.3185$, $S(\partial_3) = 0.2033$, $S(\partial_4) = 0.1394$,

Based on the above score values, the ranking of all suppliers is obtained:

$x_1 > x_2 > x_3 > x_4$.

From the above result, we can notice a slight difference in the ranking of the two methods, i.e., the alternatives $x_1$ and $x_2$ have swapped their positions while the remaining alternatives have the same rank. However, in the existent method [39], weight information is assumed to be known beforehand, and aggregation is performed by irrational AOs.

### 7.3 q-rung orthopair fuzzy VIKOR method

Further, the findings of the established approach are validated by comparing them to the q-rung orthopair fuzzy VIKOR method created by Cheng et al. [73]. To make the Cheng et al. approach compatible with the problem under consideration, the linguistic words and their corresponding symbolic translations from the provided data given in Tables 2–4 are removed. We acquire the following calculated results.

Utilizing the q-rung orthopair fuzzy weighted averaging (q-ROPFWA) operator, the individual q-rung orthopair fuzzy decision matrices are integrated into group decision matrix, which is shown in Table 14.

Next, the best and worst ratings of the alternatives are determined as provided below:

$\mathscr{H}^+ = \{\langle 0.5507, 0.3973\rangle, \langle 0.7107, 0.4060\rangle, \langle 0.6749, 0.3323\rangle, \langle 0.2196, 0.6931\rangle,$
$\langle 0.6940, 0.3246\rangle\}$,

$\mathscr{H}^- = \{\langle 0.4096, 0.5073\rangle, \langle 0.5074, 0.5072\rangle, \langle 0.4016, 0.5406\rangle, \langle 0.4863, 0.4071\rangle,$
$\langle 0.2828, 0.5241\rangle\}$.

Based on the best and worst ratings, individual regret, group utility, and compromise degree are determined as follows:

$r_1 = 0.38712, r_2 = 0.79705, r_3 = 0.8038, r_4 = 1.1697$.

According to the above individual regret values, we have $x_1 > x_2 > x_3 > x_4$.

$g_1 = 0.1315, g_2 = 0.2132, g_3 = 0.1817, g_4 = 0.3125$.

According to the above group utility values, we have $x_1 > x_3 > x_2 > x_4$.

$c_1 = 0.0000, c_2 = 0.4876, c_3 = 0.4049, c_4 = 1.000$.

According to the derived compromise degrees, we have $x_1 > x_3 > x_2 > x_4$.

One can easily check that both the conditions of the VIKOR are satisfied, so the compromise solution consists of just $x_1$.

From Table 15 and Fig 1, it can be seen that all the ranking results of the two approaches are different except $x_4$. The difference is due to the absence of linguistic terms in the approach

**Table 14. q-rung orthopair fuzzy group decision matrix.**

|  | $c_1$ | $c_2$ | $c_3$ | $c_4$ | $c_5$ |
|---|---|---|---|---|---|
| $x_1$ | $\langle 0.5507, 0.3973\rangle$ | $\langle 0.5940, 0.4406\rangle$ | $\langle 0.6749, 0.4308\rangle$ | $\langle 0.3000, 0.5536\rangle$ | $\langle 0.6940, 0.5241\rangle$ |
| $x_2$ | $\langle 0.5096, 0.5073\rangle$ | $\langle 0.6628, 0.4060\rangle$ | $\langle 0.4467, 0.3323\rangle$ | $\langle 0.4863, 0.6931\rangle$ | $\langle 0.2828, 0.3860\rangle$ |
| $x_3$ | $\langle 0.5377, 0.4741\rangle$ | $\langle 0.7107, 0.5072\rangle$ | $\langle 0.4016, 0.5406\rangle$ | $\langle 0.2196, 0.4071\rangle$ | $\langle 0.3760, 0.3246\rangle$ |
| $x_4$ | $\langle 0.4096, 0.4094\rangle$ | $\langle 0.5074, 0.5072\rangle$ | $\langle 0.5935, 0.4431\rangle$ | $\langle 0.4321, 0.5145\rangle$ | $\langle 0.4844, 0.3462\rangle$ |

**Table 15. Decision results obtained by using different methods.**

| Method | Ranking results |
|---|---|
| q-RFL PROMETHEE II method ($M_1$) [62] | $x_1 > x_4 > x_3 > x_2$ |
| q-RFL aggregation-based method ($M_2$) [39] | $x_1 > x_2 > x_3 > x_4$ |
| q-rung orthopair fuzzy VIKOR method ($M_3$) [73] | $x_1 > x_3 > x_2 > x_4$ |
| Proposed method ($M_4$) | $x_2 > x_1 > x_3 > x_4$ |

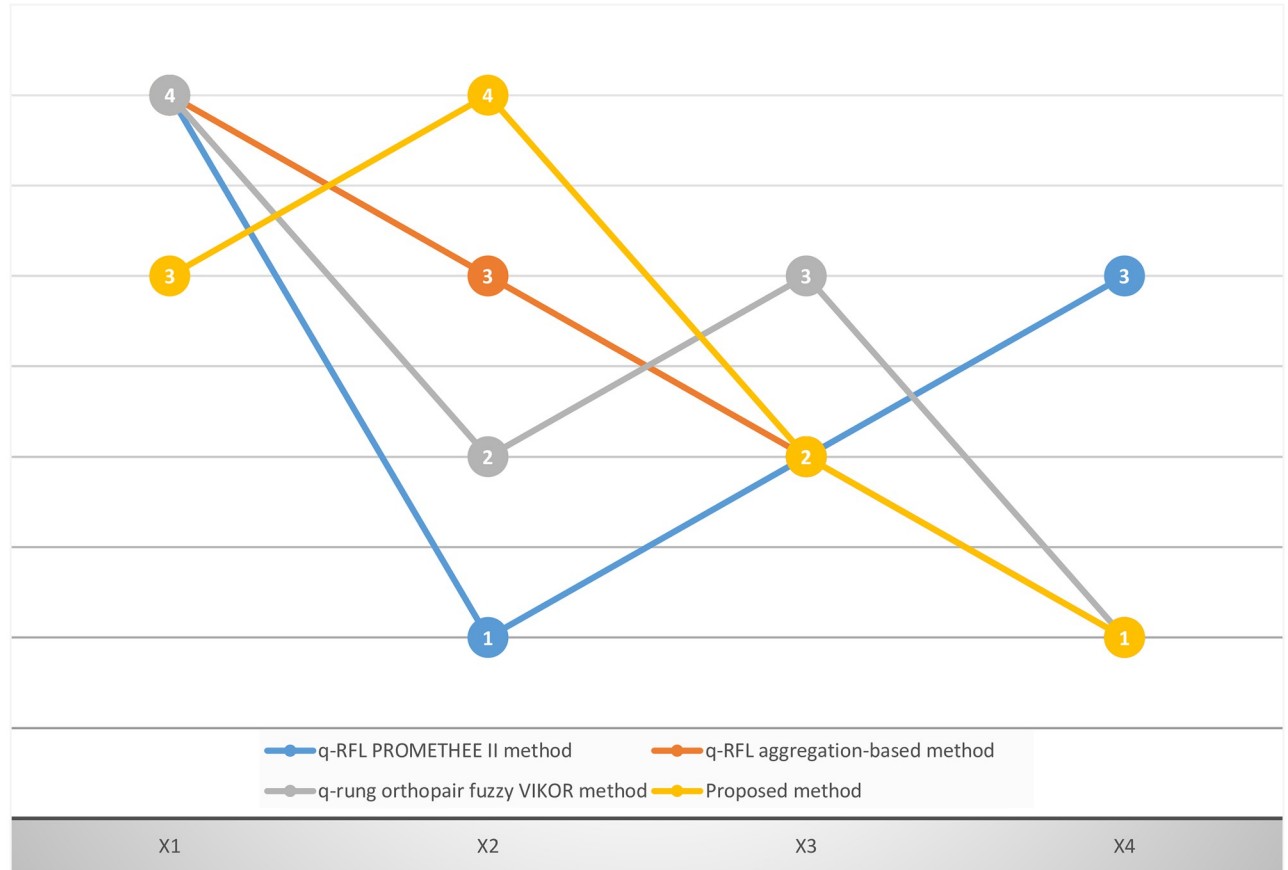

**Fig 1. Ranking of alternatives through different methods.**

of Cheng et al. [73]. It is basically developed on q-ROFSs, which only quantitatively process the uncertain data. This technique is unsuitable for modeling decision information when DMks prefer utilizing linguistic values.

### 7.4 Discussion

This introduced study is evaluated for its effectiveness in resolving MCGDM issues. Here is a list of the key advantages of the proposed strategy:

**i.** Our proposed method employs novel q-RFL AOs, whereas existing methods rely on the inefficient operational law (see Section 3 for details). Therefore, the presented approach is superior to extant q-RFL techniques.

**ii.** The proposed method employs revised operations based on weighted arithmetic or weighted geometric AOs (Eqs (12) and (26)) to aggregate the data provided by different DMKs. These AOs take into account the distinctive characteristics of q-RFLS, leading to a notably enhanced and resilient evaluation process.

**iii.** Unlike the existing approaches, the presented methodology introduces a formulation for ascertaining the weights associated with DMKs. Existing MCGDM methods either require

this information in advance or assign equal importance to all DMKs, a practice that lacks rationality.

**iv.** The suggested method is capable of solving decision-making problems in q-RFL environments with partially known or fully unknown weight information (detailed in Section 6). In contrast, the extant method [62] is only applicable for weight information that is entirely unknown. In addition, Ju et al. [39] method is restricted to known weight information and cannot be applied to q-RFL-based MCGDM problems with partially known or fully unknown weight information.

**v.** The ranking order is determined by combining the VIKOR and QUALIFLEX algorithms. On the one hand, the suggested method considers both the group utility and the individual regret simultaneously. The sensitivity study showed that it is robust. On the other hand, listing all of the available permutations is a simple and effective way to take into account many indexes.

**vi.** The developed method accepts evaluation values in both linguistic and numeric form, whereas the existing method [73] can only be used to calculate q-ROFNs but fails to account for q-RFLNs. q-RFLNs inherit the superiority of q-ROFNs and linguistic term sets and relaxes the membership and non-membership criteria. Consequently, information loss is inevitable when using q-ROFNs-based decision-making methodologies [73–75]. In the meantime, our devised method cannot only handle scenarios in which the weight information of the DMKs is unknown but can also reduce information loss.

However, the designed framework has an array of limitations that are outlined below:

**i.** The developed algorithm relies on an existing distance measure, but this measure has been found to exhibit certain drawbacks. For instance, consider the data points $\partial_1 = ((s_0, 0), \langle 0.38, 0.33 \rangle)$ and $\partial_2 = ((s_0, 0), \langle 0.3774, 0.06001 \rangle)$. When we calculate their distance using this existing measure, we obtain $d(\partial_1, \partial_2) = (1/14)|(1 + 0.38^2 - 0.33^2) \cdot 0 - (1 + 0.3774^2 - 0.06001^2) \cdot 0| = 0$. This result suggests that the two data points are identical, although their numeric values differ significantly.
This incongruity highlights a potential issue with the existing measure: it may lead to division by zero problems when attempting to compute distances between data points that have distinct numeric characteristics. Such anomalies can introduce inaccuracies and limitations into the algorithm's performance, potentially affecting the reliability of the results generated by the algorithm.

**ii.** Despite the fact that the revised operators have filled the gaps left by the existing ones, but these only work accurately when the minimum $q$ for which the condition $0 \leq v^q + \mu^q \leq 1$ ($q \geq 1$) meets is utilized. For larger $q$, these operators may produce the misleading result, for example, consider three q-RFLNs $\partial_1 = ((s_6, 0), \langle 0.60, 0.45 \rangle)$ $\partial_2 = ((s_5, 0), \langle 0.60, 0.45 \rangle)$, $\partial_3 = ((s_6, 0), \langle 0.58, 0.42 \rangle)$ and weight vector $w = (0.3302, 0.3624, 0.3074)^t$. In this case, the smallest $q$ for which the said condition holds is 2. Using 2-ROPFLWA on $\partial_1$, $\partial_2$, and $\partial_3$, we obtain $((s_6, -0.2856), \langle 0.5935, 0.04438 \rangle)$, which is acceptable. However, if we apply 4-ROPFLWA to $\partial_1$, $\partial_2$, and $\partial_3$, we obtain $((s_6, -0.2856), \langle 0.5941, -0.002528 \rangle)$, which is nonsensical due to the negative sign of non-membership part.

**iii.** The presented weight determination models consider only the objective aspect of weight information and disregard the subjective aspect, rendering the criteria weight ineffective.

Next, we conduct a statistical analysis to gauge the consistency of rankings in the context of the MCGDM problem. We employ a widely recognized measure, namely Spearman's Rank

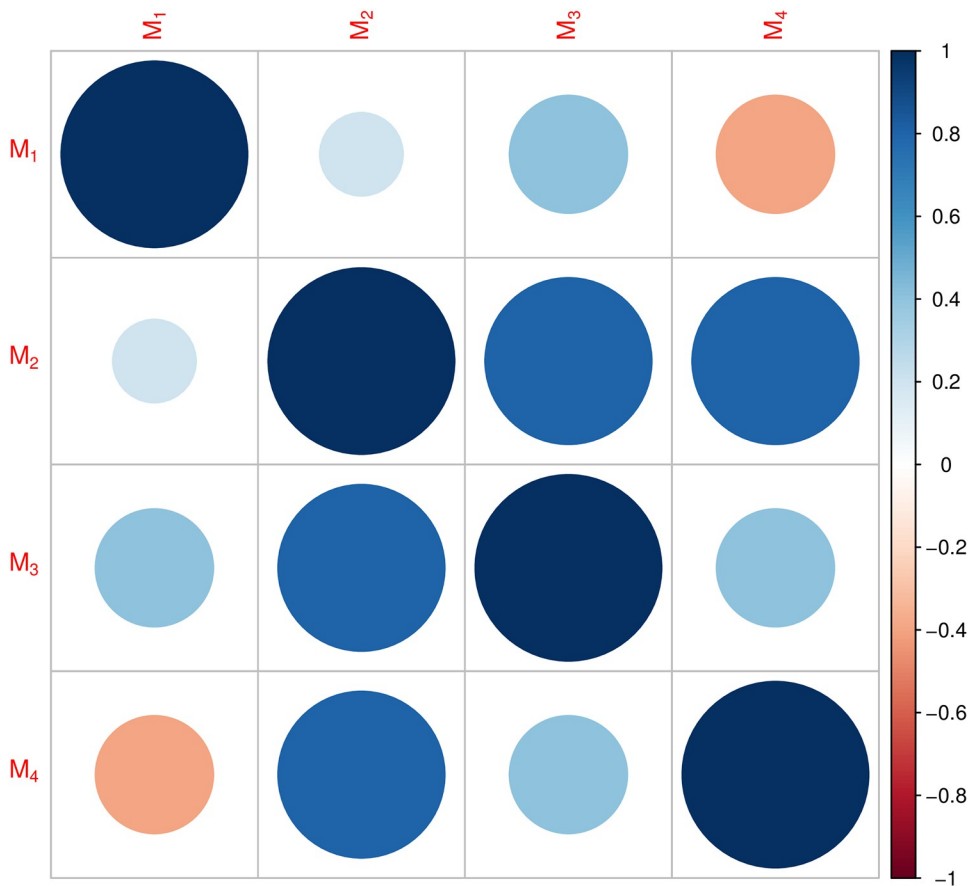

**Fig 2. Spearman's Rank Correlation plot.**

Correlation Coefficient, to assess the consistency of these rankings. The calculation of the Spearman's Rank Correlation Coefficient is carried out using the following formula:

$$ß = 1 - \frac{6}{n(n^2 - 1)} \sum_{i=1}^{n} d_i^2,$$ (59)

where, $n$ signifies the number of results and $d_i$ corresponds to the differences in ranking between the results.

From Fig 2, it's evident that the correlation values obtained are deemed highly valid, given their positive threshold, except the outcome generated by the approach outlined in Reference [62].

## 8 Conclusions

The rapid expansion of the BS industry has elevated the significance of choosing recycling suppliers for BS materials. In this context, supplier selection emerges as a multifaceted MCGDM challenge characterized by numerous alternatives and competing criteria. In response to this challenge, we developed an innovative decision-making approach integrating the VIKOR and QUALIFLEX methods. Primarily, we undertook a comprehensive review of the existing operational laws of q-RFLNs via counter-examples to scrutinize their shortcomings. To cover these

limitations, we revised the existing q-RFL operations and emphasized their advantages through illustrative examples (please refer to Examples 5-8). A series of AOs, including the q-RFLWA operator, the q-RFLWG operator, the q-RFLOWA operator, the q-RFLOWG operator, the q-RFLHWA operator, and the q-RFLHG operator have been put forward based on the revised operational laws. Further, we developed two optimization models within the framework of q-RFL for MCGDM problems in scenarios where the criteria weights are either unknown or partially incomplete. Further, we developed two optimization models under q-RFL context to address MCGDM problems where the criteria weights are either unknown or only partially available. This approach offers significant practical applicability, as it effectively addresses a wide range of scenarios encountered in real-world decision-making processes where complete criteria weight information is often not provided or remains partially incomplete. Following these developments, we constructed an integrated approach, namely VIKOR-QUALIFLEX, within the q-RFL framework and subsequently applied it to select recycling suppliers. To validate the stability of our integrated approach, we conducted an in-depth sensitivity analysis, demonstrating the strong robustness of the proposed methodology. Finally, we selected some representative algorithms for comparison, and the results of the comparative study underscore the effectiveness of the presented approach.

## Author Contributions

**Conceptualization:** Jawad Ali.

**Data curation:** Wali Khan Mashwani.

**Formal analysis:** Fatima Abbas.

**Funding acquisition:** Muhammad I. Syam.

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
