## [Decision Letter · Decision Letter 0]

17 Sep 2023

PONE-D-23-18574q-rung orthopair fuzzy 2-tuple linguistic VIKOR-QUALIFLEX decision analysis method with partial weight informationPLOS ONE

Dear Dr. ali,

Thank you for submitting your manuscript to PLOS ONE. After careful consideration, we feel that it has merit but does not fully meet PLOS ONE’s publication criteria as it currently stands. Therefore, we invite you to submit a revised version of the manuscript that addresses the points raised during the review process.

We look forward to receiving your revised manuscript.

Kind regards,

Fausto Cavallaro, PhD

Academic Editor

PLOS ONE

Reviewers' comments:

Reviewer's Responses to Questions

**Comments to the Author**

1. Is the manuscript technically sound, and do the data support the conclusions?

Reviewer #1: Yes

Reviewer #2: Yes

2. Has the statistical analysis been performed appropriately and rigorously? 

Reviewer #1: Yes

Reviewer #2: No

3. Have the authors made all data underlying the findings in their manuscript fully available?

Reviewer #1: Yes

Reviewer #2: Yes

4. Is the manuscript presented in an intelligible fashion and written in standard English?

Reviewer #1: Yes

Reviewer #2: Yes

5. Review Comments to the Author

Reviewer #1: In this study, authors proposed an q-rung orthopair fuzzy 2-tuple linguistic VIKOR-QUALIFLEX decision analysis method with partial weight information. It is an interesting study. I think that it can be accepted after major revisions.

(1) Authors are suggested to discuss the existing MCDM approach that are used in choosing a BS recycling supplier. The problems of the existing studies should be clearly discussed for motivating this study.

(2) There are many equations and mathematical symbols in this version. Authors are suggested to check them carefully.

(3) The existing operational laws of q-RFLSs should be compared with the proposed operational laws.

(4) Some recent studies about mcdm method should be included: Picture fuzzy interactional partitioned Heronian mean aggregation operators: an application to MADM process; Medical waste treatment scheme selection based on single-valued neutrosophic numbers; Pythagorean fuzzy MULTIMOORA method based on distance measure and score function: its application in multicriteria decision making process; Evaluating IoT platforms using integrated probabilistic linguistic MCDM method; Linguistic q‐rung orthopair fuzzy sets and their interactional partitioned Heronian mean aggregation operators; TOPSIS method based on correlation coefficient and entropy measure for linguistic Pythagorean fuzzy sets and its application to multiple attribute decision making.

(5) The advantages and disadvantages of the proposed method should be clearly described.

Reviewer #2: Below are detailed comments and suggestions for each section of the paper:

-Rewrite the abstract. Also, clearly describe the contributions in the Abstract section.

-Clearly describe the motivation in the Introduction section of this paper. The Scientific merit and novelty of the article are not clear. The authors should explain clearly in the abstract what is the novelty of the proposed method and what is the added value in this article?

-Imbalanced introduction to clearly show the limitation of previous works, the gap, the methodology

-Why you use VIKOR-QUALIFLEX approach? Give logical reasons.

-There are many typos in this manuscript. The authors should check typing errors throughout the manuscript. English style should also be improved.

-Send me all codes and data from the results

-Scientific discussion in terms of accuracy performance, validation data, verification metrics, evidential

analysis and comparison with other methods.

-The lack of any data analysis/data visualization in considering the features selection are obvious.

-Any explicit discussion to illustrate the limitations, pitfalls and practical difficulties of applied models

under certainty???? Prior impact assessment and cost-benefit analysis and possible consequences in

advance?? How did you verify the fairness of made decisions while contextually has different

perspectives depending on the particular feed inputs? Can this model provide correct inferences and

some explanation for the underlying phenomena???? Can users gain a mechanistic understanding of

that???? The inclusion of the uncertainty for clean datasets is high.

-The conclusions should be more carefully rewritten, summarizing what has been learned and why it is

interesting and useful.

-Expand literature review by citing related work in MADM under fuzzy set extensions such as:

10.1007/s40314-023-02254-5,10.22105/jfea.2020.247946.1004, 10.1007/s00500-021-05771-9, 10.22105/riej.2020.216548.1117, 10.1177/16878132231161485, 10.22105/jarie.2017.95312.1017, 10.1007/s40815-022-01401-0, 10.3934/math.2023577,10.22105/jfea.2022.335045.1214, 10.1016/j.nima.2020.164839, 10.56578/josa010203, 10.3390/math9010037

6. PLOS authors have the option to publish the peer review history of their article (what does this mean?). If published, this will include your full peer review and any attached files.

Reviewer #1: No

Reviewer #2: No

---

## [Author Response · Author response to Decision Letter 0]

30 Nov 2023

Dear Editor:

We are thankful to anonymous reviewers for the careful consideration of the manuscript and value able suggestions. The manuscript has been revised accordingly; see the detailed replies below.

Reviewer 1:

Comment 1: Authors are suggested to discuss the existing MCDM approach that are used in choosing a BS recycling supplier. The problems of the existing studies should be clearly discussed for motivating this study.

Reply: As per your insightful suggestion, we have discussed the existing MCDM approaches regarding BS recycling supplier.

Comment 2: There are many equations and mathematical symbols in this version. Authors are suggested to check them carefully.

Reply: We have checked all the equations and symbols carefully as per your directives.

Comment 3: The existing operational laws of q-RFLSs should be compared with the proposed operational laws.

Reply: Some further examples have been added for comparison purpose. Please refer to Examples 1 to 8.

Comment 4: Some recent studies about mcdm method should be included: Picture fuzzy interactional partitioned Heronian mean aggregation operators: an application to MADM process; Medical waste treatment scheme selection based on single-valued neutrosophic numbers; Pythagorean fuzzy MULTIMOORA method based on distance measure and score function: its application in multicriteria decision making process; Evaluating IoT platforms using integrated probabilistic linguistic MCDM method; Linguistic q‐rung orthopair fuzzy sets and their interactional partitioned Heronian mean aggregation operators; TOPSIS method based on correlation coefficient and entropy measure for linguistic Pythagorean fuzzy sets and its application to multiple attribute decision making.

Reply: We have done accordingly.

Comment 5: The advantages and disadvantages of the proposed method should be clearly described.

Reply: Thank you for your suggestion. The merits and demerits of the proposed methodology have been described in a clearer way.

Reviewer 2:

Comment 1: Rewrite the abstract. Also, clearly describe the contributions in the Abstract section.

Reply: Thank you for this indication, we have done the required changes.

Comment 2: Clearly describe the motivation in the Introduction section of this paper. The Scientific merit and novelty of the article are not clear. The authors should explain clearly in the abstract what is the novelty of the proposed method and what is the added value in this article?

Reply: In the revised version of the article, we have made several improvements to address these concerns:

Introduction Section: We have provided a more explicit and comprehensive description of the motivation behind our research in the Introduction section. This serves to clearly outline the problem or question we aim to address and why it is relevant, setting the stage for the reader to understand the context and significance of our work.

Scientific Merit and Novelty: In the revised manuscript, we have made a concerted effort to elucidate the scientific merit and novelty of our article. We have expanded upon the unique contributions and innovations of our proposed methodology, highlighting how it advances the current state of knowledge in the field. By doing so, we aim to convey a clearer sense of why our research is valuable and original.

Abstract: We have restructured and refined the abstract to explicitly convey the novelty of our proposed method and its added value in the context of the article. This includes succinctly articulating the key innovations and benefits of our research, ensuring that readers can readily grasp the significance of our work from the abstract alone.

Comment 3: Imbalanced introduction to clearly show the limitation of previous works, the gap, the methodology.

Reply: Thank you for this constructive comment. To address the raised issue, we have revisited and revised the "Challenges" section of the article.

Comment 4: Why you use VIKOR-QUALIFLEX approach? Give logical reasons.

Reply: VIKOR-QUALIFLEX represents an integrated approach that combines the strengths of both VIKOR and QUALIFLEX methodologies.

VIKOR, renowned for its capacity to provide a compromise solution, addresses the need to strike a balance between conflicting criteria. It aims to identify alternatives that offer reasonable trade-offs between the best and worst performances, ultimately delivering practical and well-balanced solutions that align with the preferences of decision-makers.

On the other hand, QUALIFLEX is distinguished by its proficiency in solving MCDM problems, particularly when the number of criteria substantially outweighs the number of available alternatives. QUALIFLEX relies on effective outranking methods, involving pair-wise comparisons of alternatives for each criterion across all possible permutations. It discerns the optimal permutation that maximizes the concordance/discordance index, with a notable ability to appropriately handle both cardinal and ordinal information.

However, VIKOR alone may not yield a unique solution. To overcome this limitation and harness the advantages of both VIKOR and QUALIFLEX, the integrated VIKOR-QUALIFLEX approach has been devised. This methodology provides a comprehensive solution that capitalizes on the strengths of each individual method, ensuring a more robust and effective approach to multi-criteria decision-making problems.

Comment 5: There are many typos in this manuscript. The authors should check typing errors throughout the manuscript. English style should also be improved.

Reply: We apologize for any previous errors, and we have diligently addressed them by eliminating all typos and enhancing the overall language quality.

Comment 6: Send me all codes and data from the results

Reply: We have attached all the Maple files to provide comprehensive details for the computational work. Please find these attachments.

Comment 7: Scientific discussion in terms of accuracy performance, validation data, verification metrics, evidential analysis and comparison with other methods.

Reply: Thank you for your comment. Please refer to Section 3, Section 6.3 and Section 7.

Comment 8: The lack of any data analysis/data visualization in considering the features selection are obvious.

Reply: Selecting the right recycling service supplier is not a straightforward task, particularly within the context of today's fiercely competitive business landscape. The decision-making process involved can be likened to a Multi-Criteria Group Decision-Making (MCGDM) problem, as it necessitates the consideration of a multitude of criteria and factors. In this intricate evaluation, it's imperative to weigh various aspects of each supplier's offerings, making the selection process anything but obvious.

Moreover, to ensure a thorough and accurate assessment of potential suppliers, experts have contributed their insights using a specialized approach: q-rung orthopair fuzzy 2-tuple linguistic terms. This distinctive method of data representation serves a dual purpose. Firstly, it helps capture the nuanced and multifaceted nature of the criteria involved, allowing for a more precise and detailed analysis. Secondly, it minimizes the risk of information loss during the data collection and evaluation phases.

By adopting this approach, decision-makers are better equipped to navigate the intricacies of supplier selection in the recycling industry, ultimately leading to more informed and well-founded decisions in a highly competitive environment.

Comment 9: Any explicit discussion to illustrate the limitations, pitfalls and practical difficulties of applied models

under certainty???? Prior impact assessment and cost-benefit analysis and possible consequences in

advance?? How did you verify the fairness of made decisions while contextually has different

perspectives depending on the particular feed inputs? Can this model provide correct inferences and

some explanation for the underlying phenomena???? Can users gain a mechanistic understanding of

that???? The inclusion of the uncertainty for clean datasets is high.

Reply: The limitations of the proposed approach are given on Page 40.

Performing prior impact assessments and cost-benefit analyses, while anticipating potential consequences in advance, is a strategic and informed approach that enhances decision-making, mitigates risks, and ensures compliance with regulations. It also fosters transparency, resource allocation efficiency, and long-term planning, promoting responsible and sustainable choices.

We ensured the fairness of our decisions by employing methods that consider various perspectives based on the specific feed inputs. While the same data was used for all methods, it's important to note that existing methods have limitations in their underlying theories, as outlined in Section 3, and several weaknesses were identified towards the end of Section 7. To bolster our evaluation, we also conducted a rigorous statistical analysis test. This comprehensive approach allowed us to make more robust and informed decisions that take into account the shortcomings of existing methodologies.

Each method has its merits and demerits when it comes to providing correct inferences and explanations for underlying phenomena. In the case of the proposed method, it has proven to be quite helpful in obtaining accurate results, as evidenced by Figure 2, which shows that our derived results align well with the existing ones. However, it's important to acknowledge that no method is perfect.

The developed method does have its drawbacks, as outlined in Section 7. These limitations may affect its applicability in certain contexts. Nevertheless, for users who are decision experts, the method can offer a mechanistic understanding of the underlying phenomena. This means that individuals with expertise in decision-making can indeed gain a deeper understanding of the processes involved, even while being mindful of the method's limitations.

Comment 10: The conclusions should be more carefully rewritten, summarizing what has been learned and why it is interesting and useful.

Reply: We have completely revised the conclusion section keeping in mind your valuable suggestions.

Comment 11: Expand literature review by citing related work in MADM under fuzzy set extensions such as:

10.1007/s40314-023-02254-5,10.22105/jfea.2020.247946.1004, 10.1007/s00500-021-05771-9, 10.22105/riej.2020.216548.1117, 10.1177/16878132231161485, 10.22105/jarie.2017.95312.1017, 10.1007/s40815-022-01401-0, 10.3934/math.2023577,10.22105/jfea.2022.335045.1214, 10.1016/j.nima.2020.164839, 10.56578/josa010203, 10.3390/math9010037

Reply: The suggested references have been cited for comparison and review purposes.

 Thanking you once again for your time and suggestions.

---

## [Decision Letter · Decision Letter 1]

8 Jan 2024

An integrated group decision-making method under q-rung orthopair fuzzy 2-tuple linguistic context with partial weight information

PONE-D-23-18574R1

Dear Dr. ali,

We’re pleased to inform you that your manuscript has been judged scientifically suitable for publication and will be formally accepted for publication once it meets all outstanding technical requirements.

Kind regards,

Fausto Cavallaro, PhD

Academic Editor

PLOS ONE

Additional Editor Comments (optional):

The paper was revised according to reviewer's comments. It can be accepted.

Reviewers' comments:

Reviewer's Responses to Questions

**Comments to the Author**

1. If the authors have adequately addressed your comments raised in a previous round of review and you feel that this manuscript is now acceptable for publication, you may indicate that here to bypass the “Comments to the Author” section, enter your conflict of interest statement in the “Confidential to Editor” section, and submit your "Accept" recommendation.

Reviewer #1: All comments have been addressed

Reviewer #2: All comments have been addressed

2. Is the manuscript technically sound, and do the data support the conclusions?

Reviewer #1: Yes

Reviewer #2: Yes

3. Has the statistical analysis been performed appropriately and rigorously? 

Reviewer #1: Yes

Reviewer #2: N/A

4. Have the authors made all data underlying the findings in their manuscript fully available?

Reviewer #1: Yes

Reviewer #2: Yes

5. Is the manuscript presented in an intelligible fashion and written in standard English?

Reviewer #1: Yes

Reviewer #2: Yes

6. Review Comments to the Author

Reviewer #1: It has been well improved. I think that it can be accepted in current form. I did not have any comments now.

Reviewer #2: I want to express my gratitude to the authors for addressing my concerns. I am pleased to inform you that, in its current form, I believe the manuscript is suitable for acceptance.

7. PLOS authors have the option to publish the peer review history of their article (what does this mean?). If published, this will include your full peer review and any attached files.

Reviewer #1: No

Reviewer #2: No

---

## [Editor Report · Acceptance letter]

26 Apr 2024

PONE-D-23-18574R1 

PLOS ONE

Dear Dr. Ali, 

I'm pleased to inform you that your manuscript has been deemed suitable for publication in PLOS ONE. Congratulations! Your manuscript is now being handed over to our production team.

Kind regards, 

on behalf of

Professor Fausto Cavallaro 

Academic Editor

PLOS ONE